# Comparison of Sea Ice Kinematics at Different Resolutions Modeled with a Grid Hierarchy in Community Earth System Model (version 1.2.1)

Shiming Xu[1,2], Jialiang Ma[1], Lu Zhou[1], Yan Zhang[1], Jiping Liu[3], and Bin Wang[1,3]

[1]Ministry of Education Key Laboratory for Earth System Modeling, Department of Earth System Science (DESS), Tsinghua University, Beijing, China
[2]University Corporation for Polar Research (UCPR), Beijing, China
[3]State Key Laboratory of Numerical Modeling for Atmospheric Sciences and Geophysical Fluid Dynamics (LASG), Institute of Atmospheric Physics, Chinese Academy of Sciences, Beijing, China

*Correspondence to:* Shiming Xu (xusm@tsinghua.edu.cn)

**Abstract.** High-resolution sea ice modeling is becoming widely available for both operational forecasts and climate studies. In traditional Eulerian grid based models, small-scale sea ice kinematics is the most prominent feature of high-resolution simulations, and with rheology models such as Viscous Plastic (VP) and Maxwell Elasto-Brittle (MEB), sea ice models are able to reproduce multi-fractal sea ice deformation and linear kinematic features that are witnessed in high-resolution observational datasets. In this study, we carry out modeling of sea ice with multiple grid resolutions by using the Community Earth System Model (CESM) and a grid hierarchy (22 km, 7.3 km, and 2.4 km grid stepping in the Arctic). By using atmospherically forced experiments, we simulate consistent sea ice climatology across the 3 resolutions. Furthermore, the model reproduces reasonable sea ice kinematics, including multifractal spatial scaling of sea ice deformation that partially depends on atmospheric circulation pattern and forcings. By using high-resolution runs as references, we evaluate the model's effective resolution with respect to the statistics of sea ice kinematics. Specifically, we find the spatial scale at which the PDF of the scaled sea ice deformation rate of low-resolution runs match that of high-resolution runs. This critical scale is treated as the effective resolution of the coarse resolution grid, which is estimated to be about 6 to 7 times of the grid's native resolution. Besides, we show that in our model, the convergence of the Elastic-Viscous-Plastic (EVP) rheology scheme plays an important role in reproducing reasonable kinematics statistics, and more strikingly, simulates systematically thinner sea ice than the standard, non-convergent experiments in landfast ice regions of Canadian Arctic Archipelago. Given the wide adoption of EVP and subcycling settings in current models, it highlights the importance of EVP convergence especially for climate studies and projections. The new grids and the model integration in CESM are openly provided for public use.

# 1 Introduction

Sea ice is the interface between the polar atmosphere and ocean, and therefore an important modulating factor of the polar air-sea interactions. The momentum input into the sea ice from the atmosphere and the ocean causes sea ice drift, as well as failures and deformations at a wide range of temporal and spatial scales. As revealed by high-resolution, kilometer-scale sea ice drift and deformation estimates with Synthetic Aperture Radars, the deformation of the sea ice is shown to be multi-fractal, with scale-invariance properties (Marsan et al., 2004; Rampal et al., 2008; Weiss and Dansereau, 2017) and quasi-linear kinematic features are observed through visual inspections (Kwok et al., 2008), including local deformation regions of sea ice failures and shearing. Furthermore, the these kinematic features are accompanied by the formation of sea ice leads and pressure ridges, and tightly coupled to sea ice and polar thermodynamic processes. While sea ice leads are hot-spots of heat and moisture fluxes during winter, sea ice ridging is responsible for producing the thickest sea ice in both polar regions.

Sea ice rheology models provide mathematical description and numerical treatments for sea ice dynamic processes, and thus essential components of sea ice models. Viscous-Plastic [VP, (Hibler, 1979)] is the most widely used rheology model, and it describes the sea ice as a two dimensional continuum with nonlinear viscosity which undergoes plastic deformations over critical shearing and compressive stresses. In order to overcome the numerical stiffness of the VP model, modelers usually adopt explicit solvers for the derived models of VP such as Elastic-Viscous-Plastic [EVP, (Hunke and Dukowicz, 1997)] instead of the original VP, or implicit solvers (Hibler, 1979; Lemieux et al., 2010). With the introduction of an artificial term of elastic waves into the VP model, the numerical solving process is transformed into an explicit formulation. An iterative process, called subcycling, is carried out in EVP in order to numerically attenuate the elastic wave and attain the solution to VP model. Due to its good numerical stability and straightforward implementation, EVP is adopted by many climate models participating in Coupled Model Intercomparison Projects (CMIP6), as well the hindcast experiments for operational sea ice forecasts (Dupont et al., 2015).

With VP rheology, the capability of sea ice models to resolve fine-scale deformations is inherently bounded by the resolution of the models' grids. In order to reproduce the observed properties of the sea ice kinematics, grids of $0.1°$ resolution or finer in sea ice regions are usually required. State-of-the-art high-resolution studies reach 2 km or finer grid stepping in the Arctic (Hutter et al., 2018; Scholz et al., 2019). Although the continuum assumption of the sea ice cover in VP (or EVP) does not necessarily hold at these resolutions, VP models are shown to be capable in reproducing realistic sea ice kinematics (Hutter et al., 2018). Unstructured grid based models and Lagrangian models have unique capabilities in modeling sea ice. Regionally focused studies can be easily carried out with variable grid sizes, such as the Arctic simulation with FESOM (Wang et al., 2018; Koldunov et al., 2019). Purely Lagrangian models such as neXtSIM (Rampal et al., 2019) are potentially free of the resolution issues for resolving small deformation features. Specifically, neXtSIM utilizes Maxwell elasto-brittle rheology (Dansereau et al., 2016; Girard et al., 2011), which is shown to simulate reasonable sea ice kinematics and scaling properties even at moderate resolution settings (Rampal et al., 2019). Traditional VP rheology is limited due to the lack of memory for past deformation events (Hutter and Losch, 2020). Besides, there are also efforts in improving the rheology model in simulating observed anisotropy in sea ice floe shape and associated deformation (Tsamados et al., 2013).

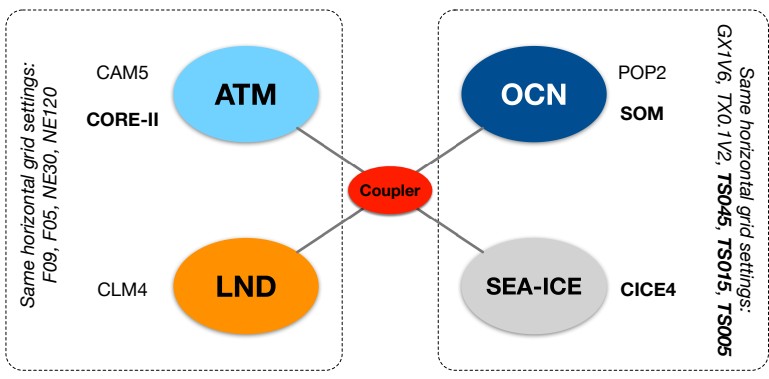

**Figure 1.** Ocean and sea ice grids and air-sea coupling in Community Earth System Model (CESM).

In this study, we carry out comparison of sea ice model simulations at different spatial resolutions with the coupled model of Community Earth System Model (CESM, version 1.2.1). CESM (http://www.cesm.ucar.edu) is developed at National Center for Atmospheric Research (NCAR) and adopted by various research groups in the world for climate studies. The component models of CESM include: Community Atmospheric Model (CAM), Parallel Ocean Program (POP, version 2), Community

Ice CodE model (CICE, version 4) and Common Land Model (CLM). The coupling between these components is carried out with the flux coupler CPL. Components can be configured to the specific experiment needs, such as atmospherically forced ocean and (or) sea ice simulations. Figure 1 shows the the coupling schematics in CESM and the common configuration of high-resolution coupled runs. The horizontal model grids (and associated spatial resolutions) of CICE and POP are the same, and the two standard configurations in CESM are: nominal 1° (GX1V6, a dipolar grid) and 0.1° (TX0.1v2, a tripolar grid).

Specifically, CICE, the sea ice component of CESM, includes comprehensive thermodynamic and dynamic processes of sea ice, including discretized ice thickness distribution, prognostic enthalpy, complex shortwave albedo and penetration schemes with snow and pond processes, EVP model, ridging parameterization, etc. CICE in CESM can be run in three major settings: (1) forced by with atmospheric reanalysis (NCEP CORE-2) with coupling to a slab ocean model (SOM), (2) ice-ocean coupled simulation under atmospheric forcings, and (3) fully atmosphere-ocean-ice coupled runs.

In our study, we design a grid hierarchy for the ocean and the sea ice model and incorporate them in CESM, including its component models of CICE and POP, as well as the coupling to the atmospheric forcings. The grid hierarchy includes three tripolar grids with the nominal resolution of 0.45°, 0.15° and 0.05°, covering the wide range of climate modeling and sub-mesoscale oriented studies. In Section 2 we introduce the grid generation method, and the integration of the grid hierarchy in CESM. Furthermore, atmospheric forcing based experiments are carried out with the new grids, and Section 3 includes the

details of the experiments and the analysis of modeled sea ice climatology and kinematics. Especially, we carry out scaling analysis and cross-resolution comparisons of the sea ice kinematics, and study the convergence behavior of EVP. In Section 4 we summarize the article and provide discussion of related topics and future research directions including multi-scale modeling.

## 2 Grid Generation and Model Integration

### 2.1 TS grids – a tripolar grid hierarchy

We design a new grid hierarchy for global ocean-sea ice modeling. The generated grids are orthogonal and compatible with many existing ocean and sea ice models, including POP, CICE, and MOM. Following the methodology in Murray (1996) and Madec and Imbard (1996), we design and implement a grid generation method that ensures global smooth transition of grid scales and supports direct model integration in POP and CICE. As shown in Figure 2, it consists of two patches, the southern patch (SP) and the northern patch (NP), divided by a certain latitudinal circle ($\phi$) in the northern hemisphere. For the SP, the grid lines are purely zonal or meridional. For the NP, there are two north poles for the grid, which are placed on the Eurasia and North America land masses. However, unlike many tripolar grids such as TX0.1V2 in CESM, this new grid features smooth grid scale transition on the boundary of SP and NP. In order to achieve this, a non-trivial grid generation process is carried out on the stereographic projection of NP. Specifically, a series of embedded ellipses are constructed based on resolution requirements, with: (1) the outermost ellipse as the projection of the latitudinal circle at $\phi$ (i.e., a circle), and (2) the innermost ellipse as the line linking the two grid poles. The foci of the outermost ellipse are both at the North Pole, and with the progression to inner ellipses, they gradually move towards the two grid poles, respectively. The ellipses form the "zonal" grid lines in NP. After the ellipses are constructed, the "meridional" grid lines in NP are constructed, starting from the southern boundary of NP, down to the innermost ellipse. During this process, it is ensured that: (1) the grid lines are constructed consecutively between adjacent ellipses, with starting points on the outer ellipse; and (2) they are linked with the meridional grid lines in SP to ensure overall continuity of the global grid. The details of the construction of ellipses and the smooth transition of meridional grid scales are further described in Appendix A. Furthermore, the meridional grid scales in both SP and NP are constructed to alleviate the grid aspect ratio, reducing meridional grid sizes in higher latitudes.

By using the grid generation method, we generate a series of tripolar grids: TS045, TS015 and TS005. These grids all have the the boundary between SP and NP ($\phi$) at $10°N$, and grid poles at: $(63°N, 104°W)$ and $(59.5°N, 76°E)$. Table 1 shows the detailed configuration of these grids. TS045 is the coarsest grid with the nominal resolution of $0.45°$, and it targets at climate modeling and the typical resolution range for ocean component models in Coupled Model Intercomparison Projects (CMIP6). TS015 and TS005 are about 3 times (or nominal $0.15°$) and 9 times (or nominal $0.05°$) the resolution of TS045, respectively.

Figure 3 shows the horizontal grid scale ($s = \sqrt{(dx^2 + dy^2)/2}$) in polar regions for TS015 and TS005. The average grid scale in TS005 is 2.45 km in the Arctic oceanic regions (north of $65°N$), and the scales of TS015 and TS045 are 7.34 km and 22.01 km, respectively. From the ocean modeling perspective, we use the Rossby deformation radii ($R$) as the proxy for mesoscale (Chelton et al., 1998), and investigate the capabilities of each grid. Specifically, the criterion in Hallberg (2013) is adopted: the mesoscale resolving is attained when $s$ is smaller than the half of the local value of $R$. Based on the annual mean WOA13 climatology of salinity and temperature (Locarnini et al., 2013; Zweng et al., 2013), we construct the global distribution of $R$. In Table 1 we note that TS015 is "almost eddying", because this grid attains mesoscale-resolving for 65% of the global ocean's area. As outlined in Figure 3, in polar regions, the ratio of $s$ to $R$ is all higher than 0.5 for TS015, indicating

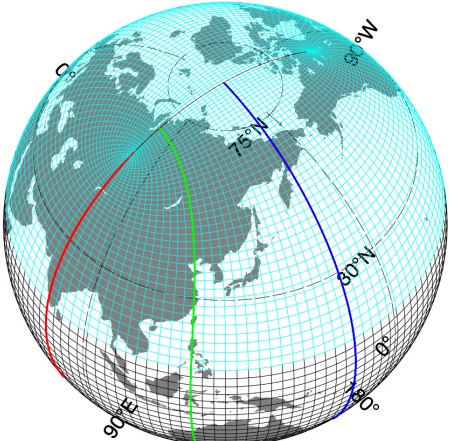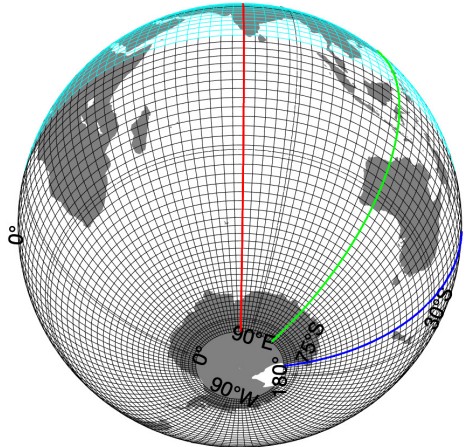

**Figure 2.** Global tripolar TS grid with northern patch (in cyan) and southern patch (in black). 0.05° grid (TS005) is shown (1 in every 60 grid points). On the boundary of the two patches, smooth grid size change is ensured in the meridional direction. Typical meridional grid lines are also shown in thick lines (red, green and blue).

**Table 1.** TS grid hierarchy

| Name | Nominal Resolution (°) | Dimension (i & j) | Zoom Level | Notes |
|------|------------------------|-------------------|------------|-------|
| TS045 | 0.45 | 800 × 560 | 1 | Long-term, climate simulation |
| TS015 | 0.15 | 2400 × 1680 | 3 | Eddy-resolving in 65% oceanic area ("almost eddying") |
| TS005 | 0.05 | 7200 × 5040 | 9 | Fine-scale, submesoscale ocean modeling |

no mesoscale resolving capability for this grid for these regions. But for TS005, the mesoscale processes in polar regions with relatively deep bathymetry (e.g., within the Arctic Ocean) can be resolved.

## 2.2 TS grid integration in CESM

The integration of the TS grids in CESM involves the following 3 steps. First, the generation of land-sea distribution and
5    bathymetry, as well as the technical implementation of the grids in both POP and CICE. Model bathymetry is generated at the

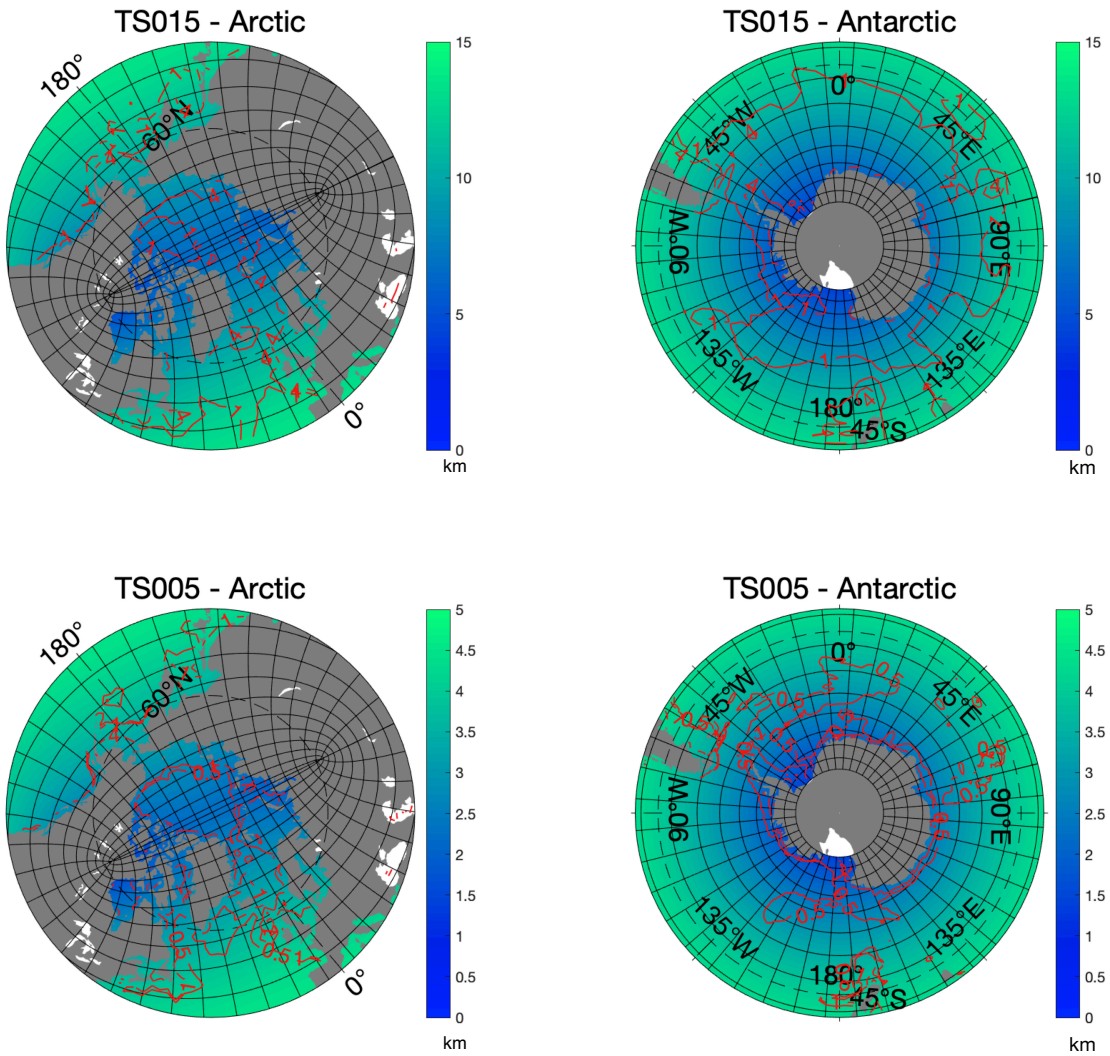

**Figure 3.** Horizontal grid scale of TS015 and TS005 in the polar regions. For grid lines, 1 in 60 (or 180) points are shown for TS015 (or TS005) grid. $\sqrt{(dx^2 + dy^2)/2}$ is shown by filled contour (in kilometers), and regions of mesoscale-resolving capability ($s/R < 0.5$) outlined by red contour lines.

grid locations and 60 vertical layers, based on ETOPO1 dataset (ETO). The vertical coordinate consists of 10-meter equal-depth layers in the top 200 meters, with gradual increase of layer depth to 250 meters in the deep oceans (up to 5500 meters).

Second, we configure the model according to the grid resolution, including the choice of parameterization schemes and related parameters. Specifically, we adopt the full thermodynamic and dynamic model processes in CICE, mainly following the standard configurations of parameterization schemes in Hunke and Lipscomb (2008). The main processes relating to sea ice dynamics include: EVP rheology model (Hunke and Dukowicz, 1997), the ridging/rafting scheme (Lipscomb et al., 2007) and ice strength model (Hibler, 1979), and transport-remapping based advection (Dukowicz and Baumgardner, 2000). Model configuration and parameters are aligned across the three grids, with details shown in Appendix B. The major difference among the grids is that we choose shorter thermodynamics and dynamics time steps for grids with higher resolution (Tab. 2). Furthermore, since sea ice kinematics is the focus of this study, different EVP subcycle numbers are chosen for each grid. While 120 subcycles per hour is adopted for some 1° resolution CMIP simulations (Jahn et al., 2012; Xu et al., 2013), we experiment with larger values of cycle counts (up to 960 subcycles per dynamic time step), as shown in Table 2.

In our study, CICE is coupled to the slab ocean model (SOM) in CESM, which provides a climatological seasonal cycle of ocean mixed layer depth and heat potential. The same configuration for SOM is used for experiments with TS045, TS015 and TS005. The reason why we use SOM instead of a fully dynamic-thermodynamic ocean model (such as POP) is three fold. First, in this study we focus on the simulation of sea ice kinematics and the inter-comparison across different resolution settings. Therefore, by using a single-column model for the ocean, we eliminate the factors that may compromise the comparability, including the inconsistency in modeled ocean processes across the resolution range, ocean and coupled turbulence, etc. Second, since atmospheric forcing is the major driver of sea ice drift and kinematics, we consider SOM eligible for the purpose of this study. Third, using SOM with CICE in CESM greatly alleviate the computational overhead for long-term simulations, especially for TS005 (0.05° grid). As is shown in the next section, with CICE coupled to SOM, we simulate comparable Arctic sea ice climatology and kinematic features (cracking events, etc.) among TS045, TS015 and TS005, and the computational cost and time-to-solution remains manageable. Potential compromises pertaining to the use of SOM are discussed further in Section 4.

Fourth, we force the sea ice component with TS grids with atmospheric forcings from CORE-2 dataset, which is also used in the Ocean Model Intercomparison Project (Griffies et al., 2016). Specifically, CORE-2 dataset contains a Normal-Year Forcing (NYF) with the climatological annual cycle based on NCEP atmospheric reanalysis, and it has a spatial resolution of about 2° (T62) with four-times daily wind stress fields. NYF dataset is mainly based on year 1995 of NCEP atmospheric reanalysis, with interpolation-based smoothing at the end of December with data from year 1994, and flux corrections to ensure overall energy balance. Following the common practice in CESM, for the coupling of atmospheric state variables (such as air temperature, humidity, etc.), a bilinear interpolator is used between T62 and each TS grid. For wind stress, the patch-recovery algorithm is adopted to ensure good structure of wind fields on the ocean-sea ice grid. Patch-recovery is a high-order interpolation method based on local reconstruction of the forcing fields, and it ensures consistent wind stress forcing across the 3 grid resolutions in this study. For fluxes, a first-order conservative interpolator is utilized. All these interpolators (in total 6) are generated through CESM mapping toolkit and ESMF regridding toolkit (https://www.earthsystemcog.org/projects/esmf/).

**Table 2.** Time stepping and EVP configurations.

| Grid | $\Delta_t$ for thermodynamics | $\Delta_t$ for dynamics | EVP subcycles per dynamics step |
|------|------|------|------|
| TS045 | 60 min | 30 min | 120 / **240** / 480 / 960 |
| TS015 | 20 min | | 120 / **240** / 480 / 960 |
| TS005 | 15 min | 7.5 min | 120 / **240** / 960 |

All the model integration for TS grids in CESM, including grid files in the format of POP, and interpolation files, are openly available (details in *Code and data availability*).

## 3   Experiments and Analysis

### 3.1   Spin-up experiments and Arctic sea ice climatology

The spin-up simulations with the new grids are based on CESM D-type experiments, and configured specifically for each grid. CESM D-type experiments are based on CORE-2 NYF dataset and coupling to SOM, and it is usually used for the spin-up for the sea ice and ocean-sea ice coupled system. For CICE, the time stepping are based on 240 EVP subcycles (i.e., the number of timesteps for elastic wave damping, or NDTE) per dynamics time step for all three grids, following the settings in Table 2. The experiments are outlined in Figure 4.a. Specifically, the experiment with TS045 starts on Jan. 1st with no sea ice, and

the model is gradually reaching an equilibrium state for both sea ice coverage and volume. After 25-year's experiment with TS045, the spun-up status is migrated onto TS015. Similarly, after another 5-year's experiment with TS015, the spun-up status is further migrated onto TS005. Specifically, we carry out the analysis of Arctic sea ice climatology based on experiments with the default value for NDTE (240) for TS045, TS015 and TS005. The experiments with other values for NDTE show little differences in overall sea ice coverage and volume, but does greatly impact the modeling of kinematics (details in Sec. 3.2).

As shown in Figure 4.b, for TS045, the Arctic sea ice extent (SIE) and volume (SIV) approach equilibrium after about 5 and 30 years, respectively. Besides, with migrated status from experiments with lower-resolution grids, the runs with TS015 and TS005 attain quasi-equilibrium status towards the end of the 37th model year. The annual cycles of the sea ice coverage, computed as the mean monthly SIE and SIV of the last 5-year's model output, agree well among the three grids (Fig. 4.b and Fig. 5.a). The differences in SIE and SIV are within 5%. The spatial distributions of sea ice are also consistent (Fig. 5.c through

h). The model results are also in good agreement with observational climatology of seasonal cycle for sea ice extent (NSIDC sea ice index, year 1979 to 2002), with minor overestimation of the sea ice coverage during winter (Fig. 5.a). As shown in Figure 5.f through g, the overestimation in March is present in outer regions to the Arctic Ocean, including Okhotsk Sea, Labrador Sea, southern part of Greenland Sea (near Iceland), and the southern part of Barents Sea. During summer when the sea ice coverage is mainly present within the Arctic Basin, and all three grids reproduce reasonable sea ice coverage regarding

observations (Fig. 5.c through e).

The modeled SIV peaks in April (Fig. 5.b), and the modeled seasonal cycle is consistent with existing sea ice thickness reconstructions of PIOMAS (Schweiger et al., 2011). The overall thickness pattern show thick ice (over $3m$) in the regions north of the Canadian Arctic Archipelago (CAA) and Greenland, as well as within CAA. Among the three grids, higher resolution grids simulates slightly higher sea ice volume by the end of winter (by only 4%), which is witnessed in central Arctic (Fig. 5.f through h). During summer and early months of the winter, all three grids simulate lower SIV as compared against PIOMAS (Fig. 5.b), with one particular region with underestimation of ice thickness in the Atlantic sector of the Arctic (Fig. 5.c through e). In general, we consider the model simulates reasonable Arctic sea ice climatology, especially during winter. Furthermore, with good consistency across the three resolutions, we carry out the analysis of sea ice kinematics with high-frequency outputs of these experiments.

One major obstacle of running high-resolution models is the huge amount of computational overhead and the long durations for climate simulations. In this study, we utilize an Intel processor-based cluster with 24 cores/node for all the experiments. For TS045 and TS015, more than 5 simulated year per day (SYPD) can be attained with less than 1000 cores for all the experiments. For TS005, the simulation speed is about 1 SYPD with 1920 cores and NDTE=240, which is reasonable given the high resolution of the grid. The utilization of larger computational facilities for TS015 and TS005 remains an important direction of future work.

### 3.2 Sea ice kinematics and scaling on representative days

In order to study the sea ice kinematics, we output two years (36-37) of daily mean sea ice fields for all three TS grids (Fig. 4.a). Spatial scaling analysis of these deformation rates are performed by using the methods in Marsan et al. (2004). The deformation rates in their invariant forms are defined in Equation 1 through Equation 3, including: shearing rate ($\dot{\varepsilon}_{shear}$), divergence rate ($\dot{\varepsilon}_{div}$) and total deformation rate ($\dot{\varepsilon}_{total}$). They are computed from the daily mean prognostic sea ice drift speeds ($u$'s and $v$'s) defined on Eulerian grid locations (with Arakawa-B staggering in CICE). Specifically, the line integral of a specified model region is computed to for the spatial derivatives and the associated spatial scale (Fig. C1). The deformation rates are then computed and binned according to the specific spatial scale, in order to derive statistics including the probability density of deformation rates at different scales. Appendix C includes a detailed routine for the Arakawa-B staggered grid of CICE.

$$\dot{\varepsilon}_{div} = \frac{\partial u}{\partial x} + \frac{\partial v}{\partial y} \tag{1}$$

$$\dot{\varepsilon}_{shear} = \sqrt{\left(\frac{\partial u}{\partial x} - \frac{\partial v}{\partial y}\right)^2 + \left(\frac{\partial u}{\partial y} + \frac{\partial v}{\partial x}\right)^2} \tag{2}$$

$$\dot{\varepsilon}_{total} = \sqrt{\dot{\varepsilon}_{div}^2 + \dot{\varepsilon}_{shear}^2} \tag{3}$$

We have manually chosen two typical days that are representative of Arctic sea ice drift patterns: Dec. 20th and Feb. 6th (Fig. 6.a and b). A winter-time Arctic Oscillation (AO) index is constructed based on sea-level pressure (SLP) of 50-yr's NCEP reanalysis data, and applied to the NYF dataset. Specifically, AO is defined as the leading EOF mode for the weekly mean SLP in the northern hemisphere ($20\,^\circ N$ and north) for the extended winter months (November to April) of years 1950 to 2000. The

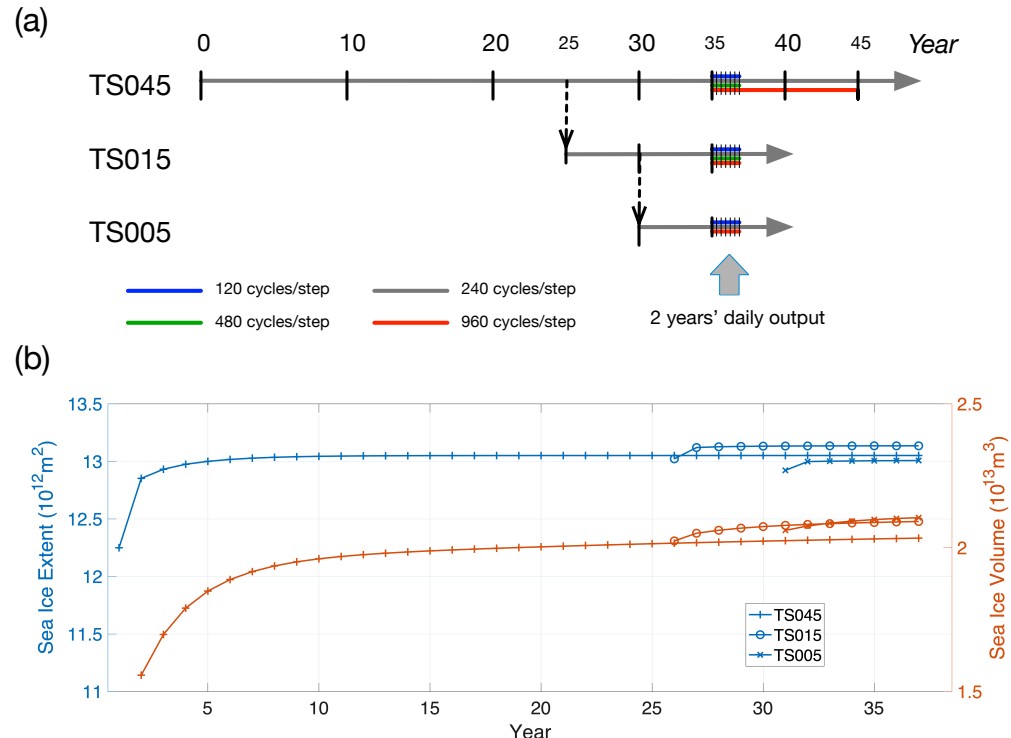

**Figure 4.** Spin-up experiments of sea ice simulations with TS grids (a) and spin-up of Arctic sea ice extent and volume (SIE and SIV, in panel b) up to year 37. These experiments are based on D-cases in CESM: the sea ice component CICE is forced with Normal Year Forcing of CORE and coupled to the Slab Ocean Model. For TS045, the model is initialized with an ice-free condition, and the numerical integration is carried out until equilibrium. For TS015, a snapshot of the spun-up conditions of TS045 at the end of year 25 is used to initialize the integration, and the experiment is carried out for 12 years, up to year 37. For TS005, a snapshot of the spun-up status of TS015 at the end of year 30 is used for model initialization, and the experiment is carried out for 7 years, up to year 37.

50-year SLP sequence is detrended and the seasonal cycle is removed for the EOF computations. The leading mode explains 13.9 % of the total variance, with the normalized spatial pattern (unitless) and the principle component (time series) shown in Figure S1. The wintertime AO index of the NYF dataset is shown in Figure 6.c. The winter of the NYF forcing dataset corresponds to an overall neutral AO status, and the variability of wintertime weekly AO index (25 hPa as in Fig. 6.c) is also on par with the average intra-seasonal variability of the 50-year NCEP reanalysis data (22 hPa as in Fig. S1.b). As a reference, the summertime AO of the NYF dataset is mildly negative (Fig. S1.c).

The two representative days are: (1) Dec. 20th, on which date the high-pressure center resides in Beaufort Sea, and a negative AO index is witnessed, and (2) Feb. 6th, on which date the high-pressure center shifts towards the eastern hemisphere, the low-pressure system in the Atlantic sector extends further into the Arctic, and a slightly positive AO index is present. Furthermore, because asymptotic convergence of sea ice kinematic to increase in the EVP subcycle count is witnessed in existing studies

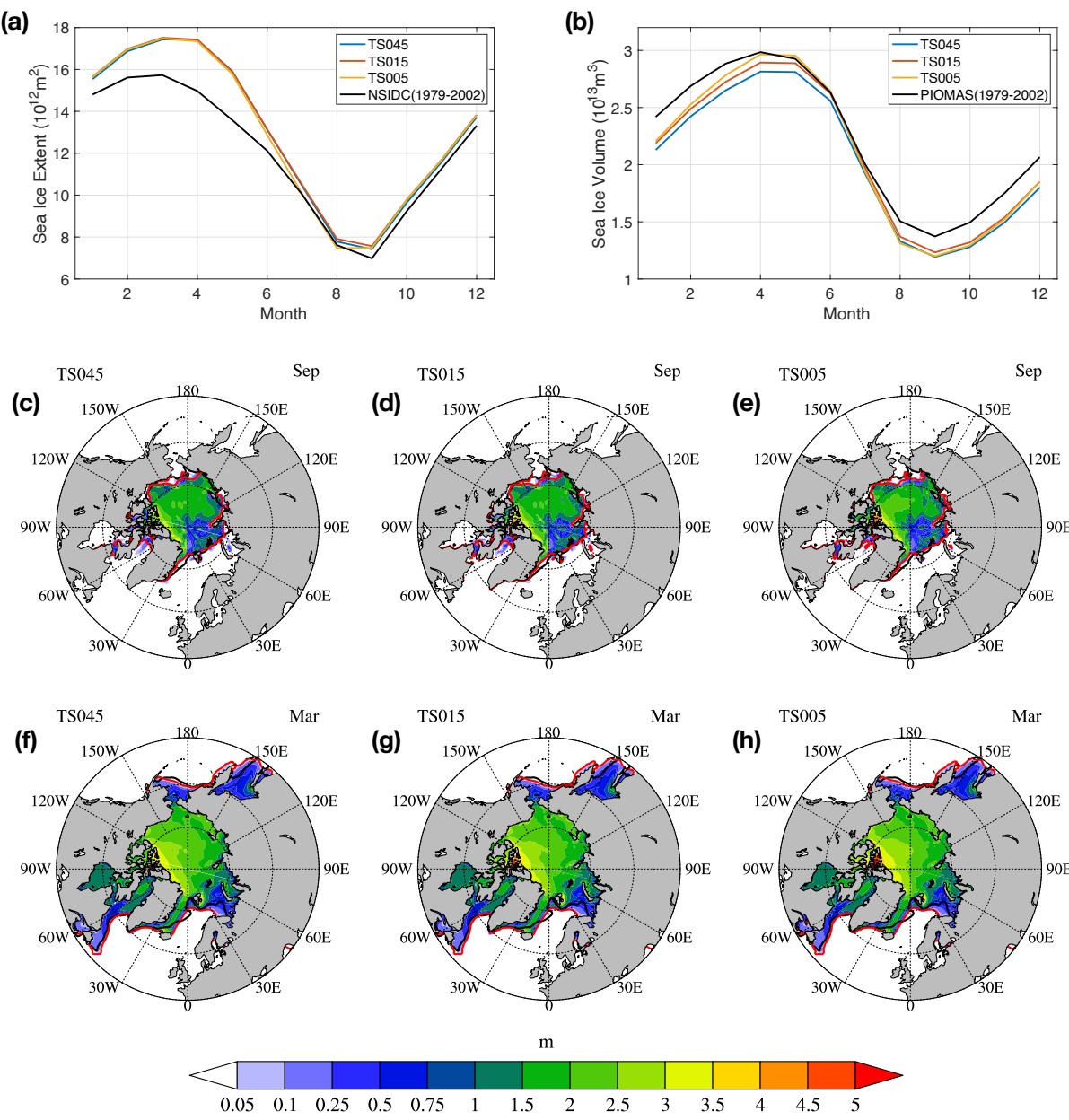

**Figure 5.** Climatological seasonal cycle of Arctic SIE (panel a) and SIV (panel b), computed as 5-year mean of year 33 to 37. September (or March) sea ice thickness fields for TS045, TS015 and TS005 are shown in panel c (or f), d (or g) and e (or h), respectively. Within each panel (c through h), the satellite-observed and modeled sea ice edge (sea ice concentration at 15%) are marked by black and red lines, respectively. The climatology SIE, as well as as March and September sea ice edge, are computed from the NSIDC Arctic sea ice dataset from passive microwave sensors (SSMI/SMMR), for the years between 1979 and 2002. The climatological annual cycle for SIV (in panel b) is computed from PIOMAS dataset for the same period (1979 to 2002).

(Lemieux et al., 2012; Koldunov et al., 2019), we limit the analysis of kinematics and scaling to the experiments with the largest EVP subcycles (NDTE=960) for each grid.

Figure 7 shows the deformation fields for Dec. 20th. On the Pacific side, there is divergence in the Beaufort Sea and Chuckchi Sea, and accompanying convergence in East Siberian Sea. Shearing rate of up to 10 $\%/d$ is also present in these regions, with a shearing arc extending from Beaufort Sea across the basin to Severnaya Zemlya and another one along the Siberian Shelf. In the Atlantic sector of the Arctic Ocean, there is extensive divergence to the north of Svalbard, and minor convergence to the north of Greenland. In the regions of thinner ice and marginal ice zones, large deformation rates are present.

The overall deformation pattern on Dec. 20th is consistent across TS045, TS015 and TS005. Comparing TS045 and TS015, we witness much finer structure of the deformation fields in all aforementioned regions. Specifically, both narrower shearing and divergence regions and higher rate of deformation rates are present in TS015. The differences of the kinematics between TS015 to TS005 are mainly present in fine structure of the deformation and shearing events. With TS005, the model simulates more and finer sea ice kinematic features, such as the network of shearing in Beaufort Sea and along the Siberian Shelf.

Figure 8 shows the deformation fields for Feb. 6th. On the Pacific side, the divergence (convergence) region is in Beaufort Sea (Chuckchi Sea). The divergence in Laptev Sea is paired with the convergence to the north of Queen Elizabeth islands (see Fig. 6.b for the sea ice drift). A shearing belt is present across the basin from the Canadian Arctic Archipelago to the north of Franz Josef Land. In the Atlantic sector, the convergence to the north of Barents Sea corresponds to the divergence in the north, including the north of Greenland and the north of Franz Josef Land. The simulation results are consistent across the three grid resolutions, including the major regions of divergence (or convergence) and those of shearing, except for very large shearing belts across the basin.

Similar to Dec. 20th, TS045, TS015 and TS005 simulates consistent deformation fields, including the location and strength of deformation events in the Arctic basin. There is a clear separation of the level of detail for the deformation systems across the resolution. The kinematic features which can be detected by visual inspections, are better defined in TS005. Especially the networks of deformations, such as those in the Davis Strait and to the north of Fram Strait and Greenland, contain over 20 major linear features in TS005. For comparison, the run with TS015 produces much fewer features, while that with TS045 only produces the major 1 or 2 deformation features. However, on both days, there is also evidence that, even at TS005, the model is limited for resolving fine-scale kinematic structures. Some large kinematic features, especially in the central Arctic, have ends that are not well defined, which is clearly not bounded by the grid sizes. This applies to both shearing and divergence. On the grid's native resolution, the simulated kinematic features contain both physical deformation and the deformation caused by numerical issues, such as limitation of grid resolution and non-convergent solutions. As a consequence, the effective resolution of the model is usually coarser than the grid's native resolution, which we further evaluate using the statistics of sea ice deformation.

We examine the the probability density function of total deformation rates, which were previously investigated for RGPS data (Marsan et al., 2004). Specifically, the region outlined in Figure 7 is used for study, and we show the daily and 3-day cumulative probability distribution function (PDF) for total deformation rates in Figure 9. The 3-day deformation data is computed from the 3-day mean velocity fields for the period of Dec. 19th to Dec. 21th, and Feb. 5th to Feb. 7th, respectively. With

higher resolution, the model consistently simulates more extreme deformation events, corresponding to a flatter cumulative PDF for the deformation rates.

Since the three grids have nearly exact grid stepping ratio of 1:3:9, we carry out: 1) the spatial coarsening of the model output of TS005 onto TS015 and TS045, and (2) that of TS015 onto TS045. The cumulative PDFs for each resolution, including modeled with the native resolution as well as scaled from higher resolutions, are shown in each panel in Figure 9 with the same symbol. At larger spatial scales (7.3 km and 22 km), the slopes of the PDFs become steeper for both TS005 and TS015 on Dec. 20th, but witness much smaller changes on Feb. 6th. The slopes of cumulative PDFs from scaled rates of TS005 are shallower than non-scaled rates of TS015, indicating that the effective resolution of the TS015 run is lower than its native resolution of 7.3 km. Similarly, at the spatial scale of TS045's native resolution (22 km), the slopes from TS005 and TS015 are shallower than that from TS045. This indicates that at the scale of 22 km, more extreme deformation events are present in the run with TS005 and TS015 than in TS045. Therefore, in terms of modeling realistic shape of the PDF of sea ice deformation rates, the model's effective resolution is coarser than the grid's native resolution.

By treating the cumulative PDFs of the scaled results of TS005 as the "truth", we evaluate the effective resolution of grids of coarser resolutions. Specifically, for TS015, we define the effective resolution scale as the scale at which the PDF's tail slope of the scaled results from TS015 reaches that from TS005. We further scale down the model results of both TS005 and TS015 to attain the slopes at spatial scales coarser than 22 km (or 0.45°). We attain same slope of PDF tail for TS015 as TS005 at the scale of 42 km for Dec. 20th. For Feb. 6th, the scale with same PDF tail slope for TS015 and TS005 is 50 km. This result gives certain hint of the effective resolution of TS015 (with grid resolution of about 7.3 km). Regarding the cumulative PDFs of sea ice kinematics, the effective resolution is variable among days with different kinematic features, and about 6 to 7 times the native resolution of TS015. Also, since we witness flatter cumulative PDFs for the scaled results of TS005 than TS015, we expect that at the spatial scale of 2.4 km (TS005's native resolution), the physical cumulative PDFs are flatter than the model output of TS005, which have the slope of -1.0 for Dec. 20th, and -0.5 for Feb. 6th.

For cumulative PDF of 3-day mean sea ice deformation (lower panels of Fig. 9), we witness slight steepening of the PDF tails for both days than those of daily deformations. This is due to the temporal averaging on Eulerian grid points which attenuates the large deformations, causing fewer large deformation events in the PDF. On Dec. 20th, the PDF tail slope is -2.7 at the scale of 22 km (from the scaled results of TS005). For reference, the analysis with RGPS data in Marsan et al. (2004) in which the PDF tail slope of -2.5 is found for 13-20 km scale (Fig. 3 of the reference). However, it is worth noting that a Lagrangian-tracking based method of our model output is needed for formally compare the results (Hutter et al., 2018), which we plan to carry out with historical simulations and the new grids.

We further carry out spatial scaling analysis for these two representative days (Fig. 10). Specifically, the moments ($q$) that are adopted to evaluate the structure function of scaling and the multi-fractality are: 0.5, 1, 2 and 3. The polynomial for the least-square fitting of the structure function is: $\beta(q) = a \cdot q^2 + b \cdot q$. At $q = 1$, the three resolutions show slight differences for $\beta$: for both days, and higher resolution starts with a slightly higher mean deformation rate (0.12 and 0.09 for TS005 on Dec. 20th and Feb. 6th, respectively). At $q = 3$, $\beta(q)$ is in the range of 0.7 (TS045) and 1.1 (TS005) on Dec. 20th, and between 1.1 and 1.6 on Feb. 6th for all three grids. At higher orders (e.g., $q = 2$ or $q = 3$), high-resolution produces faster decay of deformation

rates with spatial scales (larger $\beta$) on Dec. 20th, but on Feb. 6th, the differences of decay rates are less pronounced between the grids. For 3-day mean deformation fields, the mean deformation rate is generally lower than at 1-day scale, and the difference on higher orders is also evident between Dec. 20th and Feb. 6th. Furthermore, no negative value of $\beta$ is detected at $q = 0.5$, which is consistent with Marsan et al. (2004) (Fig. 4 of the reference).

The structure function of $\beta(q)$ shows a strong curvature, and this applies to all the grids on both days, indicating multi-fractal spatial scaling of the sea ice kinematics. On Dec. 20th, the curvature level ($a$) is higher in TS005 (0.12) than TS015 (0.09) and TS045 (0.05). On Feb. 6th, the curvature levels are more consistent across the 3 resolutions (0.16 for TS045, 0.17 for TS015, and 0.23 for TS005). The differences in the statistics of scaling is the result of both the localization of the specific deformation field, as well as the resolution of the grid. Besides, there is slight drop in $\beta$ when evaluating 3-day deformation fields with 1-day

counterparts. This is consistent with existing works in which formal temporal scaling is carried out, which indicate decrease in $\beta$ with longer time scales. However, our analysis here is based on model outputs on Eulerian grids, and a formal temporal scaling based on Lagrangian diagnostics of our model data is planned in future study.

Based on the numerical experiments with NYF dataset, we show that the sea ice kinematics, including the deformation and its spatial scaling, are distinctive on different days. On the two representative days, the deformation fields show multi-fractality,

which is consistent with existing studies with observational datasets and modeling results (Marsan et al., 2004; Rampal et al., 2019). On Feb. 6th, there are more larger deformation events than Dec. 20th: higher 95-th percentile is present for all 3 resolutions, including scaled results (Fig. 9). Large-scale sea ice drift patterns were found to be greatly accurately determined by geostrophic winds and the associated SLP field and AO indices (Rigor et al., 2002). The associated sea ice deformations at smaller spatial and temporal scales, due to the multi-fractality and large scale-small scale linkage, is highly dependent on the

atmospheric forcings which contain inherent variability at different scales. Furthermore, the sea ice deformation is also greatly dependent on sea ice status, such as ice thickness and strength. The experiment in this study aim at obtaining a converged sea ice state that reflect reasonable Arctic climatology, therefore the ice thickness and multi-year ice coverage is higher than the ice condition when existing satellite observations such as RGPS were carried out. Historical simulations which are driven by high-resolution, inter-annually changing atmospheric forcings, as well as coupling to dynamical atmospheric and oceanic

models, are needed to compare with coinciding observations such as RGPS.

### 3.3   Wintertime kinematics

We extend the analysis of spatial scaling to the winter months (Dec. to Feb.). Daily deformation fields are used to construct the time sequence of the spatial scaling exponent $\beta$ for $q = 1$ during this period (Fig. 11). The value of $\beta$ shows very large day-to-day variability throughout the winter, but mostly within -0.1. During winter, the analyzed region (Fig. 7) mainly consists

of packed ice, with the ice concentration close to 100% and the mean ice thickness over 2 m (Fig. 5.h). Similarly, as reported by Hutter et al. (2018) which is a model-based study, both packedness and thickness contribute to an exponent close to 0, and the central Arctic shows very low scaling factor ($0 < \beta < 0.09$) in January. This is consistent with our result which also show $\beta$ within 0.1. We do notice that in Hutter et al. (2018), hourly sea ice deformation fields are used to compute the scaling coefficients, compared to daily fields in our current study. However, the spatial-temporal scaling in Hutter et al. (2018) show

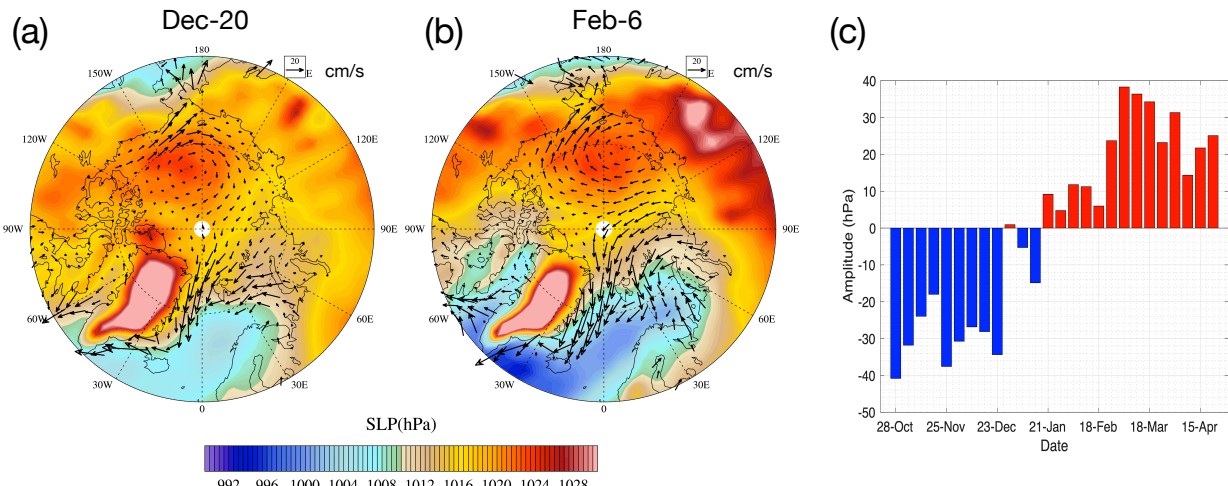

**Figure 6.** Daily sea-level pressure from NYF dataset (filled contour) and modeled sea ice motion by TS045 (vectors) on Dec. 20th (a) and Feb. 6th (b). Wintertime Arctic Oscillation (AO) index of the NYF dataset is in panel c. See the text for detailed methods for the computing of AO.

very close values for $\beta$ between hourly and daily results (Fig. 7 of the reference). We consider the low scaling coefficient as produced by our model runs reasonable for characterizing sea ice kinematics in packed ice.

Furthermore, as revealed in Figure 11, distinctive phases are present during the 3 months. During January, the exponent is between 0.02 and 0.03, and lower (closer to 0) than the previous month of December (mainly between 0.04 and 0.06). The value of $\beta$ also grows (more negative) towards February (around 0.04). There are several potential contributing factors to the differences in $\beta$. First, in order to ensure continuity at the beginning/end of the year, for December, NYF dataset is based on the interpolation of year 1994 and 1995 of NCEP reanalysis. Potentially, the atmospheric processes during this month may contain attenuated spatial and temporal variability. Second, in NYF dataset, the dominant atmospheric variability (AO) shifts from negative in December to positive in January and February. This corresponds to the systematic shift of $\beta$ during this period. However, a simple regression of AO on various scales (daily to weekly) yields no significant statistical correlation with $\beta$. Since the atmospheric forcing dominates large-scale sea ice drift, we conjecture that regarding atmospheric forcing, the fine-scale atmospheric processes (such as spatial and temporal wind variability) serve as the missing link between large-scale drift and small-scale kinematics and statistics. Lastly, sea ice status, including sea ice strength and rheology, dominates the shearing and convergence/divergence failures at local scale. The thickening of the sea ice throughout the winter months may also contribute to the increase of ice strength and the overall decrease in $\beta$.

### 3.4 Numerical convergence of EVP

In this section we evaluate the sensitivity of the modeled sea ice kinematics to the EVP subcycling and asymptotic convergence. The elastic wave term introduced the EVP is more effectively damped with more subcycles, leading to consistent deformation

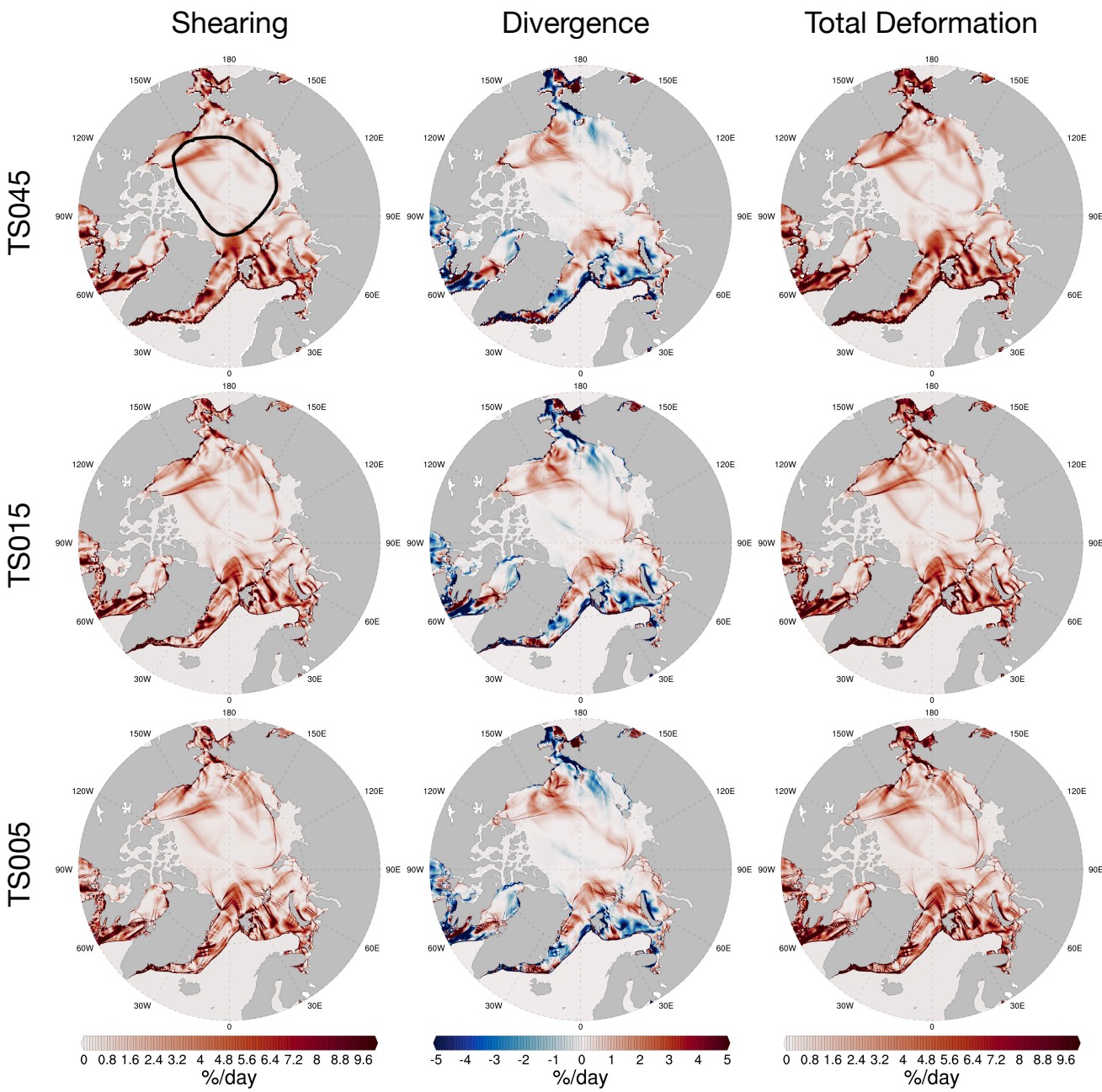

**Figure 7.** Daily deformation fields on Dec. 20th. The first, second and third row show the deformation fields for TS045, TS015 and TS005, respectively. Deformation rates include shearing rate (left), divergence rate (middle) and total deformation rate (right). The region for further statistical analysis including spatial scaling is outlined in black in the first panel. All results are based on experiments with NDTE=960.

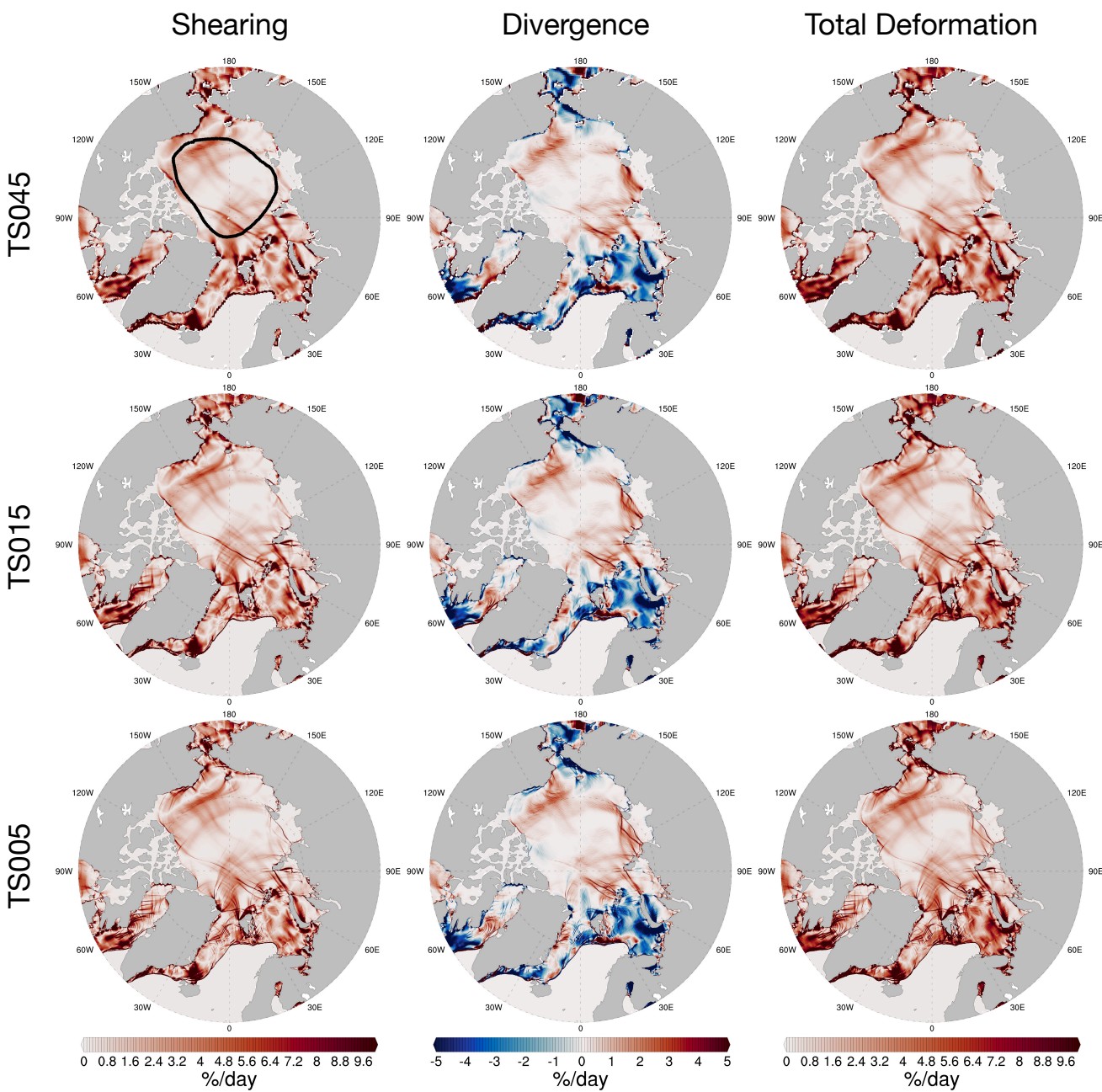

**Figure 8.** Same as in Fig. 7, but for Feb. 6th.

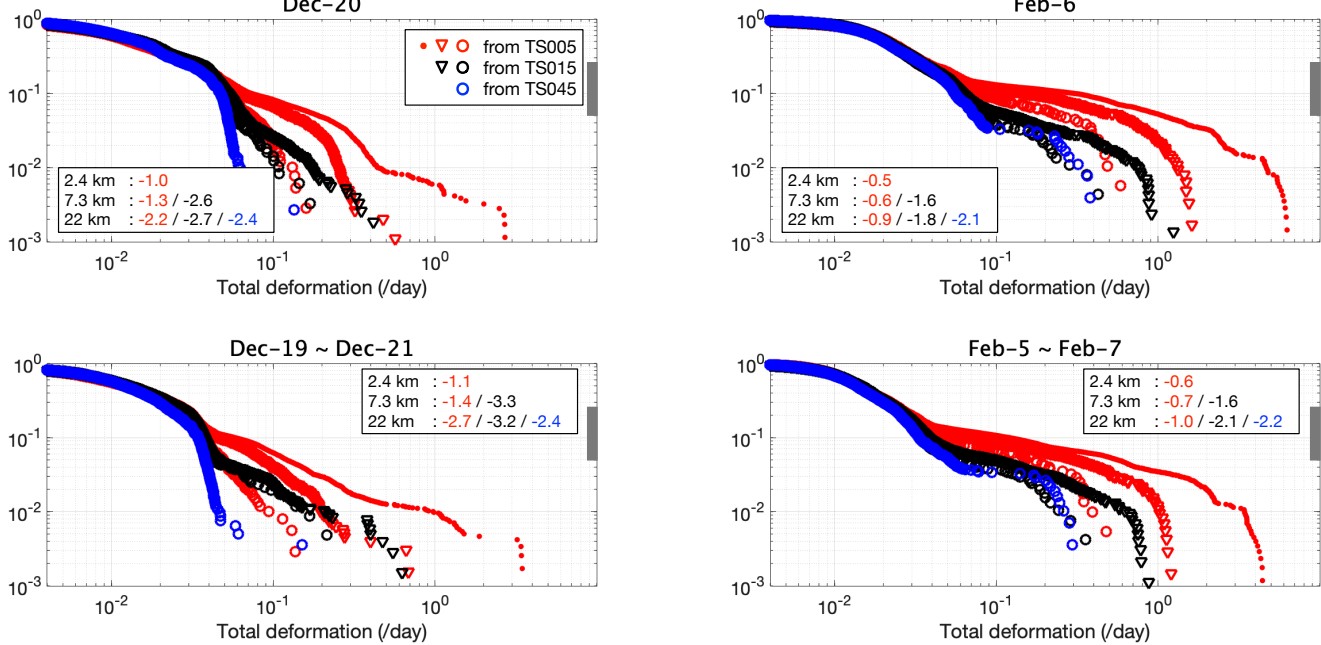

**Figure 9.** Cumulative density functions for the total deformation rates on Dec. 20th and Feb. 6th. Region of study is outlined in Fig. 7 and Fig. 8. Daily rates are shown in the first row and 3-day rates on the second row. Colors indicate the cumulative PDFs from the model results from different grids (red: TS005; black: TS015; blue: TS045). With spatial coarsening of TS005, we compute the cumulative PDF at the spatial scales corresponding to native resolution of TS015 and TS045. In turn, the spatial coarsening with TS015 is also applied to compute the cumulative PDFs at the equivalent resolution of TS045. The 3 spatial scales are marked by different shapes: 22 km (or TS045's native resolution) in circles; 7.3 km (or TS015's native resolution) in triangles; 2.4 km (or TS005's native resolution) in dots. Slopes are computed for the range of the cumulated probability between 0.05 and 0.25 for all cumulative PDFs (marked out in grey in each panel), which correspond to 95th and 75th percentile of the deformation fields, respectively. All results are based on experiments with NDTE=960.

fields. This asymptotic convergence of the deformation fields to EVP subcycling is examined in this study. In Figure 12 we show the probability density function (or PDF) of daily deformation rates during wintertime (Dec. to Feb.) for the three grids. All the simulations attain a good shape for the tail of of the PDF, approaching the slope of -3. For the total deformation rate, there exists a well defined mode at $0.1\%/d$ to $0.2\%/d$ for runs with NDTE=960, and at NDTE=120, a slight shift of mode to higher values (between $0.5\%/d$ to $1\%/d$ for TS015 and TS005). At different resolutions, the EVP subcycling count plays a different role in the shape of PDF. For TS045, the PDF is consistent between different subcycle counts. But for TS015 and TS005, there are much more evident differences: (1) with larger NDTE, there are more regions with smaller shearing and divergence (less than 0.1 %/d); (2) there is general convergent behavior of the shape of the PDF when NDTE is large (e.g., NDTE>480 for TS015). This behavior also applies for specific days (see Fig. S2 and Fig. S3 for the PDF on Dec. 20th and Feb. 6th, respectively). Besides, the PDFs of high-resolution runs (TS015 and TS005) show better tail structure, and furthermore, the slopes are also better characterized (i.e., closer linear fittings at -3) with larger values of NDTE. For TS045, the tail of the

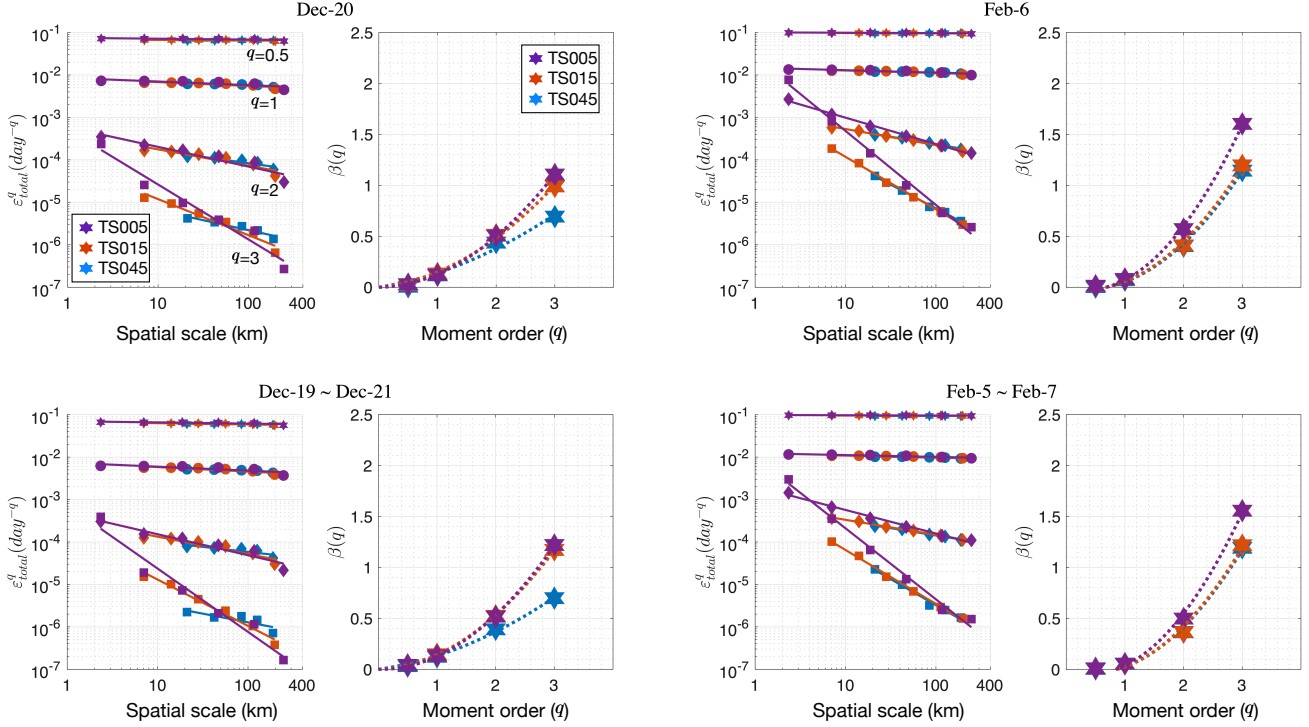

**Figure 10.** Spatial scaling of total deformation rate for daily deformation fields (top row) and 3-day mean deformation fields (lower row). The daily mean velocity fields on Dec. 20th and Feb. 6th, as well as the 3-day mean velocity fields centering on Dec. 20th and Feb. 6th are used to derive the scaling curves. Each panel contains the scaling curves and the structure function relating the scaling coefficients ($\beta$'s) to the moment of order ($q$). Similar to Fig. 7, 8 and 9, the results are based on experiments with NDTE=960. Detailed methodology used to compute the deformation rates with the model outputs is outlined in Appendix B.

PDF suffers from the lack of samples for large deformation events. This indicates that insufficient EVP subcycle count leads to overall non-convergent deformation rate distributions.

A visual inspection of the deformation fields reveals the loss of the kinematic features when the convergence is not attained. Figure 13 shows the daily total deformation fields on Dec. 20th as simulated with different NDTE values for each grid. The more remarkable difference is between the runs with NDTE=120 and NDTE=960 with the highest resolution (i.e., TS005). Although the linear kinematic features are well defined with NDTE=960, the deformation field is much noisier for the run with NDTE=120, with only the larger features detectable. The noise level is at about 1 %/$d$, which corresponds to the mode of PDF in Figure 12. With larger values of NDTE, the noise level decreases and the deformation rate around the linear kinematic features becomes smaller. As a result, a convergent PDF and linear feature maps are attained. One exemplary region is the Canadian Arctic Archipelago (CAA), where the landfast sea ice dominates during winter in the model run with NDTE=960. For example, Figure S4 shows that the 2-week mean sea ice velocity within the CAA is lower than $5 \times 10^{-4} m/s$ (Lemieux et al., 2015). The detailed total deformation rate field within the CAA for TS005 (third row of Figure 13) is further shown in Figure

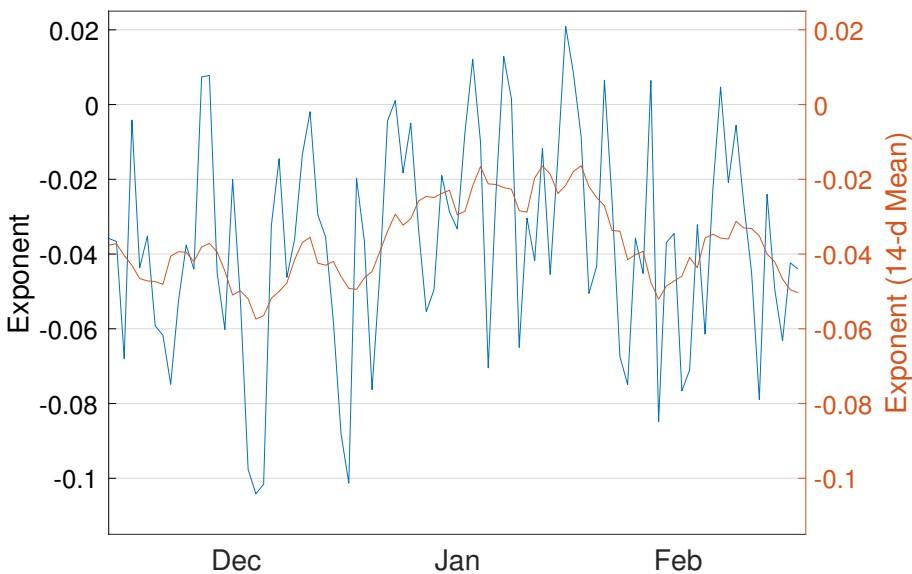

**Figure 11.** Spatial scaling exponents of the total deformation for $q$=1 for daily fields during winter. The region analyzed is outlined in the first panel of Fig. 7. Model output with TS005 grid and NDTE=960 is used for the analysis. Exponent sequence from daily fields (blue line, left y-axis) is accompanied by its 14-day running mean (orange line, right y-axis).

S5. The experiment with NDTE=120 show much higher and noisier sea ice deformation fields than that with NDTE=960. Overall, the results show that with the traditional EVP implementation, with higher resolution, even more subcycles are needed to reach convergence for the simulated kinematics. This causes further increase in simulation cost, given that the dynamics time step is already decreased in high-resolution runs, due to CFL conditions in processes such as advection (Tab. 2).

5    For TS045, although the PDF of the deformation rates are similar among runs with different NDTE values, the deformation field is also slightly more noisy in runs with NDTE=120 and NDTE=240 than that with NDTE=960. The region with the biggest difference is within the Canadian Arctic Archipelago. We further evaluate the effect of subcycling by carrying out the experiments with NDTE=240 and NDTE=960 for TS045 further to year 45 (Fig. 4). We show the difference of March (September) sea ice thickness between NDTE=960 run and NDTE=240 run on year 45 in Figure 14.a (14.b). Although there

10  is only less than 1% difference in the basin-scale ice volume between the two experiments, the sea ice is remarkably thinner in the CAA in the run with NDTE=960.

The difference in ice thickness is all-year around (i.e., in both September and March). Specifically, in the run of NDTE=960, an ice arch forms by the eastern end of the Lancaster Sound and near Amund Ringnes Island, resulting in thicker ice in these regions. In other parts of the CAA, the ice is thinner by about 15 to 20 cm on average, and up to 1 meters thinner in certain areas.

15  As shown in Figure 14.c, the run with NDTE=240, which started from year 1, has already attained equilibrium in sea ice thick-

ness and volume in the CAA well before year 35. However, in the run with NDTE=960 which starts from year 35, the model status gradually shifts to another equilibrium status for ice thickness towards year 45. The overall volume difference is uniformly 100 $km^3$ in March and September, which consists of about 6.5% of the total volume in September. We further attribute the difference of winter ice thickness to the thermodynamic (Fig. 14.d), advection (Fig. 14.e), and dynamical ridging/rafting (Fig. 14.f) contributions during the freeze-up seasons, computed as multi-year mean fields for December, January and February (DJF) of year 41 to 45. Specifically, in CICE model, the 3-month mean ice volume tendencies due to thermodynamic growth, ice advection, and ridging are computed from the model diagnostics for ice volume budget at Eulerian grid points. As compared with the run with NDTE=960, there exist small deformation events in the CAA in the run with NDTE=240, which are arguably due to non-convergent EVP solutions (see also Fig. 13 for comparisons of deformation fields). Therefore, there is more ice thickness increase in the run with NDTE=240 due to these events (Fig. 14.f). Also, since September ice is thinner in the run with NDTE=960, it is intuitive that the winter thermodynamic ice growth should be higher for NDTE=960. On the contrary, the thermodynamic ice thickness growth is in general lower in the run with NDTE=960, except for regions with the biggest thickness decrease in September as compared with the run with NDTE=240 (Fig. 14.d and Fig. 14.b). This is mainly due to the fact that the thermodynamic ice growth is also closely tied to the kinematics processes. With more noisy sea ice movement fields in the run with NDTE=240, the sea ice formation and growth is also promoted with very small deformation events, resulting in more thermodynamic growth in the run with NDTE=240. As compared with ridging and rafting, the sea ice advection is mainly responsible for redistributing the ice mass and thus plays a minor role for the overall ice volume in the CAA in our experiments (Fig. 14.e).

The dependence of the modeled sea ice thickness in the CAA on EVP convergence in our study is purely numerical, but calls for further attention from both modelers and modeling data users. It highlights the importance of numerical convergence of EVP (or any other candidate solvers to VP) on the modeled sea ice climatology in fast ice regions. Since the experiments are idealized in this study, the effects in realistic simulations may be also subjected to grid resolution and staggering, as well as coupling and feedback processes.

## 4  Summary and discussion

In this paper we carried out sea ice simulations with a multi-resolution framework with Community Earth System Model. A grid hierarchy is constructed with the resolution range spanning climate simulations (0.45°) to sub-mesoscale modeling (0.05°). At 0.05°, the grid resolution in the Arctic region is approximately 2.45 km. The grid hierarchy is incorporated into CESM, and by using atmospherically forced experiments, we simulate and evaluate sea ice kinematics and scaling properties with a multi-resolution approach. We have found good consistency of the Arctic sea ice climatology and kinematics across the resolution range. As shown in the spatial scaling analysis on the representative days, the modeled sea ice deformation is characterized by multi-fractal scaling for all three grid resolutions. In our study, high-resolution (0.05°) runs yield the most trustworthy kinematic features, and the multi-resolution simulations provide a unique approach for evaluating sea ice kinematics in lower resolutions. Furthermore, the convergence of Elastic-Viscous-Plastic rheology model is evaluated, which

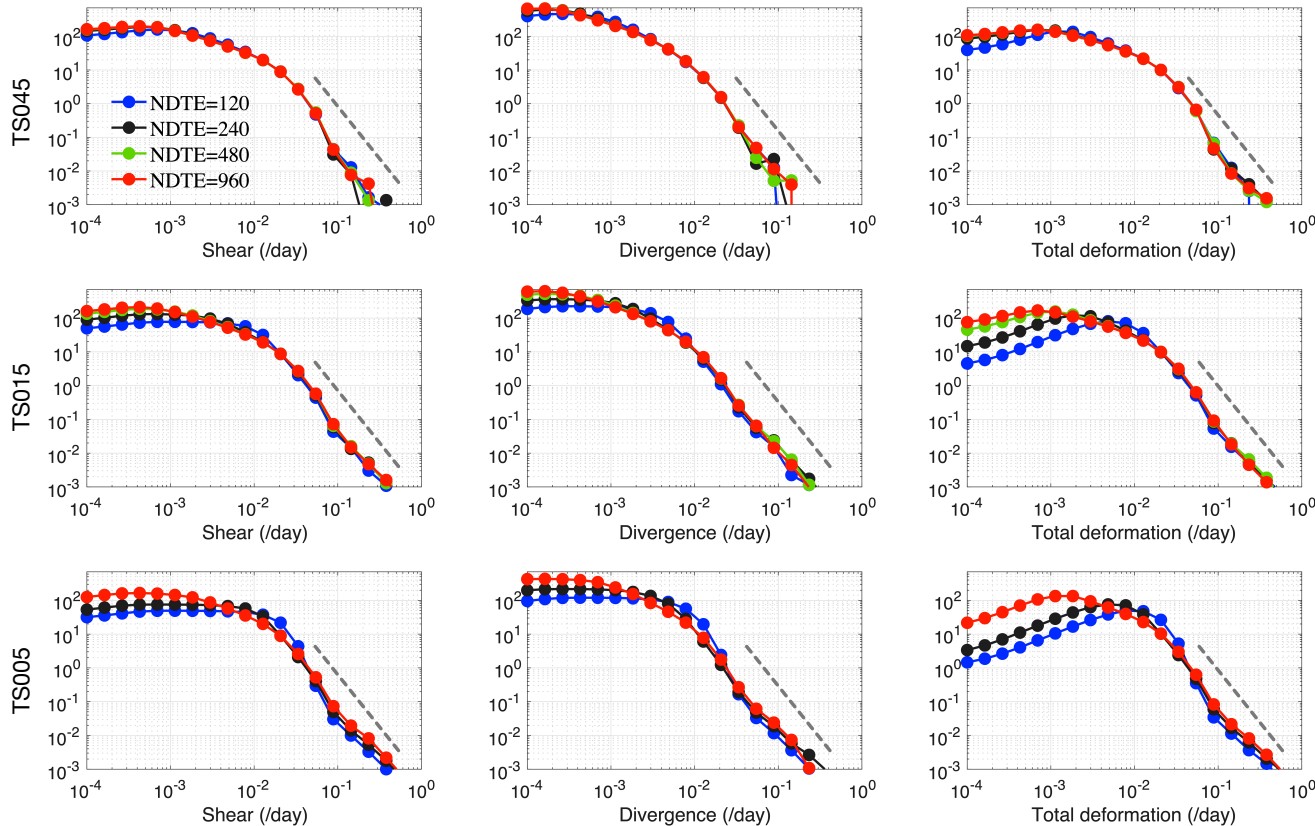

**Figure 12.** Probability density of modeled daily deformation rates during winter months (Dec. to Feb.) with TS045 (first row), TS015 (second row) and TS005 (third row). The region of study is outlined in Fig. 7. Shearing rate (left column), divergence rate (central column) and total deformation rate (right column) are shown. Runs with different EVP configurations (NDTE from 120 to 960) are marked by the same color as in Fig. 4.a. The theoretical slope for the PDF tail of -3 is shown in each panel for reference (grey dashed line).

show significant impact of EVP convergence on kinematic statistics, as well as landfast ice in Canadian Arctic Archipelago. The framework of utilizing the grid hierarchy, including TS045, TS015 and TS005, provides an infrastructure for multi-resolution simulations for both ocean and sea ice in the future. The three grids and the model integration are openly provided for public use for the version 1.2.1 of CESM.

5     Sea ice kinematics has been the focus of the high-resolution sea ice modeling community in recent years. Based on SAR remote sensing such as ASAR and RADARSAT, kilometer-level or finer observations of sea ice drift and deformation are made possible, and serve as the backbone for the validation of high-resolution sea ice simulations. Modeling studies with structured grids such as MITgcm (Spreen et al., 2017; Hutter et al., 2018) are similar to our study, although model specifics are different. The sea ice deformation and scaling can be easily derived with model output for spatial scaling, but Lagrangian

10   based diagnostics are required for the full analysis of temporal scaling. In our study, we mainly utilized model output for the study of spatial scaling properties in Section 3.2, with initial study of 3-day mean drift fields. Specifically, we have observed

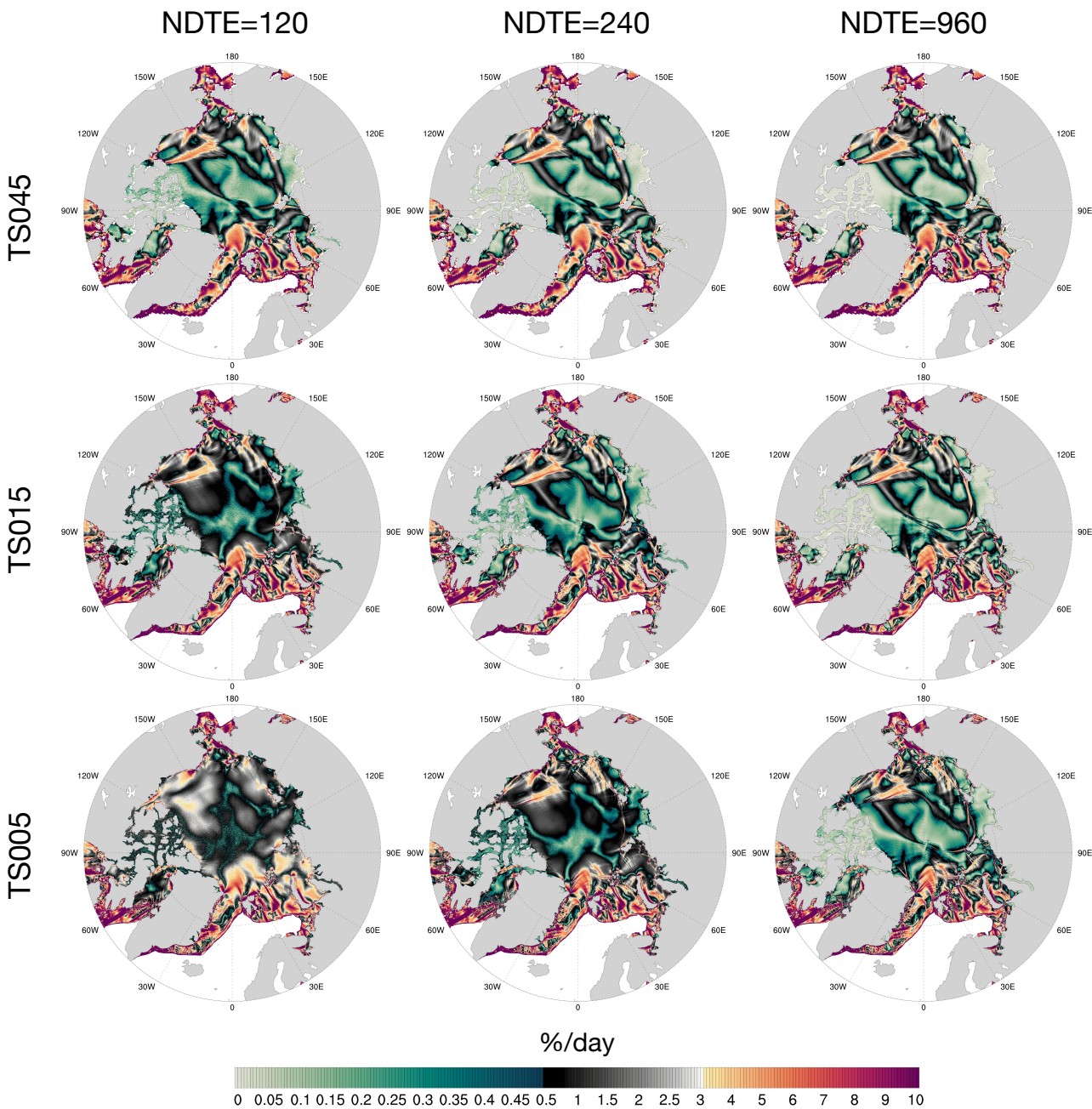

**Figure 13.** Total deformation rate on Dec. 20th with NDTE=120 (left), 240 (central) and 960 (right). Each row represents a specific grid, including: TS045 (top), TS015 (middle) and TS005 (bottom). Results with NDTE=960 are reproduced from Fig. 7. Colormap is adjusted from Fig. 7 for increased resolution of under 1 $\%/d$.

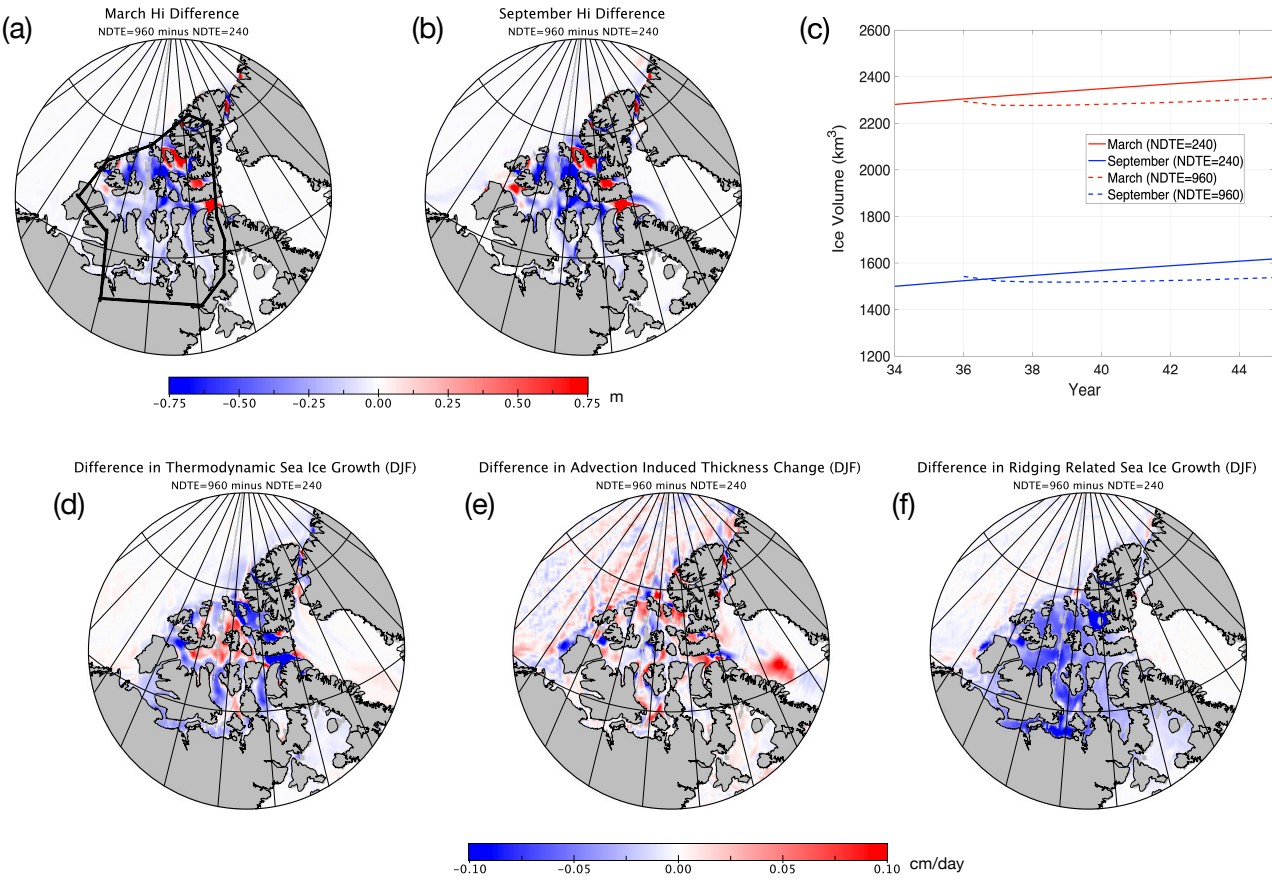

**Figure 14.** March (a) and September (b) sea ice thickness difference between the run with NDTE=240 and that with NDTE=960 of year 50 for TS045 (NDTE=960 minus NDTE=240). Sea ice volume in March and September within the CAA (outlined in panel a) since year 41 are shown in panel c. Differences in mean sea ice growth rate due to thermodynamics (d), advection (e), and dynamic ridging (f) during winter months (DJF) are computed for multiple winters (DJF of years 41 to 50). Blue (red) color indicates lower (higher) ice growth during freeze-up in the run with NDTE=960 than NDTE=240 during winter.

time-varying scaling properties, and the scaling coefficient correlates with the leading mode of atmospheric forcing (i.e., Arctic Oscillation). As future work, a full exploration of AO and ice condition dependent scaling analysis is planned with the grid hierarchy and CESM. Besides, the scaling analysis remains an important tool for evaluating sea ice kinematics, but it maybe insufficient for fully evaluating the sea ice deformation properties. Novel statistics based on linear kinematics features (LKF) are proposed in recent studies, such as Hutter and Losch (2020) and Ringeisen et al. (2019). The utilization of a full suite of diagnostics in our modeling framework, including temporal-spatial scaling and LKF based approaches, serve as an important direction for future work.

In this study, the NYF dataset of CORE-2 is utilized. This potentially compromises the comparability of the model results with satellite observations such as SAR-based sea ice kinematics (Kwok et al., 2008; Marsan et al., 2004). The inter-annual forcing dataset (IAF) of CORE-2 and the JRA-55 used by Ocean Model Intercomparison Project, phase 2 (OMIP2) can be utilized for historical simulations with the proposed grids. Furthermore, comparison can be made with specific satellite observations such as RGPS (Kwok et al., 2008). Although with large-scale, coarse atmospheric forcings such as CORE-2, sea ice models can produce multi-fractal sea ice deformation events, how these events are governed by the multi-scale atmospheric processes remains unclear. With different atmospheric forcing datasets such as CORE-2 and JRA-55, we plan to carry out comparative study by using higher versions of the coupled model (i.e., version 2 of CESM). Besides, the dynamical and thermodynamic feedback of the sea ice deformation can also be studied with an atmosphere-sea ice coupled modeling framework, with multi-resolution setting for both the atmospheric and the sea ice component model, as in CESM.

Sea ice rheology models are key to the simulation of sea ice dynamics and reproducing linear kinematic features. Together with other parameter schemes including the sea ice strength (H79 in this study) and ridging, parameters of these schemes are utilized for tuning the models towards certain observations (Bouchat and Tremblay, 2017). Specifically, sea ice strength parameter ($P^*$) and eccentricity of the elliptic yield curve of EVP are found to be tunable parameters to improve the modeling of sea ice dynamics. During the tuning of the sea ice models, the aforementioned novel statistics can also be integrated for improving rheology models such as the yield curve shape (Ringeisen et al., 2019). In Girard et al. (2009), EVP is found to be unable to reproduce observed distribution of deformation rate in RGPS dataset. Furthermore, this study confirms much better consistency with scaled deformation fields of the model output. Similarly in our study, we argue that the model's output should be studied on a coarser scale, i.e., the model's effective resolution, instead of the grid's native resolution. Another issue with modeling sea ice at very high resolution (such as TS005) is the prominent observed anisotropic characteristics. In order to explore this issue, the anisotropic rheology models such as EAP (Tsamados et al., 2013) can be utilized for a comparative study with standard EVP (or VP) with a very high resolution setting (1 to 2 km in the Arctic Basin). This is planned in our future work with the updated version of the sea ice component (version 5 of CICE) in CESM.

While EVP provides a numerically stable and easy-to-implement solver for the traditional VP model, the convergence of EVP solutions to VP model is the focus of many recent efforts (Lemieux et al., 2012; Kimmritz et al., 2015; Koldunov et al., 2019). Based on a traditional implementation of EVP in our model, we have witnessed asymptotic converging behavior of sea ice kinematics fields with increased EVP subcycle count. But this comes at a large computational overhead: at 960 subcycles per step for TS005, the simulation speed has halved compared with 240 subcycles per step, and over 50% of the computational

time is consumed in EVP, with less than 0.5 SYPD with 1920 processor cores. Ideally, an efficient and scalable implicit solver promises a more elegant and numerically sound solution to the solving problem of VP model (Lemieux et al., 2010). Also, adaptive methods that complements the convergence and efficiency problems of traditional EVP solver, such as Kimmritz et al. (2015) and Koldunov et al. (2019), are considered in our future work for the integration of the upgraded version of the coupled
or sea ice model.

    In Sec. 3.4, we have shown that the mean states for ice thickness and volume can be systematically shifted due to the numerical behavior of EVP solver. Since this issue is purely numerical, the uncertainty caused by it is different from other factors, such as the choice of ice strength parameterization scheme. In our study, the region which is most sensitive is landfast ice in the CAA, where significant decrease of ice volume is witnessed when solver convergence is attained with enough
subcycle counts. Furthermore, more subcycling is required for higher resolution to reach convergence. Given the wide adoption of VP and EVP in climate models, using the model outputs for climate research and applications (such as the projection of shipping routes) could face potential compromises if the convergence issue is overlooked. Especially, careful choices should be made for configuring EVP at different resolutions (Kiss et al., 2020). Besides, since asymptotic behavior of the sea ice kinematics to EVP subcycling is investigated in this study, the convergence of VP solver needs further formal definition for
the strict intercomparison of multiple solvers in the future (Lemieux et al., 2012). In Koldunov et al. (2019) the authors also discovered sensitivity of modeled ice thickness to EVP subcycling, but an increase in ice thickness with more EVP subcycles (Fig. 2 of the reference). Also the region with most significant thickness change is different in Koldunov et al. (2019), covering both regions in the CAA and north of CAA and Greenland. Compared to our study, the different behavior to EVP subcycling in Koldunov et al. (2019) may be due to the differences in the numerical experiments, as well as model physics, including grid
resolutions, ice thickness distribution settings, etc. Although both show relationship of ice thickness to rheology model, more analysis is needed for the attribution and explanation of aforementioned differences. Besides, tidal processes and interactions with sea ice are potentially important for the simulation of landfast ice (Lemieux et al., 2018). Since these processes are absent in our current model, we plan to include parameterization schemes that account for the their influence on sea ice kinematic in the future.

With the wide availability of high-performance computing utilities and the progress in model developments, high-resolution and even multi-resolution simulations are becoming more common for climate modeling community. While $1°$ models are still dominant in Coupled Model Intercomparison Projects (CMIP), high-resolution models are informative of potential model biases and parameterization improvements. For example, three resolutions ($1°$, $0.25°$ and $0.1°$) are built into ACCESS-OM model for ocean-sea ice coupled simulations (Kiss et al., 2020). In GFDL's most recent ocean-sea ice model [OM4.0, (Adcroft
et al., 2019)], two resolutions are adopted, including OM4p5 ($0.5°$) and OM4p25 ($0.25°$). Parameterization schemes in the ocean and the sea ice models are chosen and tuned to each specific resolution. For example, mesoscale eddy induced mixing parameterization is usually adopted for low-resolution ocean models ($0.5°$ or coarser), but inactive for higher resolutions. In our study, we use three resolutions for the study of Arctic sea ice kinematics, including: $0.45°$, $0.15°$ and $0.05°$. As is shown in the scaling analysis, this multi-resolution framework enables comparative analysis across the resolution. Given that the modeled
sea ice climatology is reasonable and consistent among the three resolutions, we consider the results adequate for the analysis

of kinematics and scaling based on the coupling to the Slab Ocean Model. Beyond the lower computational overhead of this approach, we can also attain an equilibrium status for the sea ice with fewer model years. Especially for 0.05° grid (TS005), the computational cost and the duration is prohibitively high to fully spin-up the ocean-sea ice coupled model. Furthermore, it reduces the uncertainty of ocean's modeling on the sea ice, including: (1) ocean model's parameterization schemes that are
not aligned and potentially not well-tuned between the different resolutions; and (2) the avoidance of ocean and ocean-sea ice coupled internal variability that may potentially compromise the comparability across the resolutions. For future work, long-term, inter-annually forced simulation of the coupled ocean-sea ice system is planned, under the multi-resolution framework and the spin-up strategy as adopted in this study. Specifically, the comparison with coinciding satellite observations can be carried out such as the RGPS dataset.

*Acknowledgements.* The authors would like to thank the editors and referees for their invaluable efforts in improving the manuscript. This work is partially supported by: National Key R & D Program of China (no.: 2017YFA0603902), the General Program of National Science Foundation of China (no.: 42030602), and Tsinghua University Initiative Scientific Research Program (no.: 2019Z07L01001). This work is also partially supported by Center for High-Performance Computing and System Simulation, Pilot National Laboratory for Marine Science and Technology (Qingdao). The authors would also like to thank the National Supercomputing Center at Wuxi for the computational and
technical supports during the numerical experiments.

The authors gratefully acknowledge the CESM development team at NCAR and the user community of CESM for providing the open-source usage of CESM and its component models, and their great help. We also would like to express sincere thanks for using ESMF utilities during the integration of TS grids in CESM.

*Code and data availability.* The three grids in this article (TS045, TS015 and TS005) are openly provided in the binary grid format of POP
and CICE. Modifications to CESM (version 1.2.1) are made to incorporate the new grids, with model configurations of CICE and coupling with atmospheric forcing dataset (T62). Auxiliary datasets are also provided, including: (1) coupling interpolators of TS grids to T62, and (2) CICE monthly history files during March. The grids and model integration is also available for Community Integrated Earth System Model [CIESM, (Lin et al., 2020)]. All the code and data are hosted at: https://doi.org/10.5281/zenodo.3842282.

CESM code and ESMF regridding utility are available for download at: http://cesm.ucar.edu (last accessed: 2020-Feb-20) and https:
//www.earthsystemcog.org/projects/esmf/regridding (last accessed: 2020-Mar-12).

## Appendix A: Orthogonal grid for northern patch

The construction of the orthogonal grid in the northern patch (NP, as in Sec. 2) involves a numerical process of two steps. First, a series of embedding ellipses is constructed on the stereographic projection of NP from the North Pole (Fig. A1.a). The outermost ellipse is a circle defined by the boundary between NP and SP. The innermost ellipse is a direct line that crosses
the North Pole and links the two grid poles, which are prescribed locations on land. A smooth transition from the circle to the

innermost ellipse is achieved, by controlling the semi-major axes ($a$'s) and semi-minor axes ($b$'s) of the ellipses. Without the loss of generality, we rescale the projected NP, so that the outermost ellipse is the unit circle and the innermost ellipse resides on the $x$-axis, and the layout is shown in Figure A1.a. Therefore, we have: (1) for the outermost ellipse, $a = b = 1$; and (2) for the innermost ellipse, $a = \alpha$ and $b = 0$, where $\alpha$ is half of the distance between the two grid poles. To ensure a continued

change of meridional grid scales on NP-SP boundary, the change in $b$ should be equal to that in $a$ on the outermost ellipse. In Figure A1.b we show a possible relationship between $a$ and $b$: $a = \alpha \cdot b^2 + \beta \cdot b + \alpha$, where $\beta = 1 - 2\alpha$. It can be computed that the slope of the curve equals 1 when $a$ (or $b$) approaches 1, ensuring same change speed in $a$ and $b$ and a smooth meridional scale transition on the NP-SP boundary. The ellipses, including $a$'s, $b$'s and center locations, can then be determined according to these configurations and the required resolution. In this paper, we have chosen the following parameters for TS grids: the

NP-SP boundary ($\phi$) at about $10°N$, and the two poles at $(63°N, 104°W)$ and $(59.5°N, 76°E)$, which are $180°$ apart but on different latitudes (red squares in Fig. A1.a).

    Second, with the embedded ellipses, the orthogonal grid can be constructed from the outermost circle. Since NP is directly linked to SP at latitude of $\phi$, we specify the grid points on this boundary (i.e., the outermost circle), and extend into NP to form the meridional grid lines. The construction process is iterative, starting from these points on the outermost ellipse, and for each

step constructing lines between two adjacent ellipses. In each step, the ending locations of the lines are located on the inner ellipse, under orthogonality constraints. Fig. A1.a shows a specific example between two adjacent ellipses (marked by blue and dash-blue). This whole approach is similar to the grid generation methods that are based on numerical integration processes, as in Madec and Imbard (1996) and Xu et al. (2015).

    Figure A2 show the meridional and zonal grid scales ($dx$ and $dy$) along typical meridional grid lines for TS005. As is shown,

there is smooth transition of the meridional grid scale on the boundary of SP and NP (at about J=2520 for TS005). Also the overall grid scale anisotropy is kept lower than 1.5 in the oceanic areas.

## Appendix B: Sea ice model (CICE) configuration and parameters

CICE (version 4.0) is the sea ice component of the CESM (version 1.2.1). CICE is a full thermodynamic-dynamic model for sea ice, and with the coupling framework in CESM, CICE is forced with NCEP CORE-2 atmospheric forcings and coupled

to the Slab Ocean Model. Sea ice thickness distribution with 5 categories is adopted in our experiments, which is also the default configuration for CICE in CESM. In the vertical direction, we use 4 ice layers and 1 snow layer. The sea ice dynamics mainly include four components: (1) Elastic-Viscous-Plastic rheology model with an elliptic yield curve and the aspect ratio of 2 (Hunke and Dukowicz, 1997); (2) the ice strength model in Hibler (1979), refereed as H79, in which the ice strength is related to the mean ice thickness; and (3) the ice ridging scheme as in (Lipscomb et al., 2007); (4) the advection scheme of

transport-remapping (Dukowicz and Baumgardner, 2000). Specifically, in Ungermann et al. (2017) H79 is found to produce more reasonable basin-wide sea ice thickness distribution than Rothrock (1975), and this is also confirmed in our earlier study comparing the two ice strength schemes. For thermodynamics, the Delta-Eddington (D-E) radiation scheme is adopted, with explicit meltpond formulation (Briegleb and Light, 2007). The atmospheric and oceanic stress on sea ice is parameterized as

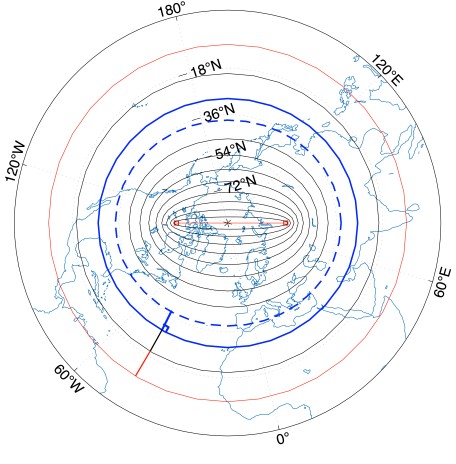

(a) Embedding ellipses

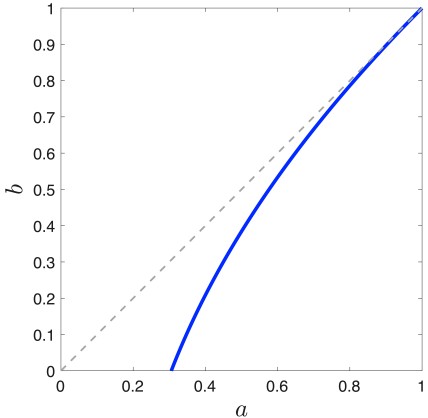

(b) Relationship between ellipses' axes

**Figure A1.** Construction of the embedding ellipses and the orthogonal grid. In panel a, we show in red the outermost ellipse (a circle) and the innermost ellipse (the direct link between the two grid poles marked by red square). The numerical process of constructing an orthogonal grid is carried out between two adjacent ellipses, starting with points the outermost ellipse, and recursively down to the innermost one. The construction of a single step on the current ellipse (blue) to the next ellipse (dashed blue) is shown in panel a. In panel b, we show the relationship between the major axis and minor axis of the ellipses under a quadratic form (see Sec. A for details).

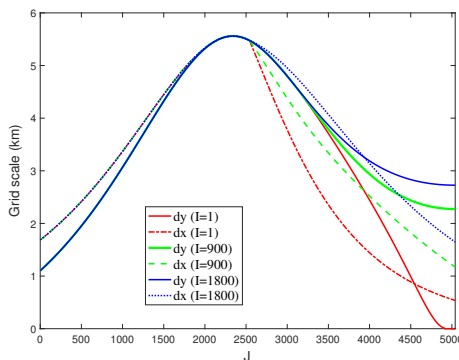

**Figure A2.** Grid sizes along the "meridional" grid lines for TS005 (I & J dimensions are 7200 and 5040). Three typical lines are chosen: I=1, I=900 (1/8 in the I-direction), and I=1800 (1/4 in the I-direction). Grid sizes in both I and J directions retain continued changes. The corresponding grid lines are also shown in Figure 2 with the same color coding. Note that when I=1, $dy$ (meridional grid size) approaches zero when J is close to the northern end (J=5400), which does not affect the model's time stepping since it is near the grid pole which resides on the land.

**Table B1.** Key model parameters of CICE in numerical experiments.

| Parameter | Value | Notes |
|---|---|---|
| $P^*$ | $2.75 \times 10^4 N/m$ | Ice strength parameter for H79 |
| $C$ | 20 | Empirical constant parameter for H79 |
| $a^*$ | 0.5 | $e$-folding scale for participation function during ridging |
| $\mu_{rdg}$ | $4\ m^{0.5}$ | $e$-folding scale for ice ridging |
| $\rho_s$ | $330\ kg/m^3$ | Snow density (used in D-E) |
| $R_{fresh}$ | $100\ um$ | Freshly-fallen snow grain radius (used in D-E) |
| $R_{nonmelt}$ | $500\ um$ | Seasoned snow grain radius (used in D-E) |
| $R_{melt}$ | $1000\ um$ | Melting snow grain radius (used in D-E) |
| $c_w$ | 0.00536 | Ice-water drag coefficient |

follows. For atmosphere, the boundary layer process is carried out to determine the flux exchanges as well as wind stresses, following the coupling routine in CESM. For oceanic drag, the stress is parameterized with the coefficient $c_w$ and dependent on ice-ocean drift speed differences. Key parameters of CICE in our experiments are shown in Tab. B1.

## Appendix C: Scaling analysis for kinematics

5    With Arakawa-B grid staggering in CICE, we carry out the spatial scaling analysis of sea ice drift fields according to Marsan et al. (2004). A exemplary case with 9 local grid cells is shown in Figure C1. For the computing of a specific spatial derivate (i.e.,

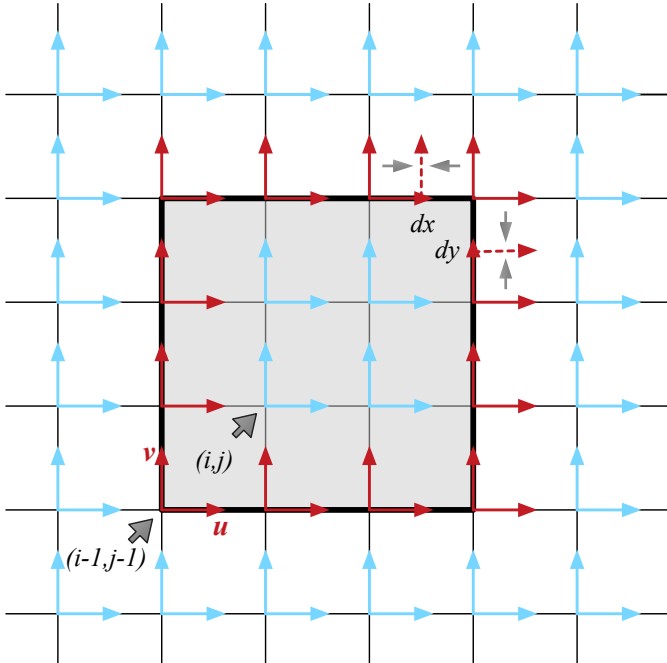

**Figure C1.** Example for spatial scaling analysis with a local area of 9 (i.e., 3×3) model cells. The horizontal (or vertical) direction is in the model's eastern (or northern) direction, indexed by discrete indices of $i$ (or $j$). The sea ice drift velocities that are used to compute the line integral for $\frac{\partial u}{\partial x}$ and $\frac{\partial v}{\partial y}$ are marked by red. For contrast, other velocities are marked by blue. Since velocities are defined on the top-right corner of each cell, they are interpolated on the cell edges to compute the integral. The interpolated velocites are marked by dashed vectors, corresponding to the averaging operation in Eqs. C2.

$\frac{\partial u}{\partial x}$), the line integral is computed to calculate the flux through the boundary of the specified area (Eqs. C2). The deformation rates can be computed according to Eqs. 1 through Eqs. 3. For the area from $i_1$ to $i_2$ and $j_1$ to $j_2$, the line integral consists of the flux computation over the eastern ($i = i_2$), western ($i = i_1 - 1$), northern ($j = j_2$), and southern ($j = j_1 - 1$) boundaries. The spatial scale is defined as the square root of the integrated cell areas.

$$\frac{\partial u}{\partial x}\Big|_{Length} = \frac{1}{Area}\oint u \cdot dy \tag{C1}$$

$$\frac{\partial u}{\partial x}\Big|_{i_1,i_2,j_1,j_2} = \frac{1}{\sum_{i=i_1}^{i_2}\sum_{j=j_1}^{j_2}dx_{i,j}\cdot dy_{i,j}}\left(\sum_{j=j_1}^{j_2}\frac{u_{i_2,j-1}+u_{i_2,j}}{2}\cdot dy_{i_2,j} - \sum_{j=j_1}^{j_2}\frac{u_{i_1-1,j-1}+u_{i_2-1,j}}{2}\cdot dy_{i_1-1,j}\right) \tag{C2}$$

*Author contributions.* SX designed the grid generation algorithm and carried out grid generation. SX and JM carried out model integration of TS grids in CESM, as well as numerical experiments. SX, LZ, JM and YZ analyzed the model outputs. All the authors contributed to the writing fo the manuscript.

*Competing interests.* The authors declare no conflict of interest.

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
