# Peer review of "Comparison of Sea Ice Kinematics at Different Resolutions Modeled with a Grid Hierarchy in Community Earth System Model (version 1.2.1)"

_Geoscientific Model Development, 2020_

## Referee Comment (RC1) · Nils Hutter (Referee) · 7 Jul 2020

The paper describes the creation of a multi-resolution suite of grids for CESM with a focus on their use for sea-ice modelling in the Arctic. The authors study the effect of grid resolution and number of EVP subcycling steps on the statististical properties of sea ice deformation as well as sea ice extent and volume. In particular, the localization of sea ice deformation in shear and failure lines such as leads and pressure ridges is studied. The authors present their model configuration as a starting point for more dedicated studies on sea ice dynamics and climate simulations and share the corresponding code and data. The simulations are analysed without optimising model

parameters to the specific grid resolution and the evaluation of the simulations need to be improved. Therefore, I recommend the manuscript for publication in Geoscientific Model Development after consideration of my general and specific comments.

_______________-

General comments:

1) The authors present untuned model runs with biased ice thickness fields (with too thick ice in the Beaufort Gyre) and also the ice volume in the simulations differs with the resolution used. However, the authors describe good agreement of sea ice coverage and volume of all simulations although a comparison with a sea ice thickness product, e.g. PIOMAS, is missing. So first, I suggest a thorough evaluation of the ice thickness and to study the differences in sea ice state between the different resolution simulations. I see two potential ways how to handle these different resolutions simulations that produce different sea ice state:

(a) If you are interested to study the effect of different resolution on sea ice dynamics (which is the topic of Section 3.3), all simulations should produce comparable sea ice distributions (concentration fields thickness as well volume and extent). Otherwise it is not possible to disentangle the effect of the change in resolution and the change in sea ice state on the dynamics. Systematic tuning methods (Massonnet et al., 2014; Ungermann et al., 2017; Sumata et al., 2019) could be used for all three simulations to optimize the parameter choices for each simulations by minimizing the model-observations misfit (for instance concentration, thickness, and drift). To resolve the issue of too thick ice in the Beaufort Gyre, drag coefficients and the ice strength parameterization could be tuned. The tuned simulations are then a good starting point for further multi-resolution studies and also the various parameters determined in the optimization will provide insight in how model physics change with resolution.

(b) If such systematic tuning is not possible due to limited computational resources, the authors should be more cautions with statements regarding the good agreement
of ice thickness fields and agreement of all three simulations. The differences between the three simulations should be described and interpreted in details. Possible reasons for the different sea ice distributions should be provided along with guidance what limitations with regard to the kinematic studies originate from the different sea ice distributions.

2) The analysis of deformation rates in the manuscript is limited to two 3-day intervals. Since the scaling properties of sea-ice deformation are highly variable (Stern & Lindsay, 2009) and strongly impacted by atmospheric winds (Herman & Glowacki, 2010), limiting the analysis to such a short time interval does not allow robust conclusions on the model capability to simulate multi-fractal deformation rates. It can not be excluded that the two dates chosen for the analysis mainly highlight the imprint of the atmospheric forcing. Another problem with the too short interval are the CDF of deformation rates that do not show power-law tails due to strong fluctuations (although stated differently by the authors). I suggest to extend this analysis to at least one entire winter. This will reduce the impact of specific wind conditions, smooth the CDFs, and allow a more robust interpretation of the presented results with regard to the models ability to simulated strongly localized deformation rates along leads and pressure ridges. In addition I recommend to remove all statements on temporal scaling based on these two 3-day intervals from this manuscript, as now temporal scaling analysis is performed by the authors.

3) The good agreement of deformation fields between the different resolutions surprised and impressed me. In your simulations only the degree of detail in deformation feature increases, but the general patterns agree across the different resolutions. Knowing that ice fracture is a chaotic process that is very sensitive to small variations in ice strength these results puzzles me, as I was expecting that the deformation fields diverge very fast due to the different deformation history. At high resolution, a deformation event which is associated with divergence reduces concentration and thickness, and thereby the ice strength, such that deformation is more likely to appear in the same

spot again. This effect should not be so effective in coarse resolution simulations as the reduction in concentration and thickness is much smaller due to the size of the grid box. This different memory should cause different reactions to the same atmospheric forcing. Do you see a reduction in concentration and thickness along the simulated LKFs in all your simulations? Do you see reoccuring deformation lines in all simulations? Your results indicate rather that in general this described feedback is not so strong and that fracture is mainly driven or better prescribed by the forcing, which would be an interesting result. This aspect of your results is definitely worth more discussion in the paper and maybe some additional analysis.

————————-

Specific comments:

P1, Line 3, "multi-fractality": of what? Please add scaling of sea-ice deformation

P1 Line 19, "kilometer-scale" satellite observations: SAR images have a resolution in the range of tens of meters. The drift and deformation products derived from consecutive SAR images have a kilometer-scale resolution. Please be more specific.

P2 Line 1, "Linear kinematic features": You have not described what these linear kinematic features are. Please describe once what they are (failure and shear lines where deformation is localized).

P2 Line 7: In the VP framework, the transition between viscous and plastic deformation depends on the stress states and not the concentration. The concentration influences the stress states by scaling the ice strength, but there is no direct link as suggested by your description. Please clarify.

P2 Line 14: CMI -> CMIP (here and elsewhere in the manuscript)

P2 Line 15: This is true for VP/EVP models. For other rheologies that include memory of past deformation, as the Maxwell elasto-brittle rheology, also coarser grid resolution might produce similar deformation statistics.

P2 Line 18: The continuum assumption is part of all continuum sea-ice models regardless what rheology they use. Please consider not explicitly mentioning the rheology here.

P2 Line 22, "main driver": It is not clear to me what you mean with main driver. Please clarify this sentence.

P4, Table1: Please be more specific with the grid descriptions in the "Notes: column, such that the table is understandable without reading the text. There is enough space for that.

P4 Line 21-22: at the grid location and 60 vertical layers,...

P4 Line 25: Please rewrite sentence.

P6 Figure 3: Please think about using the same limits for the contour plots for both grids. This would make it easier to see the difference between them. The contour lines are also hardly visible, you might also want to use a brighter red instead.

P8 Line 6-8: The thickness anomaly in Beaufort Gyre could also be caused by too weak ice and not properly tuned ice strength parameterization. The thick ice north of CAA and Greenland is then advected by the ice drift and accumulates within the Gyre.

P8 Line 13, "With the warm start-up, the experiments with TS005 approaches equilibrium towards year 42.": Only for the extent, the volume is still decreasing. Please clarify.

P8 Line 17, "The overall sea ice coverage and volume of TS005 is also in good agreement with satellite observations and PIOMAS dataset.": I would not describe the strong overestimation of sea ice extent in winter as a good agreement. In addition, I miss the comparison with the PIOMAS dataset in the figure. Please state where to find this comparison.

P8 Line 17-19: I do not understand why using the same parameterizations for all three

grids is a reason for reasonable results. It is known that model parameters need to be adapted to different grid resolutions to show similar physics (e.g. Williams & Tremblay, 2018). Please clarify or rewrite.

P9 Line 8, "removed of seasonal cycle": -> and the seasonal cycle is removed

P10 Figure5, "satellite-observed": Please state which satellite product is used for this comparison

P10-11 AO index analysis It is not clear why this analysis is needed here. As the corresponding explanation is rather complex, please consider to remove them from manuscript for clarity.

P11 Line 8: sybcycle count -> subcycles

P11 Line 17-18: "The kinematic features with TS005 are richer and much narrower, such as the network of shearing in Beaufort Sea." Do you want to say that in TS005 more and finer features are simulated?

P11 Line 35: The region for the analysis you have chosen is problematic as it mixes pack-ice regions with coastal regions. In coastal regions stable deformation features, like flaw lead, are found that show nearly constantly very high deformation rates, which impacts the presented CDFs. I suggest to use the entire Arctic Ocean as study region and filter all grid points that are closer than 150-200km to the coast as done in other scaling studies.

P12 Line 3: Please be cautious for two reasons: (1) just because the PDF/CDF of sea ice deformation shows a power-law tail does not mean it is multi-fractal. To show mulit-fractality a scaling analysis of the moments of sea-ice deformation need to be performed that shows a non-linear convex structure function (you do this analysis but it is described later). (2) The distributions shown in Figure 9 show hardly power-law distributions. I suggest to use the methodology of Clauset et al. (2009) to test for power-law distributions.

P12 Line 6: What do you mean with "spatial scaling"? Are you coarse-graining the high resolution simulation to coarser grid resolution? Please clarify.

P12 Line 6-21: (1) The CDF in Figure 9 hardly show power-law tails and deviate from strait tails. It is not clear how you determine the power-law slopes. I recommend to use larger time intervals for the analysis to reduce the imprint of certain atmospheric forcing conditions and second to use the methodology presented in Clauset et al. (2009) to test for power-law distributions.

(2) It is unclear to me how you relate the slopes the CDF-tails of coarse-grained deformation rates to the nominal resolution of the grid. Please describe this concept more in detail.

P12 Line 22-26: Given the limitations of your analysis (short-time interval, no clear power) I do not recommend a direct comparison with observations or at least mention these limitations.

P12 Line 32, "about 1.3 on Feb. 6th for all three grids": I see values from 1.2 to 1.3.

P13 L1-2, "Furthermore, no positive value of $\beta$ is detected at q = 0.5, which is consistent with Marsan et al. (2004) (Fig. 4 of the reference).": Please clarify. In Marsan et al. (2004) beta is positive for q=0.5. Also in your Figure 9 beta seems to be positive for q=0.5. What would be the physical interpretation of negative scaling exponents if you find them in your model?

P13 Line 6-8: Please be more specific. Do you mean that with increasing resolution, deformation rates are more localized with yields to more pronounced scaling?

P13 Line 8-9: Temporal scaling is indicated by the decrease in $\beta$ for the daily field and 3-day field for Dec. 20th, and not evident for Feb. 6th. This could be also caused by just smoothening of deformation fields due to advection. To test for for temporal scaling a Lagrangian analysis is needed that follows the ice deformation with the drift. Please remove this sentence or add analysis.

P13 Line 14, "indicating less dominant large-scale 15 features on Feb. 6th.": or a more heterogenous distribution of deformation rates along the LKFs.

P13 Line 15-16, "Furthermore, there is more effective temporal scaling on Dec. 20th than Feb. 6th, as shown for C-CDFs in Figure 9 and structure functions in Figure 10.": Please remove this sentence since no temporal scaling analysis is done.

P14 Line 4, "Figure 7": Do you mean Figure 12?

P14 Line 6, "noisier": It is really hard to spot the noise in Fig. 12 except you zoom very strongly in certain regions. Could you find better ways to show this? For instance, plot or average the difference between the deformation rates in a grid cell and its local surrounding (couple of grid cells). This would shift the focus on the noise. Or just zoom on a certain subdomain where the noise is seen.

P17 Figure 9: This figure needs more explanation in the caption: What do the colors refer to (NDTE?)? Are 0.05°, 0.15°, and 0.45° the grid resolution and why do you not use the names T005, etc. here?

P17 Line 5, "equilibrium in sea ice thickness and volume": But volume is still increasing, please clarify how this fits to the claimed equilibrium.

P22 Line 17: MITGcm -> MITgcm

P22 Line 20, "with initial study with temporal scaling analysis with 3-day mean drift fields": Please remove, since you have not done a temporal scaling analysis.

P23 Line 22-23: Remove one "in our study".

P22 Line 28 - P23 Line 3: This paragraph is rather a summary of on going research in the sea ice modelling community and your future plans and not a conclusion of your study. Please remove it here or move to the state of research in the introduction.

P23 Line 5: Which efforts? Please add citations.

P23 Line 30 is -> are

P24 Line 3-4, "Given that the modeled sea ice climatology is reasonable and consistent among the three resolutions": In the high resolution run, the sea ice climatology is distinctively different from the two coarser runs, which indicated that parameters of the sea-ice model need to be tuned for each specific resolution to reach the same climatology. I agree that using a slab-ocean in this study is fine, but further tuning of sea ice model parameters would be required to obtain runs with comparable sea ice climatology. Please elaborate on this.

References:

Stern, H. L. and Lindsay, R. W.: Spatial scaling of Arctic sea ice deformation, J. Geophys. Res.-Oceans, 114, c10017, https://doi.org/10.1029/2009JC005380, 2009.

Herman, A., & Glowacki, O. (2012). Variability of sea ice deformation rates in the arctic and their relationship with basin-scale wind forcing. The Cryosphere, 6(6), 1553–1559. https://doi.org/10.5194/tc-6-1553-2012

Clauset, A., Shalizi, C., and Newman, M.: Power-Law Distributions in Empirical Data, SIAM Rev., 51, 661–703, https://doi.org/10.1137/070710111, 2009.

Williams, J., & Tremblay, L. B. 2018. The dependence of energy dissipation on spatial resolution in a viscous-plastic sea-ice model. Ocean Modelling, 130, 40 – 47.

Massonnet, F., Goosse, H., Fichefet, T., and Counillon, F.: Calibration of sea ice dynamic parameters in an ocean-sea ice model using an ensemble Kalman filter, J. Geophys. Res.-Oceans, 119, 4168–4184, https://doi.org/10.1002/2013JC009705, 2014.

Ungermann, M., Tremblay, L. B., Martin, T., and Losch, M.: Impact of the Ice Strength Formulation on the Performance of a Sea Ice Thickness Distribution Model in the Arctic, J. Geophys. Res., 122, 2090–2107, https://doi.org/10.1002/2016JC012128, 2017.

Sumata, H., Kauker, F., Karcher, M., and Gerdes, R.: Simultaneous Parameter Optimization of an Arctic Sea Ice–Ocean Model by a Genetic Algorithm, Mon. Weather Rev., 147, 1899–1926, https://doi.org/10.1175/MWR-D-18-0360.1, 2019.

---

## Referee Comment (RC2) · Frederic Dupont (Referee) · 10 Jul 2020

General comments:

The authors present the results of one component of a new earth system, namely the sea-ice. They first introduced a new grid generation for a tripolar grid. Then, they show initial results of the sea-ice component under normal-year-forcing (a kind of climatological forcing with synoptic scales sampled from different years), including multi-scale analysis of the sea-ice deformation. Finally, they show the sensitivity of the dynamics of the model to a long-debated parameter in the dynamics, namely the number of subcycling in order that allows for artificially slow elastic waves in order to

solve for the viscous plastic equations. This is to my mind the most interesting bit.

The manuscript is well-written. The experiments are well described. The results are well presented and analyzed using existing diagnostic tools.

1-One comment about the grid generation, it looks very similar to the ORCA grid we use in NEMO, but I see only a short mention in Appendix (p25) to Madec and Imbard (1996). It would be important to mention how your method differs and improves on existing ones, otherwise it sounds like you are reinventing the wheel. One additional suggestion would be give more context on why you are doing this.

2-One general concern would be that the experiments are done with climatological atmospheric forcing and slab ocean, which means that, while it helps standardizing the experimental framework, it is not necessarily realistic as it lacks the increasing spectrum at high wave numbers present in the atmospheric and oceanic fields as one increases the resolution of the model.It is not a major problem but it would be worth discussing.

3-One important metric which seems to be missing is the ice drift error (only Fig.6 shows it for two different dates). Given the experimental framework chosen here, would it possible to add one? It is important to support that overall sea-ice volume, ice export out of the Arctic ocean or the convergence in the Beaufort gyre are reasonably modelled.

4-Another is that as Lemieux et al. (2012, JCP) showed, one cannot claim a true convergence of the EVP solver (it does not convergence in the numerical sense). Please also add a discussion on this and define what you mean by "convergence".

5-Since the paper goes relatively in depth in analyzing sea-ice deformation –which is usually an prelude for intense discussion on rheology models– I recommend that you broaden a bit more your discussion at the end (p.22-23). For instance, about the effect of the form of the yield curve in viscous-plastic models, Bouchat and Tremblay (2017,

JGR) for instance claims that decreasing the ice strengh and eccentricity improves their simulation (thickness, drift and deformation), while Ringeisen et al. (2018, cryosphere) claim that the angle in intersecting fractures from viscous plastic models is nowhere realistic...

Minor comments (given in the order of appearance in the text and figures):

1-any other changes to the ice physics except for ndte? There is only a reference to CESM D-type experiments and a short list of default schemes (p3 line 25 to p7 line 14), but it would be good that those are listed somewhere with chosen parameters.

2-line 14: operational forecast BASED on Dupont et al. (2015), i.e. we were not reporting on the operational implementation in this paper but on the general long hindcast prior to it.

3-line 22: main driver OF

4-line 32: NCEP CORE: requires a reference and likely mislabed (CORE 2 might be more accurate)

5-page 5, line 4-5 (and last column of table 2), giving the number of subcycling per hour is misleading as in CICE the restoring time for the elastic waves is function anyway of the larger transport+thermodynamic timestep. I see no argument in reporting this (#/hour) except artificially increasing the cycling number when resolution is below 1 degree. Table 2 in fact shows that you did not go above 1000 cycles per timestep. Please remove.

6-page 5, line 12, "ocean status" might be "ocean processes"?

7-Fig.5a: missing what years are used in the NSIDC climatology.for the comparison

8-the text refers to PIOMAS but Fig.5b does not show the comparison. Can you add it please?

9-p11 line 1, Nice analysis of NYF Artic Oscillation. I was always concerned of a

particular bias with repeatingly using NYF. So at least the winter wind pattern is mildly neutral. What about summer though? Can we say it is also neutral?

10-p13, line 17 "geostrophic" is mispelled

11-Fig.7: TS005 appears too smooth in the central Arctic (this is also noted in the text). Could it be an issue with the "convergence"?

12-Fig.9 has an unclear color key (it seems to be function of the run and spatial filter) and Fig.10: is missing one altogether. Please elaborate so that the figure is self-readable.

13-Fig.11: values are getting noisy past 1e-1 for TS045, sounds like a lack of resolution compared to TS015 and TS005.

14-Fig.13: interesting that ndte has such an impact on thickness even for the lowest resolution (TS045), whereas pattern of deformation are equivalent (top row of Fig.12). [please check that the top row is indeed showing different ndte results!]. I suspect this is because most of the changes in thickness are insise the Canadian Arctic Archipelago (CAA) where the deformation is not plotted. I suspect that the ice in the CAA is referred as "landfast", but it would be nice to have a more rigorous definition (is it in terms of some velocity threshold?).

---

## Referee Comment (RC3) · Véronique Dansereau (Referee) · 11 Jul 2020

This study compares EVP sea ice simulations at different spatial resolutions and with different level of convergence (i.e., number of subiterations of the solver) of the model solution. The comparison is made on the basis of the simulated deformation rates, which are analyzed in terms of their spatial distribution (fields), probability density function, cumulative probability density function and scaling properties, in space. Unlike recent studies which have used these metrics (e.g., Rampal et al., 2019, Hutter et al., 2018 and others), the authors use a climatology as their atmospheric forcing and analyze the simulated deformation rates after spin-up and stabilization of their modeled

seasonal cycle. Daily and 3-days mean deformation rate fields are used as the basis of their analyses. Two days are compared, which correspond to different AO scenarios and hence circulation patterns in the Arctic.

Analyses of the statistical properties of the simulated deformation rates (i.e., the shape of the PDF) are performed for different level of convergence of the model solution and demonstrate that simulation at higher resolution require a larger number of subiteration of the EVP solver to obtain heavy-tailed PDFs that are indicative of spatial scaling properties in sea ice deformation.

While the climatological approach is different than previous studies and perhaps eases the comparison of the modeled dynamics under different atmospheric forcing scenarios, it also precludes a direct, quantitative comparison between the model and observations, which I believe is a weakness of this study, especially considering that the physical processes that could be responsible for the difference in the results between the two atmospheric scenarios analyzed are only vaguely discussed. It also makes it hard to put the study into a temporal context (e.g., sea ice thickness and extent fields cannot be related to a specific time period and especially do not seem representative of recent sea ice conditions).

I note that a lot of care has gone into building the grid and chosing the model (atmosphere and ocean) components. These choices are clearly explained and justified. However, while a lot of information is given on the model grid, surprisingly little information is given on the dynamics part of the sea ice component (thickness resdistribution scheme parameters, rheology/mechanical parameters), which is obviously of high importance in determining the simulated dynamics. References should be included to redirect the reader towards the EVP parameter values used. I also suggest including a table with these values (P*, ellipse ratio, etc.) so that to avoid having to dig for these values into other papers and ease eventual comparisons to other similar scaling analyses.

Importantly, no information is given on the method used for the scaling analyses. A subsection to 3.2 that explains the steps taken towards these analyses should definitely be included in a revised version of the paper. Information on the impact of the choice of the region, period of time, the exclusion or not of grid cells close to coasts, the exclusion or not of scaling data points to evaluate the structure functions, etc., should also be given, as all of these factors can have a significant impact on the results. Also, how do your results differ if other days than Dec 10 and Feb 6 are chosen?

Overall, section 3.2, which presents the scaling and PDF analysis, is very hard to follow. It includes some contradictions, uses of wrong words, some important misunderstandings, etc. I make several specific comments to this effect below. The figures associated with this section are in my point of view incomplete, which makes the appreciation of the results difficult. I also give suggestions below on how to improve them.

Section 3.3 contains some intersting results on the effect of the convergence of the EVP model on the simulated dynamics.

Overall, the paper needs some proof-reading to improve the conciseness and accuracy of the formulations used. I also found many grammar mistakes but stopped raising them up at some point. I moreover found that sometimes, jargon-like formulations were used that unfortunately hide the real meaning of the sentences. An important point is the use of the term "multi-scale modeling" for what is really a comparison of model simulations across resolution. This crucially needs to be clarified.

In brief, I consider that major revisions are required. In my point of view, the points I raise in the specific comments below need to be adressed in a first time. Another review of the paper should be conducted in a second time, in order to better appreciate the results, their meaning and their importance, in the context of this study and provide further suggestions on how to improve the manuscript it its gobality.

Page 1, title: I find the use of the term ''multi-scale modeling" unfortunately misleading and inaproppriate. The paper is effectively about comparision of model simulations

performed at various spatial resolutions, while multi-scale modeling refers to codes that can effectively resolve processes occuring different space/time scales by coupling physically and numerically models of these specific processes. A more appropriate title would be "Comparison of sea ice kinematics at different spatial resolutions modeled with a hierarchy CESM..." or "Cross-resolution comparison of CESM sea ice simulations". Please also correct any mention to multi-scale modeling in the text for consistency.

Abstract, line 2 : "Sea ice kinematics is the most prominent feature of high-resolution simulations." There is no need to use high-resolution for kinematics features to be prominent in sea ice simulations. Please sea my comment just below about alternative rheological models, which do resolve the signature of kinematic features at medium and low resolution (> 20 km, Rampal et al., 2019).

Abstract, line 3 : "such as Viscous Plastic" current models are able to reproduce multi-fractality and linear kinematic features". This is one of my major comments: please be carefull as to make the distinction between multi-fractality in space and in time, throughout the entire text.

Abstract, line 4 : "we carry out multi-scale sea ice modeling". No, you carry out a comparison of simulations at different resolutions.

Abstract, lines 6-7 : "including multi-fractal deformation and scaling properties that are temporally changing". In the light of my other comments below, I would precise "multi-fractal deformation in space" and not put too much weight on the temporal part. Your abstract should highlight your strong results and the temporal aspect of the scaling analysis is not one.

Abstract, line 8 : "effective spatial resolution". This effective spatial resolution has not been defined and cannot be understood here. I believe you mean that the model can resolve kinematic features that are 6 or 7 times the width of a model's grid cell? If so, this should be explained clearly and in simple words (i.e., rewrite lines 8-9) in the

abstract and redefine later (see my other comment below).

Page 2, line 1 : "scale-invariance properties" Cite Marsan et al., 2004 there and Kwok et al., 2008 after "linear-kinematic features". Also, many other and more recent references can be added to Marsan et al., 2004 regarding scale-invariance, especially scale-invariance in time.

Page 2, line 6 : "most popular". A more objective term would be "most widely used".

Page 2, line 7-8 : "a plastic medium for packed ice under shear and pressure". This formulation is vague and unfortunately not accurate: the VP model describes sea ice as undergoing plastic deformation for over-critical shearing and compressive stresses only. Please modify the sentence accordingly.

Page 2, lines 15-19 : "In order to reproduce the observed properties of the sea ice kinematics, grids of 0.1 degree resolution or finer are usually required". This is true perhaps only in the VP or EVP rheology cases. The MEB rheology has the capability to localize deformation in space at the nominal grid cell scale, whatever the resolution of the grid (Dansereau et al., 2016, Rampal et al., 2019). Mention of this fact unfortunately come only in the conclusion, whereas an adequate literature review in your introduction should distinguish between the VP/EVP and other existing continuum rheologies (EB, MEB, Elastic-decohesive).

Page 2, line 20 : "multi-resolution sea ice modeling". I think that "we carry a comparison of sea ice model simulations at different spatial resolutions", or "cross-resolution comparison" would be clearer and more accurate.

Page 3, line 12 : "For the SP (...) For the NP". And the same for the lines below.

Page 4, line 4 : "a suite", a series?

Page 4, line 16 : This sentence is not clear: is there a repetition of "for TS015" there?

Page 4, line 19 : "within the Arctic Ocean".

[Figure]

Page 5, lines 2-3 : This sentence is unclear and a bit repetitive.

Page 5, line 4 : "a series", replace by "different subcycle numbers". Maybe rephrase as "We choose shorter thermodynamics and dynamics time steps for our higher resolution grids"?

Page 7, line 4 : "Potential compromises of using SOM". This needs to be rephrased, for instance as "Potential compromises pertaining to the use of SOM".

Page 7, line 6 : "in the Ocean Model Intercomparison Project".

Page 7, lines 8 to 12 : I understand here that you interpolate the same wind field onto your different (3) resolution grids. Does the interpolation ensures that the input (wind) energy is conserved accross resolutions? If not, this will impact your scaling results. I believe that a clear mention to this effect, in this paragraph, would be a valuable addition.

Page 7, lines 7-8 : Can you specify to which years corresponds the climatological annual cycle based on NCEP atm. reanalysis that you use? It would help understanding the ice coverage and thickness value that you obtain in your simulations at equilibrium (see my comment about these results just below).

Page 7, line 18 : Can you perhaps spell NDTE?

Page 7, line 25 to page 8, line 4 : You mention here a minor overestimation in the sea ice extent (cover) in some parts of the Arctic and underestimations in others. What is the basis for this comparision? From figure 5, I understand it is satellite sea ice edge data (from NSIDC), but this should be clearly mentionned in the text as well. Also the year or period of this satellite data should be mentionned with the corresponding years on which the climatology used to force your model is based.

Page 8, lines 4-5 : "consistent with existing sea ice thickness reconstructions by PI-OMAS". Also, in the same line as my previous comment, please mention the year for these PIOMAS thickness reconstruction, or insert a figure. It seems to me that there is

indeed a lot of ice stocked into the Beaufort Sea and that such thick ice conditions (up to 5 meters and more than 4 meters over a wide region, in September!) have not been seen at least in the last decade.

Page 8, line 10 : Can you explain in a few words what is a warm start-up?

Page 8, line 15 : "a minor decrease" of what? Please specify "both sea ice extent and volume" or merge this part of the sentence within the next one.

Page 8, line 30 : "two years' daily mean sea ice fields (...)". Rephrase, eg. "two years (41-42) of daily mean sea ice fields for all three TS grids".

Page 8, lines 28–30 : This is one of my major comment/concern. In this paragraph, you mention computing the deformation invariants from the daily mean sea ice drift speeds. This is the time scale set throughout your scaling analysis of daily deformation rates. You do not however mention how deformation rate components (du/dx, dv/dy, du/dy, dv/dx) are calculated, in particular at space scales larger than that of the cells of your Eulerian grids (with Arawaka-B staggering). Because you use Eulerian grids, I am guessing that your are following a coarse-graining method such as the one used in Marsan et al., 2004, but what are the details of the method? Do you, for instance, define square boxes and use a contour integral calculation to estimate each of the deformation rate components? Or just sum the components over each box? Most importantly, in estimating deformation rates at a given space scale, do you effectively sum (i.e., average) the deformation rate components and then calculate the corresponding invariants at that scale or do you sum (i.e, average) the deformation invariants themselves over that space scale? Also, why do you choose the region outlined in Figure 7 for your scaling and PDF analyses especially? How do you deal with the presence of coasts? Do you to eliminate data within a margin of the coasts? How do you deal with coarse graining boxes that might contain land regions? All of these details will most probably affect your scaling results and should be mentionned.

Section 3.2, pages 8-9 : I my point of view, readability would be improved if the second

and third paragraphs were included after the first sentence of paragraph one of section 3.2. Then, after paragraph 6, you a paragraph or sub-section describing your method for the estimation of deformation rates at different space scales, and of the scaling exponents, is necessary.

Page 11, line 22 : "There is good agreement". I think it would be more accurate to say that the results are consistent accross spatial resolution, since the comparison here is not done on the basis of observational data.

Page 11, line 23 : "large shearing belts accross the basin". There are indeed large shearing belts and diffuse regions of shearing rates seen at all model resolutions. Are these diffuse shearing belts physical? How do they compare, for instance, to shearing rates fields inferred from RadarSAT data? To what process, physical or numerical, do you think they are related?

Page 11, line 25 : "There is a clear".

Page 11, line 26 : Please change "more well-defined" by "better defined".

Page 11, lines 31-33 : The last sentence of this paragraph is not clear. Please define clearly what you mean by "effective" resolution of the model. I guess it corresponds to the width of the simulated LKFs?

Page 11, line 34 : "the distribution of total deformation rates follow power-law distributions". What distribution? Please be precise here, e.g., "the statistical (or probability density) function of total deformation rates follow a power law".

Page 12, line 1 : I find the accronym C-CFD to be confusing here. You are calculating the cumulative probability density function of both daily and 3-days mean deformation rates. This term and the accronym cumulative PDF is used in most published scaling analyses within the sea ice community. I suggest for clarity that you use similar terms.

Page 12, line 3 : "For both daily and 3-days cumulative distributions, we attain multi-fractality across the three resolutions". I am confused here: how do you conclude that

deformation rates are multifractal from the cumulative distributions in Figure 9? This information is rather given by a scaling analysis based on different moments of the distribution of deformation rates and the estimation of the convexity of the quadratic function describing the dependance of the scaling exponents on the moment. Please clarify or remove.

Page 12, lines 7-21 : This paragraph is very confusing and I do not understand your method here. First you say that you carry out the spatial scaling of one grid onto another? How do you do that? Do you mean that you interpolate deformation rates from one grid to another? Line 7 : you mention that the slopes become steeper for the higher resolution grids for one given day but not the other. I suspect you mean the slope of the "CFDs/C-CDFs" in log-log space? How do you compute these slopes from figure 9? Can you please show these slopes on the figure so that one can evaluate at least qualitatively the goodness of fit? Lines 9-10 : "the slopes of C-CFDs from scaled rates". Do you mean interpolated doformation rates? Also, putting all of the curves on each panel of figure 9 makes it very difficult to read the figures. I would suggest separating the "non-scaled" or non-interpolated and interpolated results on different figures, or use different levels of opacity for the non-interpolated and interpolated results. Lines 13-14 : what is a realistic shape for the distribution? On what data do you base your evalution of a "realistic shape"? Also, a realistic shape for a "power-law distribution" is by definition a power law! Hence I suggest you write simply "a realistic shape of the distribution of deformation rates". Also see my previous comment about defining the "effective" resolution of the model. Line 16 : "we attain the same slope" Lines 18-19 : "the CFD of sea ice deformation rates", not kinematics. Also, please explain how you evalute this effective resolution, which is 6-7 times higher than that of the TS015 grid and what are the different days that you are analyzing.

Page 12, lines 22-26 : Why do you think the (absolute) slopes you are estimating are smaller than that of Marsan et al. 2004 at a similar time scale? It would be relevant to offer possible explanations here.

Page 12, line 26 : "to evaluate the structure function".

Page 12, line 31 : "At q = 3, the structure function is in the range...". Do you mean beta instead of the structure function?

Page 12, last paragraph : I do not think it is relevant to cite the differences in the values of beta(q) or in the shape of the structure functions between the two analyzed days if you do not try to explain these differences physically.

Page 12, line 35 : "the average deformation rate". It would be more specific to refer to the mean deformation rate or to the moment of order 1.

Figure 10 : On the scaling figures (left panels) please indicate the moment order corresponding to each set of curves and insert a legend for the different colors/model resolutions. Also, you label the y-axis with epsilon for the total deformation rate, whereas in equations (1) to (3) you use dot(varepsilon) (indeed not available in MATLAB) for this variable and the other deformation invariants. Please use consistant symbols accross the text and figures. On the x-axis of the same figure, you use the label "space scaling", which would rather be appropriate as a title for these figures. I believe you mean "space scale". Your structure function is estimated using the moment of order 0.5, but it is not shown in the scaling analyses (left panels), why? Your structure function results could also be appreciated more objectively if you included error bars for beta for each of the moments (see e.g., Rampal et al, 2019 for the definition of the error bars on beta(q)).

Most importantly, for the December case in particular it is apparent from the scaling figures (left panels) that the slope (beta) of the moments of order 2 and 3 is calculated by leaving out at least the two last points of the scaling analysis, corresponding to the largest space scales. Why is that and how considering/rejecting these points affects your results? If some data points are left out or attributed less weigth in the analyses, this should be definitely be clearly mentionned and argued for in the text.

Page 13, line 4 : Please specify that your result support multifractality of the simulated sea ice deformation in space (i.e, not in time), hence not "multifractality of sea ice kinematics".

Page 13, line 5 : How does the inclusion of the deformation rates at the two larger space scales for the moments 2 and 3 change the value of the estimated curvature? (see my previous comment on figure 9).

Page 13, line 6-8 : This sentence is a generic comment and does not offer a satisfactory explaination for the difference in results between the two time period analyzed. Either offer some physical hypotheses or refrain from comparing the two results.

Page 13, line 8-9 : If there is a slight drop between the values of beta or the curvature of the structure functions between the daily and 3-days deformation rates and this difference is not evident for the Feb. 6 case, my opinion is that this is no sufficient evidence that the model can reproduce temporal scaling. Such assumption should rather be based on a proper temporal scaling analysis that spans several orders of time scales, not a comparision between 1 and 3-days fields. This comment in my point of view should be removed because not supported from your results (and mention to it should be removed from the abstract as well, which should only state your strong results).

Page 13, line 12 : "existing studies with observational datasets and modeling results". Please put some references here.

Page 13, line 14-15 : Why is there a comparison here? This does not make sense. Do you mean "more convex on Feb. 6 than Dec. 20"? Also, why would this support less dominant large-scale features on Feb. 6? Please clarify.

Page 13, line 15 : What is a ''more effective temporal scaling"? How does the results on figure 9 and 10 support temporal scaling?

Page 13, lines 16-24 : I do not understand the link between these sentences and the

previous sentences of this paragraph. What point are you trying to make? Please explain. Also, I think it would be more accurate not to refer to Lindsay et al., 2003 as current RGPS observations. RGPS is currently not running. Maybe just drop the reference.

Page 13, line 26 : "we evaluate the sensitivity? of the modeled kinematics".

Page 13, line 27 : "the probability density function (PDF)".

Page 13, line 28 : and figure 11 : I suggest adding the estimated slope of the tail of each PDF on the graphs of figure 11 to illustrate how you estimate it from your results. It would also help putting a Gaussian distribution on each graph to visually identify the fat tails of the PDFs.

Page 13, line 28 : "for the total deformation rate".

Page 13, Section 3.3, 1st paragraph : To support your claim that insufficient convergence of the model solution (i.e., nb of subcycles) is responsible for the absence of convergence between the PDFs of the simulated distribution rates at high resolution, I think you could also mention that the tail of the log-log PDF at the higher resolution and lowest number of sub-iteration does not seem linear as in the other cases.

Page 14, line 6 : I think you mean figure 12, not 7.

Page 14, line 8 : I think you mean figure 11, not 12.

Page 14, line 8 : What is the "physical" deformation rate? Please explain more clearly.

Page 14, lines 10-12 : Instead of convergence of the kinematics, it is the convergence of the model solution, or simulated kinematics. Also, instead of "deterioration of simulation speed", I would write "increase in simulation time or cost" for more clarity.

Page 15, line 1 : "in the Canadian Arctic Archipelago". Line 3, "the CAA" again. Same mistake in other places on page 18.

Page 15, line 2 : "an ice arch"

Page 15, lines 8-10 : It would be helpful for the reader if you could further explain how you can separate these contributions in your model, by showing the term of the the dynamics/thermodynamics equation associated with each of them.

Page 18, line 2 : "due to the fact that the thermodynamic growth".

Figure 13 : Please increase the fonts of the figure titles and colorbars.

Page 22, line 1 : Please rephrase that sentence, which is not clear at all, both in the meaning and construction.

Page 22, line 2 : Change "grid stepping" for "grid resolution".

Page 22, lines 6-8 : "Multi-fractal sea ice deformation is accurately modeled by all three resolutions": please see my comment above on the inclusion or not of the all the points in your scaling analysis for the case of Dec 20. "with good agreement with observational works in terms of scaling properties" : have you try to compare the slopes of the scaling analyses (beta) for the three moments and the curvature of the structure functions with observational analyses at equivalent spatial resolution, e.g., based on RGPS data? If not, this comment should be revised.

Page 22, line 12 : "multi-scale modeling studies". Please consider my previous comment on the meaning of "multi-scale modeling".

Page 22, line 19-21 : This sentence is incomprehensible, what is the "initial study with temporal scaling analysis with 3-day mean drift fields"? I did not see these results. If you are referring to the comparison of the daily mean and 3-days mean results, see my previous comment about how a much larger range of timescales would be necessary to conduct a meaningful temporal analysis.

Page 22, line 21 : Repetition of "in our study".

Page 22, lines 23-27 : Scaling analyses of modeled deformation fields and their com-
parison with equivalent analyses of observed deformation fields date from around 2010. Such analyses and model-observation comparisons have been made by only a few sea ice research groups and besides, techniques for comparing accurately Lagrangian/Eulerian model outputs to observational deformation data, especially in the context of time scaling analyses, are complex and have been recently developped and applied. Hence I would not qualify scaling analysis as a "traditional" tool for evaluating sea ice kinematics. For further validating sea ice deformation properties simulated with your multi-resolution framework, I also suggest a comparison of simulations to observations of sea ice deformation.

Page 22, lines 31 to 33 : It is clearly stated in the paper by Rampal et al., 2019 that the MEB rheology of Dansereau et al., 2016 is used, not the EB rheology of Girard et al., 2011. Please read the paper and correct your sentence. Also, replace "which are shown" by "which is shown" and for a demonstration of the MEB model capability to localize deformation at the nominal grid cell scale, whatever the grid resolution, which explains the fact that neXtSIM does not encounter "effective" resolution issues and does not require using a sub-LFK spatial resolution to simulate adequately these features, therfore reproducing the scaling properties accross model resolutions, see Dansereau et al., 2016.

Page 22, line 6 : It is hard from the figures shown here only to witness the asymptotic convergence of the simulated modeled kinematics. Please remove that sentence or include a figure that shows this clearly.

Page 23, line 6 : "we have witnessed that".

Page 23, line 15 : What is your choice of ice strength parameterization scheme? See my previous comment about the importance of including at least a reference to the dynamic equations of the model and a table listing the EVP rheology parameter values.

Page 23, line 29 : "Multi-scale simulations". I belive you mean that the comparison of simulations across spacial resolutions is becoming common in the climate modeling

community.

Véronique Dansereau

Please also note the supplement to this comment:
https://gmd.copernicus.org/preprints/gmd-2020-160/gmd-2020-160-RC3-
supplement.pdf

---

## Author Comment (AC1) · 10 Sep 2020

The authors would like to thank Dr. Frederic Dupont (the referee) for the invaluable comments. The following are the replies for each comment, together with specific revisions that are made. The original comments are in *green italic* font, and the revisions highlighted in the revised manuscript in cyan.

**Reply to comments:**

**General comments:**

The authors present the results of one component of a new earth system, namely the sea-ice. They first introduced a new grid generation for a tripolar grid. Then, they show initial results of the sea-ice component under normal-year-forcing (a kind of climatological forcing with synoptic scales sampled from different years), including multi-scale analysis of the sea-ice deformation. Finally, they show the sensitivity of the dynamics of the model to a long-debated parameter in the dynamics, namely the number of subcycling in order that allows for artificially slow elastic waves in order to solve for the viscous plastic equations. This is to my mind the most interesting bit.

The manuscript is well-written. The experiments are well described. The results are well presented and analyzed using existing diagnostic tools.

1-One comment about the grid generation, it looks very similar to the ORCA grid we use in NEMO, but I see only a short mention in Appendix (p25) to Madec and Imbard (1996). It would be important to mention how your method differs and improves on existing ones, otherwise it sounds like you are reinventing the wheel. One additional suggestion would be give more context on why you are doing this.

**Reply to the general comment (1)**: the authors fully acknowledge the pioneering work on ocean model grid generation in Madec & Imbard (1996), as well as the similarity with our study. Specifically, in Madec & Imbard (1996), the details of the grid generation method with embedded ellipses are only briefly mentioned by the end of the paper. The resulting grids are adopted further in NEMO with various grid resolutions in different applications. However, there are three major reasons that we design and implement the grid generation method for our own use. First, we use CESM (v1.2.1) and its component models for all the experiments. Both POP and CICE use Arakawa-B staggering with a U-fold for the tripolar grid. On the other hand, as far as I know, NEMO uses Arakawa-C staggering with a T-fold tripolar grid. Although a seemingly minor issue, it does greatly affect all the numbering with many associated code changes to existing grid generation method.

Second, we want to have more flexible control of all the grid generation method and tool for the grid. The requirements include the control of grid cell size anisotropy in the polar regions, the grid size transition to the two grid poles on Eurasia and North American continents. For the TS grids in study we have adopted polynomial form (Fig. A1.b and Appendix A), but other options are possible, including exponential function and sinusoidal function. Different forms result in differences in grid scales in the Arctic region. Therefore we are yet to explore the various options of the grids, including resolution differences in CAA. Utilizing a fully-fledged grid generation toolkit enables aforementioned tasks.

Finally, the proposed tripolar grid generation method, including its implementation, serves as the basis for a comprehensive toolset for future development. In specific, we look forward to combining several grid generation methods into a complete toolset, including tripolar grids

(as in this study) and coastline/bathymetry-following techniques with complex conformal mappings (Xu et al., 2015).

In summary, we attain more flexibility with the proposed grid generation method for our ongoing model development. The similarity between Madec & Imbard (1996) and this study is that both adopt embedding ellipses, but the specific methods differs including in the construction of these ellipses as well as the numerical method to construct the grid lines.

2-One general concern would be that the experiments are done with climatological atmospheric forcing and slab ocean, which means that, while it helps standardizing the experimental framework, it is not necessarily realistic as it lacks the increasing spectrum at high wave numbers present in the atmospheric and oceanic fields as one increases the resolution of the model. It is not a major problem but it would be worth discussing.

**Reply to the general comment (2)**: the authors would like to reemphasize that the purpose of using Normal-Year Forcing based experiments, as also stated by the referee, is to align the experiments across the different grids. The NYF dataset includes synoptic signals from the atmospheric reanalysis, but lacks inter-annual variability as well as long-term trend. Therefore, NYF is also used in Ocean Model Inter-comparison Project (OMIP) to study climatological and equilibrium status of the ocean-sea ice coupled system. In response, in Section 3.1 and 4, we add further discussion of the methodology of using NYF in our study, and potential extension to Inter-Annual Forcing (IAF) dataset in the future.

3-One important metric which seems to be missing is the ice drift error (only Fig.6 shows it for two different dates). Given the experimental framework chosen here, would it possible to add one? It is important to support that overall sea-ice volume, ice export out of the Arctic ocean or the convergence in the Beaufort gyre are reasonably modelled.

**Reply to the general comment (3)**: regarding this comment, the authors have made the 2 revisions and an add-on experiment for the attribution of extra-thick ice in Beaufort Gyre. First, we have carried out comparison of Arctic basin sea ice drift for winter months. In specific, we have compared DJF sea ice drift from the model output against the climatological sea ice drift of NSIDC (corresponding months for 1980-2000). Second, we re-evaluate the modeling results against comparable sea ice volume by PIOMAS. The annual cycle of PIOMAS sea ice volume for the year from 1980 to 2000 is used to compute the climatology.

The add-on experiments mainly dealt with the sensitivity of modeled Arctic sea ice volume (especially in Beaufort Sea) to ice strength parameterization. In our experiments, as well as those in standard CESM settings, the ice strength parameterization follows Rothrock (1975) and Flato & Hilber (1995), instead of Hibler (1979) which is arguably more widely used. In specific, in the strength parameterization adopted by CESM (Lipscomb et al., 2007 JGR), the sea ice strength is assumed to be proportional to the change in ice potential energy with respect to compressive deformation, which in turn relates to the ridging (with exponential redistribution) and the specific ITD settings. For contrast, in Hibler (1979), the ice strength is directly computed with prognostic status including sea ice thickness (in linear relationship), concentration, as well as prescribed parameters of which Pstar is treated as a tunable constant. As a result of the choice of strength parameterization, the equilibrium sea ice thickness across the basin is considerably different, with thick sea ice in Beaufort Sea absent when using Hibler (1979). The figure below (Fig. 1) shows the comparison with TS045 grid.

Besides, we have also confirmed that this phenomenon is qualitatively similar with CESM's native grid (e.g., GX1V6, shown in Fig. 2), as well as TS015 (not shown). Given that Lipscomb et al., (2007, JGR) is used by default in scientifically checked cases of CESM relevant to our study, we consider this choice a reasonable choice for our experiments and analysis here. However, we consider this an important issue that should be further explored in future studies.

Fig. 1. March sea ice thickness difference for TS045 grid at equilibrium states under CESM D-type experiments. Left: Rothrock (1975) scheme; Middle: Hibler (1979) scheme; Right: difference (middle *minus* left). All values are in meters.

Fig. 2. March sea ice thickness difference for GX1V6 grid at equilibrium states under CESM D-type experiments. Left: Rothrock (1975) scheme; Middle: Hibler (1979) scheme; Right: difference (middle *minus* left). All values are in meters.

4-Another is that as Lemieux et al. (2012, JCP) showed, one cannot claim a true convergence of the EVP solver (it does not convergence in the numerical sense). Please also add a discussion on this and define what you mean by "convergence".

**Reply to the general comment (4)**: the authors apologize for using "convergence" in a less strict sense. The convergence we mentioned for EVP is not in the strict numerical sense. Rather, there is asymptotic behavior of the sea ice deformation and kinematic features with respect to EVP subcycling count. Fig. 7, 8 and 11 show the asymptotic convergence. For example, the PDFs of the deformation rates (Fig. 11) attain a good match when the NDTE is

large enough especially for TS045 and TS015. Also with larger NDTE values, the kinematics features become more refined in structure and match well when NDTE goes beyond a certain value (e.g., comparing NDTE=480 and NDTE=960 for TS045). Here we revise the manuscript of Sec. 3.3 by adding extra text to formally explain what we mean by "convergence".

5-Since the paper goes relatively in depth in analyzing sea-ice deformation –which is usually an prelude for intense discussion on rheology models– I recommend that you broaden a bit more your discussion at the end (p.22-23). For instance, about the effect of the form of the yield curve in viscous-plastic models, Bouchat and Tremblay (2017, JGR) for instance claims that decreasing the ice strengh and eccentricity improves their simulation (thickness, drift and deformation), while Ringeisen et al. (2018, cryosphere) claim that the angle in intersecting fractures from viscous plastic models is nowhere realistic...

**Reply to the general comment (5)**: following the referee's comment, the authors have added extra discussion in Section 4 on rheology models and related dynamics-related parameterization schemes, especially regarding related works. In particular, we consider the ice strength parameterization a key discussion point here. As many works on tuning the model have adopted the scheme of Hibler (1979) [including Bouchat & Tremblay (2017) which is also mentioned by the referee], the most straightforward tuning parameters are Pstar and eccentricity of the yield ellipse. But for the scheme we have adopted, we have not observed much tuning work especially regarding basin-scale simulations. One particular parameter that can be subjected to tuning is the empirical parameter of C\_f that is used to characterize frictional energy dissipation, which can be scale-dependent (details in CICE documents).

**Minor comments (given in the order of appearance in the text and figures):**

1-any other changes to the ice physics except for ndte? There is only a reference to CESM Dtype experiments and a short list of default schemes (p3 line 25 to p7 line 14), but it would be good that those are listed somewhere with chosen parameters.

**Reply**: we confirm that the thermodynamic parameterizations are kept the same across all 3 grids and different NDTE values. In our original design of the experiments, the parameters are kept the same in order to improve the comparability of the simulation results. To revise, we have now added a separate appendix (Appendix B) that includes all parameterization schemes and related parameters that are used in the experiments, including both thermodynamics and dynamics.

2-line 14: operational forecast BASED on Dupont et al. (2015), i.e. we were not reporting on the operational implementation in this paper but on the general long hindcast prior to it.

**Reply**: the text is corrected to be more precise, as "the hindcast experiments for operational sea ice forecasts in Dupont et al. (2015)".

3-line 22: main driver OF

**Reply: corrected.**

4-line 32: NCEP CORE: requires a reference and likely mislabed (CORE 2 might be more accurate)

**Reply**: corrected to be CORE 2.**

5-page 5, line 4-5 (and last column of table 2), giving the number of subcycling per hour is misleading as in CICE the restoring time for the elastic waves is function anyway of the larger transport+thermodynamic timestep. I see no argument in reporting this (#/hour) except artificially increasing the cycling number when resolution is below 1 degree. Table 2 in fact shows that you did not go above 1000 cycles per timestep. Please remove.

**Reply**: according to the referee's suggestion, the authors have removed the last column which showed the total subcycling count per model hour. Besides, we confirm that the maximum subcycle count per timestep is 960, which is indeed under 1000 cycles.

6-page 5, line 12, "ocean status" might be "ocean processes"?

Reply: revised as indicated, by changing "ocean status" into "ocean processes".

**7-Fig.5a: missing what years are used in the NSIDC climatology.for the comparison**

**Reply**: the climatological sea ice extent is computed as the annual cycle of monthly mean sea ice extent for the year from 1979 to 2000. The text is also revised to include this information.

8-the text refers to PIOMAS but Fig.5b does not show the comparison. Can you add it please?

**Reply**: the authors have added the climatological sea ice volume annual cycle from PIOMAS on Fig. 5.b. The same years (1979-2000) from PIOMAS are used to compute the annual cycle.

9-p11 line 1, Nice analysis of NYF Artic Oscillation. I was always concerned of a particular bias with repeatingly using NYF. So at least the winter wind pattern is mildly neutral. What about summer though? Can we say it is also neutral?

**Reply**: we have extended the analysis of AO to the summer months of NYF dataset, which is now incorporated into Fig. S1. In general, the summertime AO index pf CORE2 NYF is also neutral.

10-p13, line 17 "geostrophic" is mispelled

**Reply: corrected.**

11-Fig.7: TS005 appears too smooth in the central Arctic (this is also noted in the text). Could it be an issue with the "convergence"?

**Reply**: we do not consider the relative lower deformation rates in Fig. 7 an abnormal behavior of TS005. As sea ice thickness is generally thinner on the outer rim of the sea ice cover (i.e., marginal seas), the deformation rates tend to be higher as well (Kwok, J-Glaciol., 2010). In the central Arctic, for the two representative days, the seemingly smooth deformation field actually have certain shearing but low convergence/divergence (lower left panel of both Fig. 7 and 8). The convergence we mentioned is related to the concentration of deformation to local regions when NDTE is large enough.

**12-Fig.9 has an unclear color key (it seems to be function of the run and spatial filter) and Fig.10: is missing one altogether. Please elaborate so that the figure is selfreadable.**

**Reply**: both Fig. 9 and Fig. 10 are revised to include more information of the scaling curves for different resolutions. Texts that correspond to these figures are also revised.

**13-Fig.11: values are getting noisy past 1e-1 for TS045, sounds like a lack of resolution compared to TS015 and TS005.**

**Reply**: at the deformation rate of 10%/day for TS045 (shear, div, and total deformation) appear to be noisy, since with lower resolution of TS045, we do not have sufficient samples to compute the proper shape (i.e., tail) of the PDF. Therefore, we consider this a sampling issue, rather than the direct issue of the lack of resolution.

14-Fig.13: interesting that ndte has such an impact on thickness even for the lowest resolution (TS045), whereas pattern of deformation are equivalent (top row of Fig.12). [please check that the top row is indeed showing different ndte results!]. I suspect this is because most of the changes in thickness are insise the Canadian Arctic Archipelago (CAA) where the deformation is not plotted. I suspect that the ice in the CAA is referred as "landfast", but it would be nice to have a more rigorous definition (is it in terms of some velocity threshold?).

**Reply**: the difference in deformation rate between NDTE values for TS045 are mainly in the range of small values (less than 0.5%/day), as shown in the top row in Fig. 12. Note the color difference in CAA. As indicated by the referee, we revise the manuscript adopting the sea ice drift speed criterion of characterizing sea ice as landfast when the 2-week mean drift speed is lower than 5e-4m/s, as introduced in Lemieux et al (2015). The region of landfast ice (with the newly adopted criterion, as a supplementary figure) agrees well with the region of analysis.

**References:**

- Bouchat, A., and Tremblay, B. (2017), Using sea-ice deformation fields to constrain the mechanical strength parameters of geophysical sea ice, J. Geophys. Res. Oceans, 122, 5802–5825, doi:10.1002/2017JC013020.
- Dupont, F., Higginson, S., Bourdallé-Badie, R., Lu, Y., Roy, F., Smith, G. C., Lemieux, J.-F., Garric, G., and Davidson, F. (2015). A high-resolution ocean and sea-ice modelling system for the Arctic and North Atlantic oceans, Geoscientific Model Development, 8, 1577–1594, doi:10.5194/gmd-8-1577-2015
- Flato, G.M. and W. D. Hibler (1995). Ridging and strength in modeling the thickness distribution of Arctic sea ice. J. Geophys. Res.–Oceans, 100:18611–18626.
- Hibler, W.D. (1979). A dynamic thermodynamic sea ice model. J. Phys. Oceanogr., 9:817–846.
- Kwok, R. (2010). Satellite remote sensing of sea-ice thickness and kinematics: a review, Journal of Glaciology, 56(200):1129-1140
- Lemieux, J.-F., Tremblay, L. B., Dupont, F., Plante, M., Smith, G. C., & Dumont, D. (2015). A basal stress parameterization for modeling landfast ice. Journal of Geophysical Research: Oceans, 120, 3157–3173. https://doi.org/10.1002/2014JC010678
- Lipscomb, W.H., E. C. Hunke, W. Maslowski, and J. Jakacki (2007). Improving ridging schemes for high-resolution sea ice models. J. Geophys. Res.–Oceans, 112:C03S91, doi:10.1029/2005JC003355
- Madec, G. and Imbard, M. (1996). A global ocean mesh to overcome the North Pole singularity, Climate Dynamics, 12, 381–388, doi:10.1007/BF00211684
- Ringeisen, D., Losch, M., Tremblay, L. B., and Hutter, N. (2019). Simulating intersection angles between conjugate faults in sea ice with different viscous-plastic rheologies, The Cryosphere, 13, 1167–1186, doi:10.5194/tc-13-1167-2019
- Rothrock, D.A. (1975). The energetics of the plastic deformation of pack ice by ridging. J. Geophys. Res., 80:4514–4519

---

## Author Comment (AC2) · 10 Sep 2020

The authors would like to thank Dr. Véronique Dansereau (the referee) for the invaluable comments. The following are the replies for each comment, together with specific revisions that are made. The original comments are in *green italic* font, and the revisions are highlighted in the revised manuscript in red.

**Reply to comments:**

*This study compares EVP sea ice simulations at different spatial resolutions and with different level of convergence (i.e., number of subiterations of the solver) of the model solution. The comparison is made on the basis of the simulated deformation rates, which are analyzed in terms of their spatial distribution (fields), probability density function, cumulative probability density function and scaling properties, in space. Unlike recent studies which have used these metrics (e.g., Rampal et al., 2019, Hutter et al., 2018 and others), the authors use a climatology as their atmospheric forcing and analyze the simulated deformation rates after spin-up and stabilization of their modeled seasonal cycle. Daily and 3-days mean deformation rate fields are used as the basis of their analyses. Two days are compared, which correspond to different AO scenarios and hence circulation patterns in the Arctic.*

*Analyses of the statistical properties of the simulated deformation rates (i.e., the shape of the PDF) are performed for different level of convergence of the model solution and demonstrate that simulation at higher resolution require a larger number of subiteration of the EVP solver to obtain heavy-tailed PDFs that are indicative of spatial scaling properties in sea ice deformation.*

*While the climatological approach is different than previous studies and perhaps eases the comparison of the modeled dynamics under different atmospheric forcing scenarios, it also precludes a direct, quantitative comparison between the model and observations, which I believe is a weakness of this study, especially considering that the physical processes that could be responsible for the difference in the results between the two atmospheric scenarios analyzed are only vaguely discussed. It also makes it hard to put the study into a temporal context (e.g., sea ice thickness and extent fields cannot be related to a specific time period and especially do not seem representative of recent sea ice conditions).*

**Reply to the major comment**: the authors thank the referee's comment on the comparability aspects of our study. We agree with the referee on the comparison of model output with observations. We would like to make the following clarifications. First, the purpose of the experiment methodology, including using atmospheric NYF dataset as well as coupling to slab ocean model, is to increase comparability across the resolution range. Second, since the NYF dataset is used to force the ice model, there is no direct comparable observation available. In turn, the comparison we have carried out are mainly between different (grid) resolutions, including deformation, scaling properties, and EVP convergence. Third, regarding the limited results and analysis as pointed out by the referee, for revisions, we extend the analysis beyond the two representative days to the whole winter, and carry out attribution study of the scaling properties and differences among the different resolutions. Lastly, this work is a first step of our effort in using the grid hierarchy with the ocean-ice coupled settings and inter-annually forced experiments. Both aspects (coupling to dynamic ocean and IAF experiments) would require a much more higher computational overhead (especially for the TS005 grid) which we look forward to exploration in the future.

*I note that a lot of care has gone into building the grid and chosing the model (atmosphere and ocean) components. These choices are clearly explained and justified. However, while a lot of information is given on the model grid, surprisingly little information is given on the dynamics part of the sea ice component (thickness resdistribution scheme parameters, rheology/mechanical parameters), which is obviously of high importance in determining the simulated dynamics. References should be included to redirect the reader towards the EVP parameter values used. I also suggest including a table with these values (P\*, ellipse ratio, etc.) so that to avoid having to dig for these values into other papers and ease eventual comparisons to other similar scaling analyses.*

**Reply to the major comment**: regarding the referee's comment, we now add a separate appendix (Appendix B) to include all the parameterization schemes and the specific values of parameters we have adopted in the experiments. All the schemes, especially thermodynamic processes are kept the same as much as possible across the 3 resolutions, for the sake of improved comparability.

*Importantly, no information is given on the method used for the scaling analyses. A subsection to 3.2 that explains the steps taken towards these analyses should definitely be included in a revised version of the paper. Information on the impact of the choice of the region, period of time, the exclusion or not of grid cells close to coasts, the exclusion or not of scaling data points to evaluate the structure functions, etc., should also be given, as all of these factors can have a significant impact on the results. Also, how do your results differ if other days than Dec 10 and Feb 6 are chosen?*

**Reply to the major comment**: regarding to comment on the lack of the details of the methodology for scaling analysis, we now include another appendix (Appendix C) to cover these details. Specifically, the method involves line integrals that cover different spatial scales in the model's grid system, which follows Marsan et al. (2004) and Rampal et al. (2019). Besides, we update the region of scaling analysis and remove those close to land, and the results are also updated accordingly. Last but not the least, besides Dec. 20th and Feb. 6th, we extend the analysis of daily deformations to winter months, and in Sec. 3 the new contents are added.

*Overall, section 3.2, which presents the scaling and PDF analysis, is very hard to follow. It includes some contradictions, uses of wrong words, some important misunderstandings, etc. I make several specific comments to this effect below. The figures associated with this section are in my point of view incomplete, which makes the appreciation of the results difficult. I also give suggestions below on how to improve them.*

**Reply to the major comment**: the authors thank the referee for the careful examination of the manuscript and detailed comments on Sec. 3.2. We have made specific replies to each comment, and the following revisions in general. First, we update the scaling analysis of Cumulative Probability (C-CDF in previous manuscript) to be more accurate, including a more formal introduction of the effective resolution. Second, the scaling analysis for multi-fractality and structure functions are now revised. Third, we include the analysis of other wintertime days for the sake of completeness and further analysis of contributing factors.

*Section 3.3 contains some intersting results on the effect of the convergence of the EVP model on the simulated dynamics.*

*Overall, the paper needs some proof-reading to improve the conciseness and accuracy of the formulations used. I also found many grammar mistakes but stopped raising them up at some point. I moreover found that sometimes, jargon-like formulations were used that unfortunately hide the real meaning of the sentences. An important point is the use of the term "multi-scale modeling" for what is really a comparison of model simulations across resolution. This crucially needs to be clarified.*

**Reply to the major comment**: the authors would make thorough check and proof reading on the grammatical usage after all revisions have been made. Specifically, the term of "multi-scale modeling" are replaced with "multi-resolution simulation (or modeling)" throughout the paper. Indeed, as raised by the referee, the approach is multi-resolution with parallel experiments at different grid resolutions, but not multi-scale within a single experiment. We also take care to differentiate the multi-scale deformation of the sea ice cover, and the multi-scale (or multi-resolution) experiments we have carried out.

*In brief, I consider that major revisions are required. In my point of view, the points I raise in the specific comments below need to be adressed in a first time. Another review of the paper should be conducted in a second time, in order to better appreciate the results, their meaning and their importance, in the context of this study and provide further suggestions on how to improve the manuscript it its gobality.*

**Reply to the major comment**: the authors thank the referee for these invaluable comments, and the replies (with explanations and revision to be made) are as follows for each specific comment from the referee.

*Page 1, title: I find the use of the term ''multi-scale modeling'' unfortunately misleading and inapropriate. The paper is effectively about comparision of model simulations performed at various spatial resolutions, while multi-scale modeling refers to codes that can effectively resolve processes occuring different space/time scales by coupling physically and numerically models of these specific processes. A more appropriate title would be "Comparison of sea ice kinematics at different spatial resolutions modeled with a hierarchy CESM..." or "Cross-resolution comparison of CESM sea ice simulations". Please also correct any mention to multi-scale modeling in the text for consistency.*

**Reply**: the authors acknowledge the comment on the use of term "multi-scale modeling". Therefore, we modify the title of the article to: "Comparison of Sea Ice Kinematics at Different Resolutions Modeled with a Grid Hierarchy in Community Earth System Model (version 1.2.1)"

*Abstract, line 2 : "Sea ice kinematics is the most prominent feature of high-resolution simulations." There is no need to use high-resolution for kinematics features to be prominent in sea ice simulations. Please sea my comment just below about alternative rheological models, which do resolve the signature of kinematic features at medium and low resolution (> 20 km, Rampal et al., 2019).*

**Reply**: we agree with the referee that with Lagrangian framework and MEB rheology, neXtSIM could model realistic sea ice deformation at lower nominal resolution (Rampal et al. 2019). We would argue, however, that the nominal resolution of the Lagrangian models are not the limitation in resolving multi-scale deformations of sea ice, which is a great advantage of the methodology as compared with Eulerian grid based ones. Therefore, the statement we made apply to these traditional models with non-moving grids. The text is revised accordingly, as follows: "In traditional Eulerian grid based models, sea ice kinematics is the most prominent feature of high-resolution simulations". The introduction to neXtSIM and related works, including references, are now moved to Sec. 1 (from the last section), to give a better background introduction.

*Abstract, line 3 : "such as Viscous Plastic" current models are able to reproduce multifractality and linear kinematic features". This is one of my major comments: please be carefull as to make the distinction between multi-fractality in space and in time, throughout the entire text.*

**Reply**: the authors have revised the sentence to be more precise. Since this sentence is to introduce the community's status quo, it is revised as follows: "… with rheology models such as Viscous Plastic (VP) and Maxwell Elasto-Brittle (MEB), sea ice models are able to reproduce multi-fractal sea ice deformation and linear kinematic features that are witnessed in high-resolution observational dataset".

*Abstract, line 4 : "we carry out multi-scale sea ice modeling". No, you carry out a comparison of simulations at different resolutions.*

**Reply**: revised as "we carry out modeling of sea ice with multiple grid resolutions"

*Abstract, lines 6-7 : "including multi-fractal deformation and scaling properties that are temporally changing". In the light of my other comments below, I would precise "multifractal deformation in space" and not put too much weight on the temporal part. Your abstract should highlight your strong results and the temporal aspect of the scaling analysis is not one.*

**Reply**: we revise the sentence to include spatial scaling only, as follows "… including multi-fractal spatial scaling of sea ice deformation that depends on atmospheric circulation pattern and forcings". The abstract is also revised to be more precise.

*Abstract, line 8 : "effective spatial resolution". This effective spatial resolution has not been defined and cannot be understood here. I believe you mean that the model can resolve kinematic features that are 6 or 7 times the width of a model's grid cell? If so, this should be explained clearly and in simple words (i.e., rewrite lines 8-9) in the abstract and redefine later (see my other comment below).*

**Reply**: we rewrite the sentence to be more clear as follows: "By using high-resolution runs as references, we evaluate the model's effective resolution with respect to the statistics of sea

ice kinematics. In specific, we find the spatial scale at which the PDF of the scaled sea ice deformation rate of low-resolution runs match that of high-resolution runs. This critical scale is treated as the effective resolution of the coarse resolution grid, which is estimated to be about 6 to 7 times of the grid's native resolution."

*Page 2, line 1 : "scale-invariance properties" Cite Marsan et al., 2004 there and Kwok et al., 2008 after "linear-kinematic features". Also, many other and more recent references can be added to Marsan et al., 2004 regarding scale-invariance, especially scale-invariance in time.*

**Reply**: as suggested by the referee, the order of references is modified, and we have added the references for scale invariance including Rampal et al. (2008) and Weiss and Dansereau (2017).

*Page 2, line 6 : "most popular". A more objective term would be "most widely used".*

**Reply**: revised as suggested by the referee.

*Page 2, line 7-8 : "a plastic medium for packed ice under shear and pressure". This formulation is vague and unfortunately not accurate: the VP model describes sea ice as undergoing plastic deformation for over-critical shearing and compressive stresses only. Please modify the sentence accordingly.*

**Reply**: the sentence is revised as follows "…, and sea ice undergoes plastic deformations over critical shearing and compressive stresses."

*Page 2, lines 15-19 : "In order to reproduce the observed properties of the sea ice kinematics, grids of 0.1 degree resolution or finer are usually required". This is true perhaps only in the VP or EVP rheology cases. The MEB rheology has the capability to localize deformation in space at the nominal grid cell scale, whatever the resolution of the grid (Dansereau et al., 2016, Rampal et al., 2019). Mention of this fact unfortunately come only in the conclusion, whereas an adequate literature review in your introduction should distinguish between the VP/EVP and other existing continuum rheologies (EB, MEB, Elastic-decohesive).*

**Reply**: the authors agree with the referee that the statements we have made apply to models with Eulerian grids. Lagrangian model (neXtSIM) with MEB rheology does not suffer from the resolution limitation of non-moving Eulerian grids. We have also moved the paragraph introducing Lagrangian and novel rheology models in Sec. 4 to Sec. 1.

*Page 2, line 20 : "multi-resolution sea ice modeling". I think that "we carry a comparison of sea ice model simulations at different spatial resolutions", or "cross-resolution comparison" would be clearer and more accurate.*

**Reply**: we revise the sentence as "… we carry out comparison of sea ice model simulations at different spatial resolutions with the coupled model …"

*Page 3, line 12 : "For the SP (…) For the NP". And the same for the lines below.*

**Reply**: revised.

*Page 4, line 4 : "a suite", a series?*

**Reply**: revised.

*Page 4, line 16 : This sentence is not clear: is there a repetition of "for TS015" there?*

**Reply**: removed the first "for TS015"

*Page 4, line 19 : "within the Arctic Ocean".*

**Reply**: revised

*Page 5, lines 2-3 : This sentence is unclear and a bit repetitive.*

**Reply**: the sentence is revised as "We choose shorter time steps for both thermodynamics and dynamics with respect to the resolutions of the grids (Tab. 2)"

*Page 5, line 4 : "a series", replace by "different subcycle numbers". Maybe rephrase as "We choose shorter thermodynamics and dynamics time steps for our higher resolution grids"?*

**Reply**: revised "…, different subcycle numbers are chosen for each grid". See also the revision above.

*Page 7, line 4 : "Potential compromises of using SOM". This needs to be rephrased, for instance as "Potential compromises pertaining to the use of SOM".*

**Reply**: revised as suggested by the referee.

*Page 7, line 6 : "in the Ocean Model Intercomparison Project".*

**Reply**: revised by adding "the", as suggested by the referee.

*Page 7, lines 8 to 12 : I understand here that you interpolate the same wind field onto your different (3) resolution grids. Does the interpolation ensures that the input (wind) energy is conserved accross resolutions? If not, this will impact your scaling results. I believe that a clear mention to this effect, in this paragraph, would be a valuable addition.*

**Reply**: for the interpolation between the atmosphere and ocean (or sea ice), we follow the standard protocol of CESM. In specific, interpolation of air-ocean (or ice) fluxes are always kept conservative. For the dynamic coupling (i.e., treatments of winds), high-order method (Patch-Recovery) is adopted. The text is revised accordingly.

*Page 7, lines 7-8 : Can you specify to which years corresponds the climatological annual cycle based on NCEP atm. reanalysis that you use? It would help understanding the ice coverage and thickness value that you obtain in your simulations at equilibrium (see my comment about these results just below).*

**Reply**: regarding the CORE2 NYF dataset, the authors make the following statements [details in Large & Yeager (2004)]. First, the fluxes are computed with 43-year NCEP reanalysis. Second, the synoptic signals are mainly from year 1995 of the reanalysis, with transition by the end of December carried out through interpolation with data in December of 1994. We consider this a limitation for future study, and will move to inter-annual forcings in further experiments.
Furthermore, as shown in Stewart et al. (2020) in which JRA55-do reanalysis is used to generate new NYF datasets, the specific year that is chosen greatly affects the model's equilibrium status for simulating Arctic sea ice (Fig. 18 of the reference).
In our opinion, the sensitivity of sea ice climatology to the NYF forcing should be studied further, with the reanalysis's behavior in the Arctic at least a focus point (instead of the current status quo).

*Page 7, line 18 : Can you perhaps spell NDTE?*

**Reply**: revised as "number of timesteps for elastic wave damping, or NDTE"

*Page 7, line 25 to page 8, line 4 : You mention here a minor overestimation in the sea ice extent (cover) in some parts of the Arctic and underestimations in others. What is the basis for this comparision? From figure 5, I understand it is satellite sea ice edge data (from NSIDC), but this should be clearly mentionned in the text as well. Also the year or period of this satellite data should be mentionned with the corresponding years on which the climatology used to force your model is based.*

**Reply**: we revise the text to include the description over the specific region with overestimation of sea ice (mainly during winter in marginal seas). We also add in the texts the years (1979-2000) for the definition of sea ice extent climatology from NSIDC data.

*Page 8, lines 4-5 : "consistent with existing sea ice thickness reconstructions by PIOMAS". Also, in the same line as my previous comment, please mention the year for these PIOMAS thickness reconstruction, or insert a figure. It seems to me that there is indeed a lot of ice*

*stocked into the Beaufort Sea and that such thick ice conditions (up to 5 meters and more than 4 meters over a wide region, in September!) have not been seen at least in the last decade.*

**Reply**: we add on Fig. 5.b the PIOMAS sea ice volume seasonal cycle (computed with 1979-2000 monthly means) and revise the figure caption and texts to include necessary description. This period (1979-2000) aligns with that for the observational sea ice extent data from NSIDC. They are used as climatological sea ice extent/volume, since we do not have exact match between the model's climatology with existing observations/renanlysis.

*Page 8, line 10 : Can you explain in a few words what is a warm start-up?*

**Reply**: by "warm start-up" we mean "starting up the high-resolution simulation with a spun-up status from low-resolution ones". Therefore we revise the sentence as "With the spun-up climatological status of TS015, the experiment with TS005 approaches equilibrium towards year 42."

*Page 8, line 15 : "a minor decrease" of what? Please specify "both sea ice extent and volume" or merge this part of the sentence within the next one.*

**Reply**: according to the referee's suggestion, we have merged the two sentences as one: "Similar to TS045 and TS015, the experiment with TS005 produces reasonable sea ice climatologies, but with a minor decrease in both sea ice coverage (mainly in summer) and sea ice volume (all season) with respect to TS045 and TS015."

*Page 8, line 30 : "two years' daily mean sea ice fields (...)". Rephrase, eg. "two years (41-42) of daily mean sea ice fields for all three TS grids".*

**Reply**: rephrased as "two years (41-42) of daily mean sea ice fields for all three TS grids".

*Page 8, lines 28–30 : This is one of my major comment/concern. In this paragraph, you mention computing the deformation invariants from the daily mean sea ice drift speeds. This is the time scale set throughout your scaling analysis of daily deformation rates. You do not however mention how deformation rate components (du/dx, dv/dy, du/dy, dv/dx) are calculated, in particular at space scales larger than that of the cells of your Eulerian grids (with Arawaka-B staggering). Because you use Eulerian grids, I am guessing that your are following a coarse-graining method such as the one used in Marsan et al., 2004, but what are the details of the method? Do you, for instance, define square boxes and use a contour integral calculation to estimate each of the deformation rate components? Or just sum the components over each box? Most importantly, in estimating deformation rates at a given space scale, do you effectively sum (i.e., average) the deformation rate components and then calculate the corresponding invariants at that scale or do you sum (i.e, average) the deformation invariants themselves over that space scale? Also, why do you choose the region outlined in Figure 7 for your scaling and PDF analyses especially? How do you deal with the presence of coasts? Do you to eliminate data within a margin of the coasts? How do you deal*

*with coarse graining boxes that might contain land regions? All of these details will most probably affect your scaling results and should be mentionned.*

**Reply**: the authors would like to clarify that the computation of scaled deformation rates are strictly following the coarse-graining algorithm of line integration for the specific scale (for our case a square region of the grid), as carried out in Marsan et al. (2014) and Rampal et al. (2019) (Sec. 3 of the reference). In specific, a region of a certain area (in our case, a number of adjacent cells that form a square), we compute its scale (L) as well as the deformation rates by computing line integration around its outer walls. The values of $u\_x$, $u\_y$, $v\_x$ and $v\_y$ are computed and then used to compute the deformation rates. We have added an appendix to specify the computations we have carried out in more detail.
The region we chose (as shown in Fig. 7) covers the majority of the Arctic basin, and due to its square shape, it facilitates the computation of aforementioned scaled deformation rates.
The authors agree with the referee that the treatments on the outer rim of the sea ice cover (including coastal regions) affects the analysis results. For our study, in the case of adjacency to coast, since we use Arakawa-B grids in CICE, we omit any T-points that have any single vertex on land.
In the revised version of the paper, we carry out the analysis without any points close to land. We confirm that there is slight change, but no qualitative difference in the new results.

*Section 3.2, pages 8-9 : I my point of view, readability would be improved if the second and third paragraphs were included after the first sentence of paragraph one of section 3.2. Then, after paragraph 6, you a paragraph or sub-section describing your method for the estimation of deformation rates at different space scales, and of the scaling exponents, is necessary.*

**Reply**: according to the suggestion of the referee, revisions are made to re-arrange the contents of these paragraphs. First, the characteristic of the forcing data and the choosing of the representative days are covered. Second, a general description of the modeled deformation of the two days. Third, the analysis of the PDF and scaling properties, including the methodology, consistency checks, scaling analysis results.

*Page 11, line 22 : "There is good agreement". I think it would be more accurate to say that the results are consistent accross spatial resolution, since the comparison here is not done on the basis of observational data.*

**Reply**: we revise the sentence as "The simulation results are consistent across the three grid resolutions …"

*Page 11, line 23 : "large shearing belts accross the basin". There are indeed large shearing belts and diffuse regions of shearing rates seen at all model resolutions. Are these diffuse shearing belts physical? How do they compare, for instance, to shearing rates fields inferred from RadarSAT data? To what process, physical or numerical, do you think they are related?*

**Reply**: the authors agree that the comparison and validation with observational dataset (such as SAR-based deformation) is important. But because we use a NYF dataset from CORE2 for the experiments, there is no direct correspondence of the modeled sea ice field to any observations. In this paper we mainly focus on comparing the simulation of various

resolutions with certain consistency among them, but we do acknowledge this a limitation of our current work. We are looking forward to work with IAF dataset that improve comparability with observations such as RGPS.

In our opinion, very large deformation structure (such as shearing belt) is an aggregate response of the sea ice cover to large-scale forcings. Especially, there is very large variability of the (sea ice) transpolar drift on the daily scale (i.e., within synoptic scale), regarding both direction and strength. There is a wide spatial scale involved surrounding the either side of the transpolar drift. Another possible mechanism is that low-pressure systems entering the Arctic, causing large-scale deformation events.

*Page 11, line 25 : "There is a clear".*

**Reply**: corrected.

*Page 11, line 26 : Please change "more well-defined" by "better defined".*

**Reply**: revised.

*Page 11, lines 31-33 : The last sentence of this paragraph is not clear. Please define clearly what you mean by "effective" resolution of the model. I guess it corresponds to the width of the simulated LKFs?*

**Reply**: the authors would like to clarify that the model's native resolution serves as the basis of the simulation of certain phenomena (such as waves in geofluid dynamics or sea ice kinematic features), but these phenomena are not realistic on the spatial scales of the native resolution. Rather, the "effective resolution" of the model, which is usually coarser than the native resolution, is the spatial scale on which the model could realistically produce these realistic phenomena. Here we do not focus on the widths of the LKFs, but rather the shape of the modeled LKFs do not agree well with observations, in which the both ends of the (relatively) larger LKFs should feature even smaller LKFs. On the contrary, the ends of the many modeled LKFs end in a larger region with a spread-out, smaller deformation rates, with no clear structure.

*Page 11, line 34 : "the distribution of total deformation rates follow power-law distributions". What distribution? Please be precise here, e.g., "the statistical (or probability density) function of total deformation rates follow a power law".*

**Reply**: revised as "… the probability density function of total deformation rates follows a power law distribution"

*Page 12, line 1 : I find the accronym C-CFD to be confusing here. You are calculating the cumulative probability density function of both daily and 3-days mean deformation rates. This term and the accronym cumulative PDF is used in most published scaling analyses within the sea ice community. I suggest for clarity that you use similar terms.*

**Reply**: the authors would like to clarify that we use complementary cumulative distribution function (or C-CDF) for the cumulative PDF (which is more widely used by the sea ice community, as mentioned by the referee). In response to the comment, we replace the use of C-CDF with Cumulative PDF or Cumulative Probability as revisions.

*Page 12, line 3 : "For both daily and 3-days cumulative distributions, we attain multifractality accross the three resolutions". I am confused here: how do you conclude that deformation rates are multifractal from the cumulative distributions in Figure 9? This information is rather given by a scaling analysis based on different moments of the distribution of deformation rates and the estimation of the convexity of the quadratic function describing the dependance of the scaling exponents on the moment. Please clarify or remove.*

**Reply**: the authors remove this statement of multi-fractality, since this is actually the result drawn from the scaling analysis later in this sub-section.

*Page 12, lines 7-21 : This paragraph is very confusing and I do not understand your method here. First you say that you carry out the spatial scaling of one grid onto another? How do you do that? Do you mean that you interpolate deformation rates from one grid to another? Line 7 : you mention that the slopes become steeper for the higher resolution grids for one given day but not the other. I suspect you mean the slope of the "CFDs/C-CDFs" in log-log space? How do you compute these slopes from figure 9? Can you please show these slopes on the figure so that one can evaluate at least qualitatively the goodness of fit? Lines 9-10 : "the slopes of C-CFDs from scaled rates". Do you mean interpolated doformation rates? Also, putting all of the curves on each panel of figure 9 makes it very difficult to read the figures. I would suggest separating the "non-scaled" or non-interpolated and interpolated results on different figures, or use different levels of opacity for the non-interpolated and interpolated results. Lines 13-14 : what is a realistic shape for the distribution? On what data do you base your evalution of a "realistic shape"? Also, a realistic shape for a "power-law distribution" is by definition a power law! Hence I suggest you write simply "a realistic shape of the distribution of deformation rates". Also see my previous comment about defining the "effective" resolution of the model. Line 16 : "we attain the same slope" Lines 18-19 : "the CFD of sea ice deformation rates", not kinematics. Also, please explain how you evalute this effective resolution, which is 6-7 times higher than that of the TS015 grid and what are the different days that you are analyzing.*

**Reply**: the authors apologize for the lack of clarity of this paragraph. The central role of the analysis is to compute and compare the scaled C-CDFs (or cumulative probability) from different grids. For example, we can scale the model output from TS005 to the grid resolution of TS015, and directly compare the C-CDF with that of TS015. Another example is that we scale the model output from TS005 to the resolution of TS045, and scale the model output of TS015 to the resolution of TS045. Then, we can compare the tail slopes of the Cumulative Probability. Suppose that we want to evaluate the effective resolution of TS015, by using the results from TS005 as a reference, we can find the spatial scale at which the slopes of scaled C-CDF from TS015 matches that of TS015. This spatial scale is then viewed as the effective resolution of the model output of TS015.

This analysis inherently relies on high-res. (TS005) model outputs as references. Also, we compute the slope as the slope of the linear least-square fitting between 0.05 and 0.25 for the

cumulative probability (y-axis). We have re-written the whole paragraph to increase clarity. Besides, Fig. 9 is also revised to include necessary information of statistical fittings.

*Page 12, lines 22-26 : Why do you think the (absolute) slopes you are estimating are smaller than that of Marsan et al. 2004 at a similar time scale? It would be relevant to offer possible explanations here.*

**Reply**: the authors would like to clarify that the absolute values of slopes we have in the analysis (-1.6 to -2.7) are indeed smaller than that of Marsan et al. (2004), which is -2.5 at the spatial scale of 13~20km. This is only to cite these values, since there is inherently no comparability among them (different date, different time duration, models not forced with realistic atmospheric forcings). We conjecture that this value of slope is time-variant and changes with forcing and ice conditions, but this could be investigated in the future.

*Page 12, line 26 : "to evaluate the structure function".*

**Reply**: revised (on line 28).

*Page 12, line 31 : "At q = 3, the structure function is in the range...". Do you mean beta instead of the structure function?*

**Reply**: corrected to be beta(q), according to the referee's suggestion.

*Page 12, last paragraph : I do not think it is relevant to cite the differences in the values of beta(q) or in the shape of the structure functions between the two analyzed days if you do not try to explain these differences physically.*

**Reply**: in the revised version the authors extend the scaling analysis of structure function to the potential contributing factors including the atmospheric forcings.

*Page 12, line 35 : "the average deformation rate". It would be more specific to refer to the mean deformation rate or to the moment of order 1.*

**Reply**: revised to "the mean deformation rate".

*Figure 10 : On the scaling figures (left panels) please indicate the moment order corresponding to each set of curves and insert a legend for the different colors/model resolutions. Also, you label the y-axis with epsilon for the total deformation rate, whereas in equations (1) to (3) you use dot(varepsilon) (indeed not available in MATLAB) for this variable and the other deformation invariants. Please use consistant symbols accross the text and figures. On the x-axis of the same figure, you use the label "space scaling", which would rather be appropriate as a title for these figures. I believe you mean "space scale". Your structure function is estimated using the moment of order 0.5, but it is not shown in the scaling analyses (left panels), why? Your structure function results could also be appreciated*

*more objectively if you included error bars for beta for each of the moments (see e.g., Rampal et al, 2019 for the definition of the error bars on beta(q)).*

**Reply**: the authors apologize for the missing of legend for different (grid) resolutions. Actually the starting points of different resolutions differ, since only TS005 (purple) reaches down to the spatial scale of 2km, and TS015 (red) reaches down to 7km. For the sake of clarity, we add a legend to these panels and corresponding texts in the figure caption. Symbols for deformation rates are also revised to be consistent across the article, as suggested by the referee.
"Space scaling" are replaced with "Spatial scale" in the figures. We have also included the moment-order of 0.5 in the scaling analyses. Besides, error bars are added for the panels for structure functions, by using formulations in Rampal et al. (2019).

*Most importantly, for the December case in particular it is apparent from the scaling figures (left panels) that the slope (beta) of the moments of order 2 and 3 is calculated by leaving out at least the two last points of the scaling analysis, corresponding to the largest space scales. Why is that and how considering/rejecting these points affects your results? If some data points are left out or attributed less weigth in the analyses, this should be definitely be clearly mentionned and argued for in the text.*

**Reply**: the authors thank the referee for the careful examination of the figures. Indeed at very large spatial scales, we are potentially hit with the problem of lacking samples. Actually across all the spatial scales, the effective sample counts change (or decrease) dramatically. The confidence level on the mean value at these largest scales are much wider, as a result. We further examine the results in the revised version of the manuscript with the exclusion of the scales larger than 200km, which corresponds to the last points mentioned by the referee.

*Page 13, line 4 : Please specify that your result support multifractality of the simulated sea ice deformation in space (i.e, not in time), hence not "multifractality of sea ice kinematics".*

**Reply**: revised to be more precise by only mentioning spatial scaling.

*Page 13, line 5 : How does the inclusion of the deformation rates at the two larger space scales for the moments 2 and 3 change the value of the estimated curvature? (see my previous comment on figure 9).*

**Reply**: We further examine the results in the revised version of the manuscript with the exclusion of the scales larger than 200km, which corresponds to the last points mentioned by the referee (also replied above).

*Page 13, line 6-8 : This sentence is a generic comment and does not offer a satisfactory explaination for the difference in results between the two time period analyzed. Either offer some physical hypotheses or refrain from comparing the two results.*

**Reply**: regarding to this comment, we extend the analysis to daily deformation fields during the winter and carry out attribution study by relating the scaling to circulation pattern and forcing data.

*Page 13, line 8-9 : If there is a slight drop between the values of beta or the curvature of the structure functions between the daily and 3-days deformation rates and this difference is not evident for the Feb. 6 case, my opinion is that this is no sufficient evidence that the model can reproduce temporal scaling. Such assumption should rather be based on a proper temporal scaling analysis that spans several orders of time scales, not a comparision between 1 and 3-days fields. This comment in my point of view should be removed because not supported from your results (and mention to it should be removed from the abstract as well, which should only state your strong results).*

**Reply**: the authors agree with the referee that the method (based on Eulerian means) and the ensuing analysis do not strictly correspond to temporal scaling analysis. We have revised it to include the results above, but removed related statements on temporal scaling.

*Page 13, line 12 : "existing studies with observational datasets and modeling results". Please put some references here.*

**Reply**: references are added, as suggested by the referee, including Marsan et al. (2004) and Rampal et al. (2019).

*Page 13, line 14-15 : Why is there a comparison here? This does not make sense. Do you mean "more convex on Feb. 6 than Dec. 20"? Also, why would this support less dominant large-scale features on Feb. 6? Please clarify.*

**Reply**: the authors have revised the manuscript to remove this statement. Attribution to the various circulation pattern and forcing data are made in relevant part of the paper instead.

*Page 13, line 15 : What is a ''more effective temporal scaling"? How does the results on figure 9 and 10 support temporal scaling?*

**Reply**: (according to previous comment and reply related to temporal scaling) during revision we remove statements involving temporal scaling.

*Page 13, lines 16-24 : I do not understand the link between these sentences and the previous sentences of this paragraph. What point are you trying to make? Please explain. Also, I think it would be more accurate not to refer to Lindsay et al., 2003 as current RGPS observations. RGPS is currently not running. Maybe just drop the reference.*

**Reply**: as noted by the referee, there is missing link in the logic linking the analysis of typical days to atmospheric forcings. We make the following revisions that potentially improve the inherent logic in the argument. We extend the analysis to winter time daily deformation

fields, and carry out attribution study of the deformation statistics (scaling, PDF, etc.) to various factors including circulation and forcings.

*Page 13, line 26 : "we evaluate the sensitivity? of the modeled kinematics".*

**Reply**: revised as "… we evaluate the sensitivity of the modeled sea ice kinematics to the EVP subcycling …"

*Page 13, line 27 : "the probability density function (PDF)".*

**Reply**: corrected.

*Page 13, line 28 : and figure 11 : I suggest adding the estimated slope of the tail of each PDF on the graphs of figure 11 to illustrate how you estimate it from your results. It would also help putting a Gaussian distribution on each graph to visually identify the fat tails of the PDFs.*

**Reply**: as suggested by the referee, we add the estimation of slopes from the fittings, the range (on the y-axis) that is used to estimate the slopes, as well as the theoretical slope on Fig. 11.

*Page 13, line 28 : "for the total deformation rate".*

**Reply**: corrected.

*Page 13, Section 3.3, 1st paragraph : To support your claim that insufficient convergence of the model solution (i.e., nb of subcycles) is responsible for the absence of convergence between the PDFs of the simulated distribution rates at high resolution, I think you could also mention that the tail of the log-log PDF at the higher resolution and lowest number of sub-iteration does not seem linear as in the other cases.*

**Reply**: according to the referee's comment, we add another 2 sentences in the first paragraph of Sec. 3.3 (specifically, on line 1 of page 14), as follows "Besides, the PDFs of high-resolution runs (TS015 and TS005) show better tail structure, and furthermore, the slopes are also better characterized (i.e., closer linear fittings at -3) with larger values of NDTE. For TS045, the tail of the PDF suffers from the lack of samples for large deformation events."

*Page 14, line 6 : I think you mean figure 12, not 7.*

**Reply**: corrected.

*Page 14, line 8 : I think you mean figure 11, not 12.*

**Reply**: corrected

*Page 14, line 8 : What is the "physical" deformation rate? Please explain more clearly.*

**Reply**: the authors would like to clarify that by "physical deformation rate" we are referring to the inherent deformation when the EVP convergence is attained. Actually for regions outside LKFs, the deformation rate drops (without the noise), and the structure of LKFs becomes more defined. Therefore this sentence is less accurate, and we revise it as follows "With larger values of NDTE, the noise level decreases and the deformation rate around the linear kinematic features becomes smaller. As a result, a convergent PDF and linear feature maps are attained."

*Page 14, lines 10-12 : Instead of convergence of the kinematics, it is the convergence of the model solution, or simulated kinematics. Also, instead of "deterioration of simulation speed", I would write "increase in simulation time or cost" for more clarity.*

**Reply**: revised as indicated by the referee.

*Page 15, line 1 : "in the Canadian Arctic Archipelago". Line 3, "the CAA" again. Same mistake in other places on page 18.*

**Reply**: adding missing word of "the" for this case and all other cases.

*Page 15, line 2 : "an ice arch"*

**Reply**: corrected.

*Page 15, lines 8-10 : It would be helpful for the reader if you could further explain how you can separate these contributions in your model, by showing the term of the the dynamics/thermodynamics equation associated with each of them.*

**Reply**: according to the suggestion from the referee, this paragraph is extended to include a formal description of the different contributing terms to sea ice thickness, as well as the details of how to compute in the model (we adopted CICE).

*Page 18, line 2 : "due to the fact that the thermodynamic growth".*

**Reply**: revised according to the referee's suggestion

*Figure 13 : Please increase the fonts of the figure titles and colorbars.*

**Reply**: this figure is modified to improve readability.

*Page 22, line 1 : Please rephrase that sentence, which is not clear at all, both in the meaning and construction.*

**Reply**: this sentence is revised as: "In this paper we carried out sea ice simulations with a multi-resolution framework with Community Earth System Model."

*Page 22, line 2 : Change "grid stepping" for "grid resolution".*

**Reply**: revised according to the referee's suggestion.

*Page 22, lines 6-8 : "Multi-fractal sea ice deformation is accurately modeled by all three resolutions": please see my comment above on the inclusion or not of the all the points in your scaling analysis for the case of Dec 20. "with good agreement with observational works in terms of scaling properties" : have you try to compare the slopes of the scaling analyses (beta) for the three moments and the curvature of the structure functions with observational analyses at equivalent spatial resolution, e.g., based on RGPS data? If not, this comment should be revised.*

**Reply**: the authors revise this sentence according to the referee's suggestion, as follows "As shown in the spatial scaling analysis on the representative days, multi-fractal sea ice deformation is accurately modeled by all three resolutions."

*Page 22, line 12 : "multi-scale modeling studies". Please consider my previous comment on the meaning of "multi-scale modeling".*

**Reply**: we correct this case, along with all other cases, of "multi-scale modeling" to the more precise statement of "multi-resolution simulations"

*Page 22, line 19-21 : This sentence is incomprehensible, what is the "initial study with temporal scaling analysis with 3-day mean drift fields"? I did not see these results. If you are referring to the comparison of the daily mean and 3-days mean results, see my previous comment about how a much larger range of timescales would be necessary to conduct a meaningful temporal analysis.*

**Reply**: the authors agree with the referee's comment on temporal analysis. We remove the statements of referring the analysis of 3-day mean drift as temporal scaling.

*Page 22, line 21 : Repetition of "in our study".*

**Reply**: removed the second instance of "in our study"

*Page 22, lines 23-27 : Scaling analyses of modeled deformation fields and their comparison with equivalent analyses of observed deformation fields date from around 2010. Such*

*analyses and model-observation comparisons have been made by only a few sea ice research groups and besides, techniques for comparing accurately Lagrangian/Eulerian model outputs to observational deformation data, especially in the context of time scaling analyses, are complex and have been recently developped and applied. Hence I would not qualify scaling analysis as a "traditional" tool for evaluating sea ice kinematics. For further validating sea ice deformation properties simulated with your multi-resolution framework, I also suggest a comparison of simulations to observations of sea ice deformation.*

**Reply**: the authors agree that the scaling analysis involving modeled sea ice kinematics, including validation with observed high-resolution kinematics is a recent development in the community. Therefore we remove the term "traditional" from our statements. We also look forward to carrying out historical simulations with the multi-resolution grid system and coupling with a fully dynamic ocean component. We plan to start this work by using inter-annual forcing (IAF) dataset and comparing with historical observational dataset.

*Page 22, lines 31 to 33 : It is clearly stated in the paper by Rampal et al., 2019 that the MEB rheology of Dansereau et al., 2016 is used, not the EB rheology of Girard et al., 2011. Please read the paper and correct your sentence. Also, replace "which are shown" by "which is shown" and for a demonstration of the MEB model capability to localize deformation at the nominal grid cell scale, whatever the grid resolution, which explains the fact that neXtSIM does not encounter "effective" resolution issues and does not require using a sub-LFK spatial resolution to simulate adequately these features, therfore reproducing the scaling properties accross model resolutions, see Dansereau et al., 2016.*

**Reply**: the authors apologize for the imprecise statement of the rheological model in neXtSIM. Revisions have been made for both corrections to Maxwell Elasto-Brittle (MEB) rheology in neXtSIM and the addition of reference (Dansereau et al., 2016). Other errors including grammatical ones are also corrected.

*Page 22, line 6 : It is hard from the figures shown here only to witness the asymptotic convergence of the simulated modeled kinematics. Please remove that sentence or include a figure that shows this clearly.*

**Reply**: as suggested by the referee, we revise this sentence to be more precise: "Based on a traditional implementation of EVP in our model, we have witnessed asymptotic converging behavior of sea ice kinematics fields with increased EVP subcycle count."

*Page 23, line 6 : "we have witnessed that".*

**Reply**: corrected.

*Page 23, line 15 : What is your choice of ice strength parameterization scheme? See my previous comment about the importance of including at least a reference to the dynamic equations of the model and a table listing the EVP rheology parameter values.*

**Reply**: the ice strength parameterization scheme is based on Rothrock (1975) with details of implementation in Lipscomb et al., (2007). This scheme is used by CESM in its scientifically validated experiments. We have now included a new appendix to include a complete list of the specific parameterization schemes and parameters that are used in the experiments.

*Page 23, line 29 : "Multi-scale simulations". I belive you mean that the comparison of simulations across spacial resolutions is becoming common in the climate modeling community.*

**Reply**: revised as "Simulation with multiple spatial resolutions".

**References:**

Dansereau, V., Weiss, J., Saramito, P., and Lattes, P. (2016). A Maxwell elasto-brittle rheology for sea ice modelling, The Cryosphere, 10, 1339–1359, doi:10.5194/tc-10-1339-2016

Large, W., and Yeager, S. (2004). Diurnal to decadal global forcing for ocean and sea-ice models: the data sets and flux climatologies. NCAR Technical Note: NCAR/TN460+STR. CGD Division of the National Center for Atmospheric Research.

Lipscomb, W.H., E. C. Hunke, W. Maslowski, and J. Jakacki (2007). Improving ridging schemes for high-resolution sea ice models. J. Geophys. Res.–Oceans, 112:C03S91, doi:10.1029/2005JC003355

Marsan, D., Stern, H., Lindsay, R., and Weiss, J. (2004). Scale Dependence and Localization of the Deformation of Arctic Sea Ice, Phys. Rev. Lett., 93, 178 501, doi:10.1103/PhysRevLett.93.178501.

Rampal, P., Dansereau, V., Olason, E., Bouillon, S., Williams, T., Korosov, A., and Samaké, A. (2019). On the multi-fractal scaling properties of sea ice deformation, The Cryosphere, 13, 2457–2474, doi:10.5194/tc-13-2457-2019.

Rampal, P., J. Weiss, D. Marsan, R. Lindsay, and H. Stern (2008), Scaling properties of sea ice deformation from buoy dispersion analysis, J. Geophys. Res., 113, C03002, doi:10.1029/2007JC004143.

Rothrock, D.A. (1975). The energetics of the plastic deformation of pack ice by ridging. J. Geophys. Res., 80:4514–4519

Stewart, K.D., W.M. Kim, S. Urakawa, A.McC. Hogg, S. Yeager, H. Tsujino, H. Nakano, A.E. Kiss, and G. Danabasoglu (2020). JRA55-do-based repeat year forcing datasets for driving ocean–sea-ice models, Ocean Modelling, 147, 111557

Weiss, J. and Dansereau, V. (2017). Linking scales in sea ice mechanics. Phil.Trans. R.Soc.A 375: 20150352. http://dx.doi.org/10.1098/rsta.2015.0352

---

## Author Comment (AC3) · 10 Sep 2020

The authors would like to thank Dr. Nils Hutter (the referee) for the invaluable comments. The following are the replies for each comment, together with specific revisions that are made. The original comments are in *green italic* font, and the revisions are highlighted in the revised manuscript in yellow.

**Reply to comments of Referee #1:**

The paper describes the creation of a multi-resolution suite of grids for CESM with a focus on their use for sea-ice modelling in the Arctic. The authors study the effect of grid resolution and number of EVP subcycling steps on the statististical properties of sea ice deformation as well as sea ice extent and volume. In particular, the localization of sea ice deformation in shear and failure lines such as leads and pressure ridges is studied. The authors present their model configuration as a starting point for more dedicated studies on sea ice dynamics and climate simulations and share the corresponding code and data. The simulations are analysed without optimising model parameters to the specific grid resolution and the evaluation of the simulations need to be improved. Therefore, I recommend the manuscript for publication in Geoscientific Model Development after consideration of my general and specific comments.

**General comments:**

1) The authors present untuned model runs with biased ice thickness fields (with too thick ice in the Beaufort Gyre) and also the ice volume in the simulations differs with the resolution used. However, the authors describe good agreement of sea ice coverage and volume of all simulations although a comparison with a sea ice thickness product, e.g. PIOMAS, is missing. So first, I suggest a thorough evaluation of the ice thickness and to study the differences in sea ice state between the different resolution simulations. I see two potential ways how to handle these different resolutions simulations that produce different sea ice state:

Reply to the general comment (1): the authors thank the referee's comment on the modeled sea ice thickness, and would like to make the following reply. Based on extra numerical experiments, we have discovered sensitivity of modeled ice thickness to the strength parameterization scheme. By default, CESM utilizes an ice strength parameterization in Rothrock (1979), detailed in Lipscomb et al. (2007). Instead of the traditional scheme in Hibler (1975) in which ice strength is related to mean ice thickness, the ice strength is closely related to the energy conversion and dissipation during the ridging process. Fig. 1 (below) compares the equilibrium Arctic sea ice thickness with the two ice strength schemes for TS045 (CESM D-type). With the scheme of Hibler (1975), the sea ice is considerable thinner in the Beaufort Gyre (BG), and arguable more reasonable with respect to observed sea ice climatology. We further confirm that thinner ice in BG is independent of the specific grid we use, showing similar differences for the default built-in grid of GX1V6 in CESM (Fig. 2). We have also found similar results with TS015 (CESM D-type experiment) as well as CESM Gtype experiment (ocean-ice coupled run) with TS045 (results not shown). Since Rothrock (1979) and Lipscomb et al. (2007) are used by CESM, including its scientifically validated experiments, we consider this choice of ice strength parameterization reasonable for the experiments and analysis within the realm of this study. Furthermore, in a recent paper (Stewart et al., 2020), ocean-ice coupled experiments with both CORE2 NYF and normal-year based forcings from JRA55-do are compared. Fig. 18 of

the reference confirms our findings above, showing thick ice in BG for CESM (version 2), as well as drastically different ice thickness fields modeled when different years of JRA55-do are used for normal-year forcings. This indicates that NYF based experiments that produce reasonable sea ice distribution can serve the study of certain aspects of model performance, but not necessary representation of the mean status of sea ice in reality.

In summary, we consider the model output attained with CORE2 NYF in our experiments reasonable for the comparative study across the resolutions. Furthermore, we look forward to exploring the attribution of strength scheme dependent sea ice thickness in the future.

Fig. 1. March sea ice thickness difference for TS045 grid at equilibrium states under CESM D-type experiments. Left: Rothrock (1975) scheme; Middle: Hibler (1979) scheme; Right: difference (middle *minus* left). All values are in meters.

Fig. 2. March sea ice thickness difference for GX1V6 grid at equilibrium states under CESM D-type experiments. Left: Rothrock (1975) scheme; Middle: Hibler (1979) scheme; Right: difference (middle *minus* left). All values are in meters.

(a) If you are interested to study the effect of different resolution on sea ice dynamics (which is the topic of Section 3.3), all simulations should produce comparable sea ice distributions (concentration fields thickness as well volume and extent). Otherwise it is not possible to disentangle the effect of the change in resolution and the change in sea ice state on the dynamics. Systematic tuning methods (Massonnet et al., 2014; Ungermann et al., 2017; Sumata et al., 2019) could be used for all three simulations to optimize the parameter choices for each simulations by minimizing the model-observations misfit (for

instance concentration, thickness, and drift). To resolve the issue of too thick ice in the Beaufort Gyre, drag coefficients and the ice strength parameterization could be tuned. The tuned simulations are then a good starting point for further multi-resolution studies and also the various parameters determined in the optimization will provide insight in how model physics change with resolution.

**Reply to the general comment (1.a)**: the authors fully agree with the referee that it is necessary to tune the model in order to improve modeled sea ice distribution and comparability. In our study, our original intention is to align the parameterization across the resolution to ensure good comparability, at least the thermodynamics should be kept the same across the resolution range. As indicated by the referee, we have found sea ice thickness in BG greatly dependent on the ice strength parameterization scheme, as shown above. Although the scheme we have adopted is used by default in CESM, we consider it to be replaced with the more commonly used scheme in Hibler (1979) in future studies.

(b) If such systematic tuning is not possible due to limited computational resources, the authors should be more cautions with statements regarding the good agreement of ice thickness fields and agreement of all three simulations. The differences between the three simulations should be described and interpreted in details. Possible reasons for the different sea ice distributions should be provided along with guidance what limitations with regard to the kinematic studies originate from the different sea ice distributions.

**Reply to the general comment (1.b)**: according to the referee's suggestions, we further compare the Arctic sea ice climatology among the 3 grids. However, based on experiments that evaluate the effects of sea ice strength parameterization (shown above), we argue that ice strength parameterization causes much more uncertainty than the differences among the 3 grid resolutions. Since we have limited computational resource for the model runs, we follow the criteria below for the new experiments and related analyses. We align the thermodynamic parameterization schemes as well as parameter values across the 3 resolutions. We examine in detail the ice strength parameterization (including its parameters) in the modeled climatology. Furthermore, we include a new appendix showing all the parameters that are utilized in our study.

2) The analysis of deformation rates in the manuscript is limited to two 3-day intervals. Since the scaling properties of sea-ice deformation are highly variable (Stern & Lindsay, 2009) and strongly impacted by atmospheric winds (Herman & Glowacki, 2010), limiting the analysis to such a short time interval does not allow robust conclusions on the model capability to simulate multi-fractal deformation rates. It can not be excluded that the two dates chosen for the analysis mainly highlight the imprint of the atmospheric forcing. Another problem with the too short interval are the CDF of deformation rates that do not show power-law tails due to strong fluctuations (although stated differently by the authors). I suggest to extend this analysis to at least one entire winter. This will reduce the impact of specific wind conditions, smoothen the CDFs, and allow a more robust interpretation of the presented results with regard to the models ability to simulated strongly localized deformation rates along leads and pressure ridges. In addition I recommend to remove all statements on temporal scaling based on these two 3-day intervals from this manuscript, as now temporal scaling analysis is performed by the authors. **Reply to the general comment (2)**: regarding to the referee's comment on the analysis of deformation fields, we have made the following two revisions. First, we extend the study to the full winter, including the study of C-CDF and scaling analysis. The discussion on these two representative days are further carried out on daily deformation fields during the winter months. Second, as pointed out by the referee, the temporal scaling has not been carried out in strict manner, and therefore we remove the statements involving temporal scaling. Specifically, we only present the 3-day results without relating them to (or drawing any conclusion on) temporal scaling properties.

3) The good agreement of deformation fields between the different resolutions surprised and impressed me. In your simulations only the degree of detail in deformation feature increases, but the general patterns agree across the different resolutions. Knowing that ice fracture is a chaotic process that is very sensitive to small variations in ice strength these results puzzles me, as I was expecting that the deformation fields diverge very fast due to the different deformation history. At high resolution, a deformation event which is associated with divergence reduces concentration and thickness, and thereby the ice strength, such that deformation is more likely to appear in the same spot again. This effect should not be so effective in coarse resolution simulations as the reduction in concentration and thickness is much smaller due to the size of the grid box. This different memory should cause different reactions to the same atmospheric forcing. Do you see a reduction in concentration and thickness along the simulated LKFs in all your simulations? Do you see reoccuring deformation lines in all simulations? Your results indicate rather that in general this described feedback is not so strong and that fracture is mainly driven or better prescribed by the forcing, which would be an interesting result. This aspect of your results is definitely worth more discussion in the paper and maybe some additional analysis.

**Reply to the general comment (3)**: regarding to the referee's comments, we in the first place were also a little bit surprised about the consistency of deformation features across the 3 vastly different resolutions. We conjecture that this is due to the specific experiment design of using NYF forcing, as well as the specific strength parameterization scheme we have adopted. We are yet to explore deeper and carry out attribution study of why the kinematic features agree and find the factors can break such consistency. Specifically, we have 2 aspects to look into. First, we want to explore the effect of inter-annually changing forcings on the system. A constantly alternating forcing may deviate the pathway of different resolutions in terms of spatial distribution of thick/thin ice, and may cause differences in the deformation events (or strengths). Also with the ice strength parameterization scheme of Hibler (1979), we also want to carry out similar analysis comparing the deformation fields. Regarding the two specific questions, first, we do witness concentration changes with LFKs if divergence is present. Second, we witness some cases of re-occurrence of the LKFs, but we haven't carried out systematic analysis yet.

Specific comments:

P1, Line 3, "multi-fractality": of what? Please add scaling of sea-ice deformation

Reply: revised as indicated by the referee.

P1 Line 19, "kilometer-scale" satellite observations: SAR images have a resolution in the range of tens of meters. The drift and deformation products derived from consecutive SAR images have a kilometer-scale resolution. Please be more specific.

**Reply**: according to the referee's comment, we have revised the sentence as follows: "... kilometer-scale sea ice drift and deformation estimates with Synthetic Aperture Radars ..."

P2 Line 1, "Linear kinematic features": You have not described what these linear kinematic features are. Please describe once what they are (failure and shear lines where deformation is localized).

**Reply**: revised by replacing "Linear kinematic features ..." as "Linear kinematic features, including local deformation regions of sea ice failures and shearing, ..."

P2 Line 7: In the VP framework, the transition between viscous and plastic deformation depends on the stress states and not the concentration. The concentration influences the stress states by scaling the ice strength, but there is no direct link as suggested by your description. Please clarify.

**Reply**: the sentence is revised to be more precise, as follows: "it describes the sea ice as a two dimensional continuum with nonlinear viscosity and plastic deformations under high stress conditions of compression and shear".

*P2 Line 14: CMI -> CMIP (here and elsewhere in the manuscript)*

**Reply**: replaced here and every use throughout the text.

P2 Line 15: This is true for VP/EVP models. For other rheologies that include memory of past deformation, as the Maxwell elasto-brittle rheology, also coarser grid resolution might produce similar deformation statistics.

**Reply**: we revise the sentence to be more precise, as follows "With VP rheology, the capability of sea ice models to resolve fine-scale deformations is inherently bounded by the resolution of the models' grid."

*P2 Line 18: The continuum assumption is part of all continuum sea-ice models regardless what rheology they use. Please consider not explicitly mentioning the rheology here.*

**Reply**: the sentence is revised by removing the specific rheology of VP, as follows "Although the continuum assumption of the sea ice cover does not necessarily hold at these resolutions, …"

*P2 Line 22, "main driver": It is not clear to me what you mean with main driver. Please clarify this sentence.*

**Reply**: the sentence is revised as "... adopted by various research groups in the world for climate studies."

*P4, Table1: Please be more specific with the grid descriptions in the "Notes: column, such that the table is understandable without reading the text. There is enough space for that.*

Reply: the last column (Notes) of the table is extended for a more understandable description.

P4 Line 21-22: at the grid location and 60 vertical layers,...

Reply: revised according to the suggestion.

**P4 Line 25: Please rewrite sentence.**

**Reply**: we reformulate the sentence as "Second, we configure the model according to the grid resolution, including the choice of parameterization schemes and related parameters that are used."

P6 Figure 3: Please think about using the same limits for the contour plots for both grids. This would make it easier to see the difference between them. The contour lines are also hardly visible, you might also want to use a brighter red instead.

**Reply**: the figure is revised to improve clarity and readability, according to the referee's suggestion.

*P8 Line 6-8: The thickness anomaly in Beaufort Gyre could also be caused by too weak ice and not properly tuned ice strength parameterization. The thick ice north of CAA and Greenland is then advected by the ice drift and accumulates within the Gyre.*

**Reply**: the authors have carried out experiments which show very large sensitivity of equilibrium sea ice thickness (including BG) with respect to the ice strength parameterization. By replacing the default scheme adopted by CESM with Hibler (1979), the anomalously thick ice in BG is now gone. Considering the fact that the default scheme adopted by CESM is widely used in many scientifically validated experiments, including OMIP and CMIPs, we plan to carry out detailed attribution study in the future on this issue.

*P8 Line 13, "With the warm start-up, the experiments with TS005 approaches equilibrium towards year 42.": Only for the extent, the volume is still decreasing. Please clarify.*

**Reply**: we further make analysis and attribution of the differences among the 3 grid resolutions. Before that, the experiments are also continued to allow TS005 to reach an equilibrium after year 45.

P8 Line 17, "The overall sea ice coverage and volume of TS005 is also in good agreement with satellite observations and PIOMAS dataset.": I would not describe the strong overestimation of sea ice extent in winter as a good agreement. In addition, I miss the comparison with the PIOMAS dataset in the figure. Please state where to find this comparison.

**Reply**: we revise the sentence as "In general, the overall sea ice coverage and volume as modeled with TS005 are consistent with satellite observations and PIOMAS dataset". In Fig. 5.b we also add the climatological seasonal cycle from PIOMAS dataset. The years of 1979-2000 are adopted, same as the SIE climatology from NSIDC data.

P8 Line 17-19: I do not understand why using the same parameterizations for all three grids is a reason for reasonable results. It is known that model parameters need to be adapted to different grid resolutions to show similar physics (e.g. Williams & Tremblay, 2018). Please clarify or rewrite.

**Reply**: regarding the comment, we would like to make the following clarifications. First, we expect the thermodynamic parameterization schemes, along with the parameter values, to be consistent across the resolutions. The dynamics across the resolutions definitely will cause differences in the distribution of ice (both local ridging and spatial re-distribution), but we do not want to compensate these errors (or uncertainties) with thermodynamics. Second, we fully agree that in order to quantitatively improve the model's performance, the modeler (or model user) should tune the model to available observations. However, since under NYF there is no exact sea ice climatology for us to match, we therefore focus on the consistency among the 3 resolutions as long as they attain reasonable results. For future studies we look forward to carrying out IAF based experiments, for which a full suite of model tuning to match observed sea ice historical changes is planned.

*P9 Line 8, "removed of seasonal cycle": -> and the seasonal cycle is removed*

Reply: revised.

**P10 Figure5, "satellite-observed": Please state which satellite product is used for this comparison**

**Reply**: the mean annual cycle of NSIDC SIE product of year 1979-2000 from SSMI/SMMR sensors is retrieved and used as climatological seasonal cycle. The figure caption is revised to include this information.

P10-11 AO index analysis It is not clear why this analysis is needed here. As the corresponding explanation is rather complex, please consider to remove them from manuscript for clarity.

**Reply**: regarding the comment on AO analysis, the authors want to clarify that the inclusion of AO indices is to ensure that the NYF dataset is not untypical in terms of wintertime atmospheric forcings. If prominent negative or positive AO is present in the NYF dataset, we would expect much different sea ice circulation and thickness distribution.

P11 Line 8: sybcycle count -> subcycles

Reply: revised as indicated.

*P11 Line 17-18: "The kinematic features with TS005 are richer and much narrower, such as the network of shearing in Beaufort Sea." Do you want to say that in TS005 more and finer features are simulated?*

**Reply**: yes, and we revise the sentence as "With TS005, the model simulates more and finer sea ice kinematic features".

P11 Line 35: The region for the analysis you have chosen is problematic as it mixes pack-ice regions with coastal regions. In coastal regions stable deformation features, like flaw lead, are found that show nearly constantly very high deformation rates, which impacts the presented CDFs. I suggest to use the entire Arctic Ocean as study region and filter all grid points that are closer than 150-200km to the coast as done in other scaling studies.

**Reply**: we agree with the referee's comment on the effect of coastal region on the PDF and scaling analysis. We therefore update the results by limiting the analysis within the basin, to the common regions according to TS045 (the coarsest resolution among the three).

P12 Line 3: Please be cautious for two reasons: (1) just because the PDF/CDF of sea ice deformation shows a power-law tail does not mean it is multi-fractal. To show mulit-fractality a scaling analysis of the moments of sea-ice deformation need to be performed that shows a non-linear convex structure function (you do this analysis but it is described later).
(2) The distributions shown in Figure 9 show hardly power-law distributions. I suggest to use the methodology of Clauset et al. (2009) to test for power-law distributions.

**Reply**: the authors fully agree with the referee. We have made a mistake that we have included the texts describing the spatial scaling results to the analysis of C-CDF. We have removed the description of multi-fractality from this part of the paragraph. Regarding the power-law distribution, since for the end of the C-CDF tail (which corresponds to relatively larger deformation events) we are hit with very small sample count, therefore, the determination of slopes is carried out for the range of 0.05 and 0.25 for the C-CDFs (noted in the figure caption).

*P12 Line 6: What do you mean with "spatial scaling"? Are you coarse-graining the high resolution simulation to coarser grid resolution? Please clarify.*

**Reply**: yes, and we revise the sentence as " ... we carry out: (1) the spatial coarsening of the model output of TS005 onto TS015 and TS045, and (2) that of TS015 onto TS045."

P12 Line 6-21: (1) The CDF in Figure 9 hardly show power-law tails and deviate from strait tails. It is not clear how you determine the power-law slopes. I recommend to use larger time intervals for the analysis to reduce the imprint of certain atmospheric forcing conditions and second to use the methodology presented in Clauset et al. (2009) to test for power-law distributions.

**Reply**: the authors have extended the analysis to winter months and limited the analysis to strictly within the basin (away from coast) to attenuate atmospheric noises and avoid potential problems on coastal regions. Details and results are presented separately in the revised Sec. 3.2.

**(2) It is unclear to me how you relate the slopes the CDF-tails of coarse-grained deformation rates to the nominal resolution of the grid. Please describe this concept more in detail.**

**Reply**: the authors have rewritten the paragraph to include a more formal introduction of the methodology we have adopted. In brief, we use the simulation results from TS005 as a reference, by spatial coarsening of the results with TS005 to a coarser grid such as TS045, we compute the difference between the tail slopes at the grid scale of TS015. As shown, the slopes are flatter for TS005, indicating that with TS015 the model cannot simulate reasonable sea ice deformation rates with respect to the reference of TS005. We consider this difference due to that the grid-cell scale deformation is actually not realistic for TS015, and therefore a coarser spatial scale can be determined with coarsening, when the slopes from TS015 finally matches that of TS005. This scale we consider is the effective resolution of TS015 for simulating sea ice deformations.

*P12 Line 22-26: Given the limitations of your analysis (short-time interval, no clear power) I do not recommend a direct comparison with observations or at least mention these limitations.*

**Reply**: according to the referee's comment, we revise the paragraph to avoid direct comparison with Marsan et al. (2004) or any observational dataset. Afterall, due to the use of NYF, there is no direct comparability of these data.

**P12 Line 32, "about 1.3 on Feb. 6th for all three grids": I see values from 1.2 to 1.3.**

**Reply**: according to the referee's comment, we revised the sentence as "and between 1.2 and 1.3 on Feb. 6th for all three grids."

P13 L1-2, "Furthermore, no positive value of  $\beta$  is detected at q = 0.5, which is consistent with Marsan et al. (2004) (Fig. 4 of the reference).": Please clarify. In Marsan et al. (2004) beta is positive for q=0.5. Also in your Figure 9 beta seems to be positive for q=0.5. What would be the physical interpretation of negative scaling exponents if you find them in your model?

**Reply**: we correct the mistake in citing the value of beta from Marsan et al (2004), by changing "positive" to "negative". For reported data with observations [such as Marsan et al. (2004)], the value of Beta is still positive for q=0.5. For the model to report negative values for Beta, statistical issues might be the cause. Since the usual practice of using linear fittings in the analysis in Fig. 10 (left side of each panel), we usually ignore the different statistical confidence on different scales. For larger spatial scales, the sample count is significantly smaller than small scales, given that the analysis is carried out on the same dataset. Uncertainty in estimating the mean deformation rate could play an important role for the estimation of Beta, causing statistically insignificant negative values for Beta.

**P13 Line 6-8: Please be more specific. Do you mean that with increasing resolution, deformation rates are more localized with yields to more pronounced scaling?**

**Reply**: regarding the referee's comment, we consider this description appli

---

## Author Response (AR1)

The authors would like to thank the 3 referee's comments and the editor's efforts in helping us to improve the manuscript. We have made replies to the comments and corresponding revisions. This general reply include the major and common revisions that are made to the manuscript. The replies to each referee's comments are attached after this document. Besides, a marked up version of the revised manuscript is also provided to highlight the specific revisions (yellow for the replies to the comments from referee #1, cyan for those from referee #2, and red for those from referee #3).

There are several major changes to the manuscript (among which many are also common questions raised by the referees):

- (1) In the revised manuscript, we have adopted the sea ice strength parameterization scheme in Hibler (1929), denoted H79, and re-run all the experiments. All the contents, including figures and texts, are updated accordingly. Although the scheme in Rothrock (1975), denoted R75, is used by default in CICE and CESM, we attain a more reasonable basin-scale sea ice thickness distribution with H79 under the NYF forcing dataset. Some detailed analysis of the sensitivity is in the reply to the referee comments. We consider the choice of strength parameterization an important and relevant issue, but beyond the scope of this study. We intend to explore it further in future studies.
- (2) We have changed the spatial and temporal coverage for the scaling analysis. First, we have changed the region of study to be within the basin but excluding area close to land. This avoids the inclusion of semi-permanent deformation regions, which is an issue raised by two referees. Second, we have added the results of wintertime spatial scaling with daily deformation fields. This content forms an added section (Sec. 3.3) in the manuscript. Due to the variety of the factors that contribute to the kinematics (including its statistics), the scaling coefficients cannot be simply attributed to AO. Rather, it remains an open question of how various factors (such as sea ice state, fine-scale atmospheric forcings) influence the spatial scaling, and furthermore, whether current models have the potential to reproduce related processes. In this regard, we present the wintertime scaling coefficients in Sec. 3.3 and add the planned work for attributing of various factors in discussion.
- (3) Model details, including the choice of model parameters, are now in an added appendix (Appendix B), as required by all 3 referees. We keep the same model parameters as used across the resolutions (0.45-deg, 0.15-deg and 0.05-deg), since we have attained very consistent climatology without extra model tuning. For example, the wintertime sea ice volume only differs by 5% among the three resolutions. We do consider model tuning a necessary step, especially when comparing against observations. However, due to the use of NYF forcing, we lack the exact observational dataset to compare against. We plan to carry out parameter-based tuning when moving to IAF and historical experiments in the future.
- (4) Spatial scaling methodology are covered in detail in a newly added appendix (Appendix C), which is required by the 3rd referee.
- (5) We restrain from the statements about temporal scaling. A Lagrangian tracking

based approach is needed for a formal examination of temporal scaling (Hutter, 2018; Weiss & Dansereau, 2017), the mean of Eulerian ice speeds is improper for such analysis. Therefore, we only show the results with 3-day mean ice fields without using temporal scaling. Temporal scaling, as well as spatial-temporal scaling can be carried out with our model output with a Lagrangian tracking based diagnosis in the future.

The replies to the 3 referees' comments are attached after this document (**page 3 to 15** for the reply to referee #1, **page 16 to 22** for those to referee #2, and **page 23 to 41** for those to referee #3). Again, we sincerely thank the referee's for the professional and helpful review to help us improve the contents of the manuscript.

**References:**

- Hibler, W.D. (1979). A dynamic thermodynamic sea ice model. J. Phys. Oceanogr., 9:817846.
- Hutter, N., Losch, M., and Menemenlis, D. (2018). Scaling properties of Arctic sea ice deformation in a high-resolution viscous-plastic sea ice model and in satellite observations, Journal of Geophysical Research: Oceans, 123, 672–687, doi:10.1002/2017JC013119.
- Rothrock, D.A. (1975). The energetics of the plastic deformation of pack ice by ridging. J. Geophys. Res., 80:4514–4519
- Weiss, J. and Dansereau, V. (2017). Linking scales in sea ice mechanics. Phil. Trans. R. Soc. A, 375: 20150352. http://dx.doi.org/10.1098/rsta.2015.0352

The authors would like to thank Dr. Nils Hutter (the referee) for the invaluable comments. The following are the replies for each comment, together with specific revisions that are made. The original comments are in *green italic* font, and the revisions are highlighted in the revised manuscript in yellow.

**Reply to comments of Referee #1:**

The paper describes the creation of a multi-resolution suite of grids for CESM with a focus on their use for sea-ice modelling in the Arctic. The authors study the effect of grid resolution and number of EVP subcycling steps on the statististical properties of sea ice deformation as well as sea ice extent and volume. In particular, the localization of sea ice deformation in shear and failure lines such as leads and pressure ridges is studied. The authors present their model configuration as a starting point for more dedicated studies on sea ice dynamics and climate simulations and share the corresponding code and data. The simulations are analysed without optimising model parameters to the specific grid resolution and the evaluation of the simulations need to be improved. Therefore, I recommend the manuscript for publication in Geoscientific Model Development after consideration of my general and specific comments.

**General comments:**

1) The authors present untuned model runs with biased ice thickness fields (with too thick ice in the Beaufort Gyre) and also the ice volume in the simulations differs with the resolution used. However, the authors describe good agreement of sea ice coverage and volume of all simulations although a comparison with a sea ice thickness product, e.g. PIOMAS, is missing. So first, I suggest a thorough evaluation of the ice thickness and to study the differences in sea ice state between the different resolution simulations. I see two potential ways how to handle these different resolutions simulations that produce different sea ice state:

Reply to the general comment (1): the authors thank the referee's comment on the modeled sea ice thickness, and would like to make the following reply. Based on extra numerical experiments, we have discovered sensitivity of modeled ice thickness to the strength parameterization scheme. By default, CESM utilizes an ice strength parameterization in Rothrock (1979), detailed in Lipscomb et al. (2007). Instead of the traditional scheme in Hibler (1975) in which ice strength is related to mean ice thickness, the ice strength is closely related to the energy conversion and dissipation during the ridging process. Fig. 1 (below) compares the equilibrium Arctic sea ice thickness with the two ice strength schemes for TS045 (CESM D-type). With the scheme of Hibler (1975), the sea ice is considerable thinner in the Beaufort Gyre (BG), and arguable more reasonable with respect to observed sea ice climatology. We further confirm that thinner ice in BG is independent of the specific grid we use, showing similar differences for the default built-in grid of GX1V6 in CESM (Fig. 2). We have also found similar results with TS015 (CESM D-type experiment) as well as CESM Gtype experiment (ocean-ice coupled run) with TS045 (results not shown). Since Rothrock (1979) and Lipscomb et al. (2007) are used by CESM, including its scientifically validated experiments, we consider this choice of ice strength parameterization reasonable for the experiments and analysis within the realm of this study. Furthermore, in a recent paper (Stewart et al., 2020), ocean-ice coupled experiments with both CORE2 NYF and normal-year based forcings from JRA55-do are compared. Fig. 18 of

the reference confirms our findings above, showing thick ice in BG for CESM (version 2), as well as drastically different ice thickness fields modeled when different years of JRA55-do are used for normal-year forcings. This indicates that NYF based experiments that produce reasonable sea ice distribution can serve the study of certain aspects of model performance, but not necessary representation of the mean status of sea ice in reality.

In summary, we consider the model output attained with CORE2 NYF in our experiments reasonable for the comparative study across the resolutions. Furthermore, we look forward to exploring the attribution of strength scheme dependent sea ice thickness in the future.

Fig. 1. March sea ice thickness difference for TS045 grid at equilibrium states under CESM D-type experiments. Left: Rothrock (1975) scheme; Middle: Hibler (1979) scheme; Right: difference (middle *minus* left). All values are in meters.

Fig. 2. March sea ice thickness difference for GX1V6 grid at equilibrium states under CESM D-type experiments. Left: Rothrock (1975) scheme; Middle: Hibler (1979) scheme; Right: difference (middle *minus* left). All values are in meters.

(a) If you are interested to study the effect of different resolution on sea ice dynamics (which is the topic of Section 3.3), all simulations should produce comparable sea ice distributions (concentration fields thickness as well volume and extent). Otherwise it is not possible to disentangle the effect of the change in resolution and the change in sea ice state on the dynamics. Systematic tuning methods (Massonnet et al., 2014; Ungermann et al., 2017; Sumata et al., 2019) could be used for all three simulations to optimize the parameter choices for each simulations by minimizing the model-observations misfit (for

instance concentration, thickness, and drift). To resolve the issue of too thick ice in the Beaufort Gyre, drag coefficients and the ice strength parameterization could be tuned. The tuned simulations are then a good starting point for further multi-resolution studies and also the various parameters determined in the optimization will provide insight in how model physics change with resolution.

**Reply to the general comment (1.a)**: the authors fully agree with the referee that it is necessary to tune the model in order to improve modeled sea ice distribution and comparability. In our study, our original intention is to align the parameterization across the resolution to ensure good comparability, at least the thermodynamics should be kept the same across the resolution range. As indicated by the referee, we have found sea ice thickness in BG greatly dependent on the ice strength parameterization scheme, as shown above. Although the scheme we have adopted is used by default in CESM, we consider it to be replaced with the more commonly used scheme in Hibler (1979) in future studies.

(b) If such systematic tuning is not possible due to limited computational resources, the authors should be more cautions with statements regarding the good agreement of ice thickness fields and agreement of all three simulations. The differences between the three simulations should be described and interpreted in details. Possible reasons for the different sea ice distributions should be provided along with guidance what limitations with regard to the kinematic studies originate from the different sea ice distributions.

**Reply to the general comment (1.b)**: according to the referee's suggestions, we further compare the Arctic sea ice climatology among the 3 grids. However, based on experiments that evaluate the effects of sea ice strength parameterization (shown above), we argue that ice strength parameterization causes much more uncertainty than the differences among the 3 grid resolutions. Since we have limited computational resource for the model runs, we follow the criteria below for the new experiments and related analyses. We align the thermodynamic parameterization schemes as well as parameter values across the 3 resolutions. We examine in detail the ice strength parameterization (including its parameters) in the modeled climatology. Furthermore, we include a new appendix showing all the parameters that are utilized in our study.

2) The analysis of deformation rates in the manuscript is limited to two 3-day intervals. Since the scaling properties of sea-ice deformation are highly variable (Stern & Lindsay, 2009) and strongly impacted by atmospheric winds (Herman & Glowacki, 2010), limiting the analysis to such a short time interval does not allow robust conclusions on the model capability to simulate multi-fractal deformation rates. It can not be excluded that the two dates chosen for the analysis mainly highlight the imprint of the atmospheric forcing. Another problem with the too short interval are the CDF of deformation rates that do not show power-law tails due to strong fluctuations (although stated differently by the authors). I suggest to extend this analysis to at least one entire winter. This will reduce the impact of specific wind conditions, smoothen the CDFs, and allow a more robust interpretation of the presented results with regard to the models ability to simulated strongly localized deformation rates along leads and pressure ridges. In addition I recommend to remove all statements on temporal scaling based on these two 3-day intervals from this manuscript, as now temporal scaling analysis is performed by the authors. **Reply to the general comment (2)**: regarding to the referee's comment on the analysis of deformation fields, we have made the following two revisions. First, we extend the study to the full winter, including the study of C-CDF and scaling analysis. The discussion on these two representative days are further carried out on daily deformation fields during the winter months. Second, as pointed out by the referee, the temporal scaling has not been carried out in strict manner, and therefore we remove the statements involving temporal scaling. Specifically, we only present the 3-day results without relating them to (or drawing any conclusion on) temporal scaling properties.

3) The good agreement of deformation fields between the different resolutions surprised and impressed me. In your simulations only the degree of detail in deformation feature increases, but the general patterns agree across the different resolutions. Knowing that ice fracture is a chaotic process that is very sensitive to small variations in ice strength these results puzzles me, as I was expecting that the deformation fields diverge very fast due to the different deformation history. At high resolution, a deformation event which is associated with divergence reduces concentration and thickness, and thereby the ice strength, such that deformation is more likely to appear in the same spot again. This effect should not be so effective in coarse resolution simulations as the reduction in concentration and thickness is much smaller due to the size of the grid box. This different memory should cause different reactions to the same atmospheric forcing. Do you see a reduction in concentration and thickness along the simulated LKFs in all your simulations? Do you see reoccuring deformation lines in all simulations? Your results indicate rather that in general this described feedback is not so strong and that fracture is mainly driven or better prescribed by the forcing, which would be an interesting result. This aspect of your results is definitely worth more discussion in the paper and maybe some additional analysis.

**Reply to the general comment (3)**: regarding to the referee's comments, we in the first place were also a little bit surprised about the consistency of deformation features across the 3 vastly different resolutions. We conjecture that this is due to the specific experiment design of using NYF forcing, as well as the specific strength parameterization scheme we have adopted. We are yet to explore deeper and carry out attribution study of why the kinematic features agree and find the factors can break such consistency. Specifically, we have 2 aspects to look into. First, we want to explore the effect of inter-annually changing forcings on the system. A constantly alternating forcing may deviate the pathway of different resolutions in terms of spatial distribution of thick/thin ice, and may cause differences in the deformation events (or strengths). Also with the ice strength parameterization scheme of Hibler (1979), we also want to carry out similar analysis comparing the deformation fields. Regarding the two specific questions, first, we do witness concentration changes with LFKs if divergence is present. Second, we witness some cases of re-occurrence of the LKFs, but we haven't carried out systematic analysis yet.

Specific comments:

P1, Line 3, "multi-fractality": of what? Please add scaling of sea-ice deformation

Reply: revised as indicated by the referee.

P1 Line 19, "kilometer-scale" satellite observations: SAR images have a resolution in the range of tens of meters. The drift and deformation products derived from consecutive SAR images have a kilometer-scale resolution. Please be more specific.

**Reply**: according to the referee's comment, we have revised the sentence as follows: "... kilometer-scale sea ice drift and deformation estimates with Synthetic Aperture Radars ..."

P2 Line 1, "Linear kinematic features": You have not described what these linear kinematic features are. Please describe once what they are (failure and shear lines where deformation is localized).

**Reply**: revised by replacing "Linear kinematic features ..." as "Linear kinematic features, including local deformation regions of sea ice failures and shearing, ..."

P2 Line 7: In the VP framework, the transition between viscous and plastic deformation depends on the stress states and not the concentration. The concentration influences the stress states by scaling the ice strength, but there is no direct link as suggested by your description. Please clarify.

**Reply**: the sentence is revised to be more precise, as follows: "it describes the sea ice as a two dimensional continuum with nonlinear viscosity and plastic deformations under high stress conditions of compression and shear".

*P2 Line 14: CMI -> CMIP (here and elsewhere in the manuscript)*

**Reply**: replaced here and every use throughout the text.

P2 Line 15: This is true for VP/EVP models. For other rheologies that include memory of past deformation, as the Maxwell elasto-brittle rheology, also coarser grid resolution might produce similar deformation statistics.

**Reply**: we revise the sentence to be more precise, as follows "With VP rheology, the capability of sea ice models to resolve fine-scale deformations is inherently bounded by the resolution of the models' grid."

*P2 Line 18: The continuum assumption is part of all continuum sea-ice models regardless what rheology they use. Please consider not explicitly mentioning the rheology here.*

**Reply**: the sentence is revised by removing the specific rheology of VP, as follows "Although the continuum assumption of the sea ice cover does not necessarily hold at these resolutions, …"

*P2 Line 22, "main driver": It is not clear to me what you mean with main driver. Please clarify this sentence.*

**Reply**: the sentence is revised as "... adopted by various research groups in the world for climate studies."

*P4, Table1: Please be more specific with the grid descriptions in the "Notes: column, such that the table is understandable without reading the text. There is enough space for that.*

Reply: the last column (Notes) of the table is extended for a more understandable description.

P4 Line 21-22: at the grid location and 60 vertical layers,...

Reply: revised according to the suggestion.

**P4 Line 25: Please rewrite sentence.**

**Reply**: we reformulate the sentence as "Second, we configure the model according to the grid resolution, including the choice of parameterization schemes and related parameters that are used."

P6 Figure 3: Please think about using the same limits for the contour plots for both grids. This would make it easier to see the difference between them. The contour lines are also hardly visible, you might also want to use a brighter red instead.

**Reply**: the figure is revised to improve clarity and readability, according to the referee's suggestion.

*P8 Line 6-8: The thickness anomaly in Beaufort Gyre could also be caused by too weak ice and not properly tuned ice strength parameterization. The thick ice north of CAA and Greenland is then advected by the ice drift and accumulates within the Gyre.*

**Reply**: the authors have carried out experiments which show very large sensitivity of equilibrium sea ice thickness (including BG) with respect to the ice strength parameterization. By replacing the default scheme adopted by CESM with Hibler (1979), the anomalously thick ice in BG is now gone. Considering the fact that the default scheme adopted by CESM is widely used in many scientifically validated experiments, including OMIP and CMIPs, we plan to carry out detailed attribution study in the future on this issue.

*P8 Line 13, "With the warm start-up, the experiments with TS005 approaches equilibrium towards year 42.": Only for the extent, the volume is still decreasing. Please clarify.*

**Reply**: we further make analysis and attribution of the differences among the 3 grid resolutions. Before that, the experiments are also continued to allow TS005 to reach an equilibrium after year 45.

P8 Line 17, "The overall sea ice coverage and volume of TS005 is also in good agreement with satellite observations and PIOMAS dataset.": I would not describe the strong overestimation of sea ice extent in winter as a good agreement. In addition, I miss the comparison with the PIOMAS dataset in the figure. Please state where to find this comparison.

**Reply**: we revise the sentence as "In general, the overall sea ice coverage and volume as modeled with TS005 are consistent with satellite observations and PIOMAS dataset". In Fig. 5.b we also add the climatological seasonal cycle from PIOMAS dataset. The years of 1979-2000 are adopted, same as the SIE climatology from NSIDC data.

P8 Line 17-19: I do not understand why using the same parameterizations for all three grids is a reason for reasonable results. It is known that model parameters need to be adapted to different grid resolutions to show similar physics (e.g. Williams & Tremblay, 2018). Please clarify or rewrite.

**Reply**: regarding the comment, we would like to make the following clarifications. First, we expect the thermodynamic parameterization schemes, along with the parameter values, to be consistent across the resolutions. The dynamics across the resolutions definitely will cause differences in the distribution of ice (both local ridging and spatial re-distribution), but we do not want to compensate these errors (or uncertainties) with thermodynamics. Second, we fully agree that in order to quantitatively improve the model's performance, the modeler (or model user) should tune the model to available observations. However, since under NYF there is no exact sea ice climatology for us to match, we therefore focus on the consistency among the 3 resolutions as long as they attain reasonable results. For future studies we look forward to carrying out IAF based experiments, for which a full suite of model tuning to match observed sea ice historical changes is planned.

*P9 Line 8, "removed of seasonal cycle": -> and the seasonal cycle is removed*

Reply: revised.

**P10 Figure5, "satellite-observed": Please state which satellite product is used for this comparison**

**Reply**: the mean annual cycle of NSIDC SIE product of year 1979-2000 from SSMI/SMMR sensors is retrieved and used as climatological seasonal cycle. The figure caption is revised to include this information.

P10-11 AO index analysis It is not clear why this analysis is needed here. As the corresponding explanation is rather complex, please consider to remove them from manuscript for clarity.

**Reply**: regarding the comment on AO analysis, the authors want to clarify that the inclusion of AO indices is to ensure that the NYF dataset is not untypical in terms of wintertime atmospheric forcings. If prominent negative or positive AO is present in the NYF dataset, we would expect much different sea ice circulation and thickness distribution.

P11 Line 8: sybcycle count -> subcycles

Reply: revised as indicated.

*P11 Line 17-18: "The kinematic features with TS005 are richer and much narrower, such as the network of shearing in Beaufort Sea." Do you want to say that in TS005 more and finer features are simulated?*

**Reply**: yes, and we revise the sentence as "With TS005, the model simulates more and finer sea ice kinematic features".

P11 Line 35: The region for the analysis you have chosen is problematic as it mixes pack-ice regions with coastal regions. In coastal regions stable deformation features, like flaw lead, are found that show nearly constantly very high deformation rates, which impacts the presented CDFs. I suggest to use the entire Arctic Ocean as study region and filter all grid points that are closer than 150-200km to the coast as done in other scaling studies.

**Reply**: we agree with the referee's comment on the effect of coastal region on the PDF and scaling analysis. We therefore update the results by limiting the analysis within the basin, to the common regions according to TS045 (the coarsest resolution among the three).

P12 Line 3: Please be cautious for two reasons: (1) just because the PDF/CDF of sea ice deformation shows a power-law tail does not mean it is multi-fractal. To show mulit-fractality a scaling analysis of the moments of sea-ice deformation need to be performed that shows a non-linear convex structure function (you do this analysis but it is described later).
(2) The distributions shown in Figure 9 show hardly power-law distributions. I suggest to use the methodology of Clauset et al. (2009) to test for power-law distributions.

**Rep

---

## Author Response (AR2)

The authors would like to express sincere thanks to the two referees and the editor for their invaluable comments and efforts in helping us to improve the revised version of the manuscript. We have made replies to the comments and corresponding revisions, covering those from referee #1 on page 1 through 4, and from referee #2 on page 5 to 10. Besides, a marked up version of the revised manuscript is provided to highlight the specific revisions (==yellow== for the replies to the comments from referee #1, ==cyan== for those from referee #2).

Following up are the replies to the comments from Dr. Nils Hutter (the referee #1). The original comments are in *green italic* font, and the revisions are highlighted in the revised manuscript in ==yellow==.

**Reply to comments of Referee #1:**

*Review of "Comparison of Sea Ice Kinematics at Different Resolutions Modeled with a Grid Hierarchy in Community Earth System Model (version 1.2.1)" from Xu et al. (2020)*
*Nils Hutter*

*The authors have done a good job in answering to all reviewer's comments and revising the manuscript. Thank you for that. The focus of the paper remains as the original but the robustness of the presented analysis and the clarity of the manuscript have been improved significantly by the authors. Major changes include new simulations with the Hibler (1979) ice strength that lead to a more realistic ice distribution and the presented scaling analysis of sea ice deformation has been extended to a 3-month period. With the modifications the paper is a valuable contribution to the ongoing scientific discussion and I recommend the paper for publication after my minor comments are addressed:*

*General comments:*

*1. Scaling analysis: The authors perform a scaling analysis for three winter months (Dec to Feb) and include the new section 3.3 discussing the variability of the scaling exponents during that period, which I appreciate. Why do you not present these new results in Figure 9 (lower row) and Figure 10 (lower row) as a substitute of the analysis of three day intervals. As a reader, I would find this results more interesting. Nevertheless, with Figure 11 you showed that the scaling behaviour is persistent throughout the 3-month period, which answers my main concern raised in the first review.*

**Reply**: the authors thank the referee for the comments on the replies and the revisions we have made to the original version of the manuscript. We consider the wintertime, long-term analysis an independent and outstanding issue, which is beyond the analysis for the typical days (as well as the 3-day analysis surrounding these two typical days). We thank the referee for appreciating our extension of the scaling analysis to the whole winter.

*2. Future work plans: In "Summary and discussion" you state at many places your future research plans (P25: L 9-10, L12-13, L20-23; P27: L1-6,… ). I understand that these plans are the logical*

*next steps from the results presented in this study. However, I recommend to generalise these plans to recommendations to the community where further research is needed. Right now, some parts of the summary reads more as project proposal, which is inappropriate.*

**Reply**: the authors have made extensions to the 3 part of Sec. 4 pointed out by the referee, in order to improve relevance to the community's interests in related topics. Specifically, for P25:L9-10 and P25:L12-13, we revise the sentences as: "*Although with large-scale, coarse atmospheric forcings such as CORE-2, sea ice models can produce multi-fractal sea ice deformation events, how these events are governed by the multi-scale atmospheric processes remain unclear. With different atmospheric forcing datasets such as CORE-2 and JRA-55, we plan to carry out comparative study by using higher versions of the coupled model (i.e., version 2 of CESM). Besides, the dynamical and thermodynamic feedback of the sea ice deformation can also be studied with an atmosphere-sea ice coupled modeling framework, with multi-resolution setting for both the atmospheric and the sea ice component model, as in CESM*".

For P25:L20-23, we revise the sentences as: "*Another issue with modeling sea ice at very high resolution (such as TS005) is the prominent anisotropic characteristics. In order to explore its effect, the anisotropic rheology models such as EAP (Tsamados et al., 2013) can be utilized for a comparative study with standard EVP (or VP) with a very high resolution setting (1 to 2 km in the Arctic Basin). This is planned in our future work with the updated version of the sea ice component (version 5 of CICE) in CESM*".

For P27:L1-6, we make the sentences more concise, as follows: "*For future work, long-term, inter-annually forced simulation of the coupled ocean-sea ice system is planned, under the multi-resolution framework and the spin-up strategy as adopted in this study. Specifically, the comparison with coinciding satellite observations can be carried out such as the RGPS dataset*".

*Specific comments:*

*P1 L3 "sea ice kinematics"*
*add small-scale. Basin-scale kinematics are also simulated at coarse resolution.*

**Reply**: revised as: "…, small scale sea ice kinematics is the most prominent … ".

*P2 L6-8 "and quasi-linear kinematic features by visual inspections (Kwok et al., 2008), including local deformation regions of sea ice failures and shearing."*
*a verb missing in the sentence*

**Reply**: the sentence is revised as: "*and quasilinear kinematic features are observed through visual inspections (Kwok et al., 2008), including local deformation regions of sea ice failures and shearing*".

*P2 L29-30: "Purely Lagrangian models such as neXtSIM (Rampal et al., 2019) are potentially 30 free of the resolution issues for resolving small deformation features."*
*This is not correct. First, neXtSIM shows deformation features as small as the resolution of the*

*grid but it is not resolution independent. Second, the reason for this is likely to be the combination of the MEB rheology generating these LKFs and the Lagrangian advection scheme preserving them. The Lagrangian advection scheme is hardly the only reasons like it sounds in your sentence.*

**Reply**: the authors would like to clarify that by the sentence we mean that Lagrangian models do not suffer from the traditional resolution limitation of Eulerian grid based models, since there is a hard limitation (grid resolution) of these models due to the smallest grid size is fixed. So this sentence only refers to the resolution aspects, not the rheology part. On the other hand, even with MEB rheology, we would argue that Eulerian grid-based models still suffer from the resolution limitation in resolving very small deformation events.

*P2 L33 "(Hutter and Losch, 2020),"*
*This paper does not comment on potential violations of continuum assumption in high resolution simulations. The issue with deformation features in VP models is also in my opinion rather the missing memory of past deformation (as discussed in the cited paper) than the continuum assumption.*

**Reply**: the whole sentence is revised as follows: "*Traditional VP rheology is limited due to the lack of memory for past deformation events (Hutter and Losch, 2020). Besides, there are also efforts in improving the rheology model in simulating observed anisotropy in sea ice floe shape and associated deformation (Tsamados et al., 2013).*"

*P7 L10 "(Lipscomb et al., 2007; Hibler, 1979)"*
*I recommend the following to clearly state which ice strength definition you are using: ridging/rafting scheme (Lipscomb et al., 2007) and ice strength model (Hibler, 1979)*

**Reply**: the references are differentiated for the two schemes in the revised sentence.

*P12 L32: "deformation caused by numerics"*
*What is this? Do you mean noise generated by the EVP solver? Please be more specific.*

**Reply**: to be more specific, we revise the term as: "*the deformation caused by numerical issues, such as limitation of grid resolution and non-convergent solution*".

*P8 L33: "As a consequence"*
*The effective resolution is coarser than the grid resolution because a few grid cells are needed to represent steep gradients in the drift fields, not due to numeric noise. At least in converged solutions. Please clarify.*

**Reply**: we revise the previous sentence to include the two major aspect we intend to include, the limitation of grid resolution and non-convergent solution to the VP problem.

*P20 Figure11: Please use the same limits for both y-axis. Since you are showing the running mean of the same data in the second y-axis, different limits are confusing and misleading.*

**Reply**: we have revised the figure to align the y-axes, as indicated by the referee.

*P21 L28-29: "multi-fractal sea ice deformation is accurately modeled by all three resolutions"*
*Rephrase to: "the modelled sea ice deformation is characterised by multi-fractal scaling". Since you did not compared to observed multi-fractal characteristics "accurately" is misleading.*

**Reply**: rephrased as: "*..., the modelled sea ice deformation is characterized by multi-fractal scaling for all three grid resolutions*".

*P22 L8: is -> are*

**Reply**: corrected.

*P22 L9-10: "Specifically, the process dependent scaling 10 properties were found for representative days for sea ice drift and AO index."*
*Unclear what process dependent properties are. Please rephrase and clarify.*

**Reply**: we revise the sentence to improve its clarity: "*Specifically, we have observed time-varying scaling properties, which the scaling coefficient correlates with the leading mode of atmospheric forcing (i.e., Arctic Oscillation)*".

*P25 L27-28: "Based on a traditional implementation of EVP in our model, we have witnessed asymptotic and converging behavior of modeled kinematics by the increasing EVP subcycle count."*
*Two times the same sentence, please remove one*

**Reply**: revised by removing one instance of the sentence.

**Reply to comments of Referee #2:**

*General comments:*

*This is the revised manuscript accompanied by responses to the reviews of the original manuscript. The authors present the results of one component of a new earth system, namely the sea-ice. They first introduced a new grid generator for a tripolar grid. Then, they show initial results of the sea-ice component under normal-year-forcing (a kind of climatological forcing with synoptic scales sampled from a specific year) and different resolutions, including multi-scale analysis of the sea-ice deformation. Finally, they show the sensitivity of the dynamics of the model to the grid resolution and a long-debated parameter in the dynamics, namely the number of subcycling used in solving the viscous plastic equations.*

*The manuscript has improved in general but the English needs to be polished once more. The experiments are well described. The results are well presented and analyzed using existing diagnostic tools. More context for what the authors are aiming at would help however the reader as well.*

**Reply**: the authors thank the referee for the invaluable comments and the suggestions in improving the language usage of our paper. We have made the following revisions and replies accordingly.

*1-I understand that the motivation in section 3.2 was to claim spatial multi-fractality in CICE which was originally disproved by Girard et al. (2009) => need to refer and discuss their results. Moreover, since the deformation field has not converged yet with respect to the NDTE parameter for TS015 and TS005 (ndte=240 in this section), I am curious to know the sensitivity to the parameter. In passing, I note that the sign of beta varies from negative to positive without clear explanations...*

**Reply**: regarding the study with EVP and CICE in Girard et al. (2009), we would like to clarify that there is distinct differences between the model settings in Girard et al. (2009) and ours, including model resolution, period of study, as well as EVP convergence. We do take extra caution when using the model output on the original grid resolution to study statistics. In Girard et al., (2009), the authors do find better consistency of deformation rate PDFs when using scaled results of the model outputs, which is potentially free from the issues involving limited effective model resolutions. According to the referee's suggestion, we have added extra discussion in Sec. 4, relating to Girard et al., (2009).

Regarding the convergence regarding to subcycling of EVP, we haven't examined the scaling properties with lower NDTE values (i.e., smaller than 960). Currently we treat the two aspect as independent issues. With non-convergent EVP solutions, the scaling properties of the deformation rates are expected to change, but unfortunately we haven't explored this issue yet. Regarding beta values close to or higher than 0 on some days (Fig. 11), we would like to argue that these values are statistically insignificant from 0, which are usually days associated with very large, non-localized deformation events. Also the analysis method (Eulerian instead of Lagrangian) is shown to affect the scaling analysis as well (Hutter et al., 2018), and we are preparing to adopt a Lagrangian framework for a better characterization of these extreme days.

*2-About the grid generation, I am grateful that the authors gave their rationale for developing their own generator, however this is done only in their response and is not reflected in the text. The reader is missing that context and is still left in the black. Notably the reference to the pioneered work of others is left to the appendix which leaves the wrong impression that the authors claim for the own the technique in the main part of the text. In their response, when comparing to the tri-polar ORCA grids, the authors imply that POP/CICE use a different and unique "U-fold", which I believe is a misconception. The POP manual (top page 46 of https://www.cesm.ucar.edu/models/cesm1.2/pop2/doc/sci/POPRefManual.pdf) does acknowledges that their tripolar grid is a courtesy of Madec (i.e. the ORCA grid). The only differences between ORCA and POP grids are in fact just technical (ORCA grids include a redundant row of points along the western, eastern and northern edges; ORCA grids accommodate both, what they call, F- or T-folds whereas POP grids exclusively rely on the F-folding technique, which the authors refer to "U-fold"). POP Manual also refers to Murray (1996) given below.*

**Reply**: regarding the referee's comment, we have now added the necessary references during Sec. 2.1 for Murray (1996) and Madec (1996). We would like to clarify for that we do not intend to make the wrong impression of us inventing the grid generation technique, and apologize for any cases for potentially conveying the wrong message.

We do agree with the referee that the treatment of the grid's north poles is consistent between POP grids and ORCA. However, we would also like to highlight that for current POP grids (e.g., TX1 and TX0.1) and ORCA, they are inherently different: on the "turning latitude" that meridional grid sizes in POP grids are NOT continuous, while those for ORCA are.

*3-The authors in their response state that they found that their sea-ice state was more realistic with a change of ice strength formulation. I would refer them to Ungermann et al. (2017) where there is a discussion on the subject of Hibler (1979) vs Rothrock (1975).*

**Reply**: the authors thank the referee for pointing out Ungermann et al. (2017), in which MITGcm model is utilized to study the differences between Hibler (1979) (or H79) and Rothrock (1975) (or R75) for ice strength parameterization. Indeed, although a different model is used in our study, qualitatively with H79 our model simulates better ice thickness distribution than R75. The reference to Ungermann et al. (2017), and a reference is added in Appendix B. Besides, we plan to carry out detailed analysis attributing the differences we have witnessed in the future.

*4-"EVP rheology": Careful with using EVP as it were a different rheology in itself. The rheology remains VP (the constitutive laws) but EVP introduces artificially slow elastic waves for numerical (claimed) convergence. It is actually very akin to the pseudo-compressible method that was once in favour in computational fluid dynamics for solving for incompressible flows. Hunke use "EVP model", Koldunov et al. "EVP solution" and Bouillon et al. (2013) "EVP method".*

**Reply**: we have revised "EVP rheology" into "EVP model", as indicated by the reviewer.

*5-ndte=960 does not seem to be enough in TS005 as we still see some blue in the CAA (fig.13), whereas, in the coarser resolution runs, the region is fully white. Same can be said for the central Arctic where it takes an increasing ndte to get the equivalent pattern in TS045, TS015 and TS005 respectively (.e.g the pattern of (TS005, ndte=240) is equivalent to (TS015,ndte=120), and that of (TS015, ndte=960) is in between to (TS045, ndte=240,960)). Same can be said from Fig.12. By the way, I like very much the colour scale in fig. 13, much easier to see patterns.*

**Reply**: As pointed out by the referee, NDTE=960 is arguably not sufficient for the deformation fields to reach (asymptotic) convergence, as evident in Fig. 13 as well as Fig. 12. We have carried out the same experiment with higher NDTE values (i.e., 1920), and indeed the deformation fields are further "cleaner" than that for NDTE=960. Therefore, we argue that with even higher resolution, with the version of EVP solver we adopt, NDTE values should be further enlarged to ensure convergence.

Besides, as pointed out by the referee, we have had intentionally picked a more proper colorbar in highlighting the differences in deformation rates between the range of 0.25%/day and 2%/day.

*6-ndte is not indicated in fig.7-8-9. There is only one mention in the text that ndte=240 is used for all comparison of the ice deformation and scaling analysis until the next section. I would appreciate a reminder in the captions for fast browsing readers like me who like to go and forth through a paper and its figures.*

**Reply**: we would like to clarify that, for all the scaling analysis in Sec. 3.2 and Sec. 3.3, we use the model output with NDTE=960. This is pointed out in Sec. 3.2, P12:L5-6. But in Sec. 3.1, since long-term, >30-yr experiments are needed to analyze the spin-up process, for which only NDTE=240 runs are available, the analysis of basin-wide thickness and seasonal cycles are carried out (e.g., Fig. 5 and related texts). The referee is kindly guided to Fig. 4 for the specific experiments and their configurations.

Furthermore, in order to elucidate these details and avoid misunderstanding, we now have emphasized that we have used NDTE=960 results in the captions of Fig. 7 through 11.

*7-Missing Figure S4?*

**Reply**: the authors sincerely apologize for the missing of Fig. S4, due to uploading an older version of the supplementary. We have updated the supplementary (which contain updated figures, and newly added ones of Fig. S4 and Fig. S5). For a quick view, we have attached Fig. S4 as below, showing the geolocations in CAA where landfast ice manifest in our model.

[Figure]

*Fig. S4 Two-week mean sea ice velocity in Canadian Arctic Archipelago. First two weeks (Jan. 1st to Jan. 14th) of model output with TS005 is used to illustrate the region of landfast sea ice during winter.*

**Reply**: details of the model parameterization for atmosphere/ocean-ice drag are added now in Appendix B and Tab. B1.

*9-Appendix B: should you take into account grid deformation in your calculation of deformation?*

**Reply**: we confirm that we have taken into account the different grid scales during the line integrals computations. Regarding the grid deformation, the largest spatial scales during scaling analysis (Fig. 10) is about 200~300km, at which the grid deformation is still very small in terms of isotropy and orthogonality, we do not consider it an important issue in our analysis. However, when the grid is highly deformed, it should be incorporated in the analysis, by adjusting the analyzed region to maintain a generally isotropic and convex shape.

*minor comments:*

*why TS005 on year 31 (or end of year 30) does not match that of TS015 in fig.4?*

**Reply**: we would like to clarify that, the SIE and SIV values in Fig. 4 are annual-mean values. During the transitions from TS045 to TS015 and TS015 to TS005, the model inherently undergoes a shock, for which dynamic and thermodynamic processes are both contributive factors. The sudden jump is also witnessed between TS045 and TS015 (on year 26), which is only less evident than that between TS015 and TS005 (on year 30).

Therefore, these jumps are also of a technical causes. If we had plotted daily mean SIE or SIV, the shock is gone. But we choose to plot annual mean values, to avoid very busy graphs with all the

seasonal cycles of over 30-year's data.

*Abstract, line 5: I would "the" before "Community Earth System Model"*

**Reply**: we have added "the" as indicated by the referee.

*Abstract, line 8. "In specific" does not sound English*

**Reply**: we revised it into "Specifically".

*page 13, line 17-21: are still talking about fig.9? The term "cumulative" has been dropped here.*

**Reply**: revised by adding "cumulative" before "PDF".

*page 13, line 23 one instance of "CDF" remains (the others have been converted to "cumulative PDF") I don't think that the figure 9 has been corrected in that regard.*

**Reply**: revised by replacing "CDF" with "cumulative PDF". The caption of Fig. 9 is also revised accordingly.

*page 13, line 24-35: "Also, since we witness flatter cumulative PDF slopes in scaled datasets, we expect the "real" tails of cumulative PDFs at 2.4 km flatter than the modeling result from TS005, which have the slope of -1.0 for Dec. 20th, and -0.5 for Feb. 6th." Honestly, I don't understand the sentence anymore.*

**Reply**: we apologize for not conveying the message properly. The sentence is revised as follows: "*Also, since we witness flatter cumulative PDFs for the scaled results of TS005 than TS015, we expect that at the spatial scale of 2.4 km (TS005's native resolution), the physical cumulative PDFs are flatter than the model output of TS005, which have the slope of -1.0 for Dec. 20th, and -0.5 for Feb. 6th.*"

*page 13, line 25 "we expect the "real" tails of cumulative PDFs at 2.4 km flatter than..." missing "to be" before "flatter"*

**Reply**: added "to be", as indicated by the referee.

*page 13, line 27 "This is due to the temporal averaging on Eulerian grid points attenuates" misses a "which" before "attenuates"*

**Reply**: added missing "which" before "attenuates".

*page 14, line 24 "sea ice status" I think the authors meant "sea ice state"*

**Reply**: revised. See also the reply below.

*page 14, line 25-36: the whole following sentence "Since the experiments in this study target at climatologlical sea ice states" could be phrased better (these experiment in this study aim at obtaining a climatologlically converged sea ice state?)*

**Reply**: the sentence is revised as: "*the experiment in this study aim at obtaining a converged sea ice state that reflect reasonable Arctic climatology, ...*".

*CORE instead of CORE2 in appendix B, line 21.*

**Reply**: corrected.

*References:*

*Girard, Lucas, et al. "Evaluation of high-resolution sea ice models on the basis of statistical and scaling properties of Arctic sea ice drift and deformation." Journal of Geophysical Research: Oceans 114.C8 (2009).*

*Ungermann, Mischa, et al. "Impact of the ice strength formulation on the performance of a sea ice thickness distribution model in the A rctic." Journal of Geophysical Research: Oceans 122.3 (2017): 2090-2107.*

*Murray, Ross J. "Explicit generation of orthogonal grids for ocean models." Journal of Computational Physics 126.2 (1996): 251-273*

*Hunke, Elizabeth C. "Viscous–plastic sea ice dynamics with the EVP model: Linearization issues." Journal of Computational Physics 170.1 (2001): 18-38.*

*Bouillon, Sylvain, et al. "The elastic–viscous–plastic method revisited." Ocean Modelling 71 (2013): 2-12.*

*Koldunov, Nikolay V., et al. "Fast EVP Solutions in a High-Resolution Sea Ice Model." Journal of Advances in Modeling Earth Systems 11.5 (2019): 1269-1284.*